# The Boundaries of Fair AI in Medical Image Prognosis: A Causal Perspective

**Thai-Hoang Pham[1,2], Jiayuan Chen[1,2]\*, Seungyeon Lee[1,2]\*, Yuanlong Wang[1,2]\***
**Sayoko Moroi[3], Xueru Zhang[1], Ping Zhang[1,2]†**
[1]Department of Computer Science and Engineering, The Ohio State University
[2]Department of Biomedical Informatics, The Ohio State University
[3]Department of Ophthalmology and Visual Sciences, The Ohio State University
{pham.375,chen.12930,lee.10029,wang.16050}@osu.edu
{moroi.4,zhang.12807,zhang.10631}@osu.edu

## Abstract

As machine learning (ML) algorithms are increasingly used in medical image analysis, concerns have emerged about their potential biases against certain social groups. Although many approaches have been proposed to ensure the fairness of ML models, most existing works focus only on medical image diagnosis tasks, such as image classification and segmentation, and overlooked prognosis scenarios, which involve predicting the likely outcome or progression of a medical condition over time. To address this gap, we introduce FairTTE, the first comprehensive framework for assessing fairness in time-to-event (TTE) prediction in medical imaging. FairTTE encompasses a diverse range of imaging modalities and TTE outcomes, integrating cutting-edge TTE prediction and fairness algorithms to enable systematic and fine-grained analysis of fairness in medical image prognosis. Leveraging causal analysis techniques, FairTTE uncovers and quantifies distinct sources of bias embedded within medical imaging datasets. Our large-scale evaluation reveals that bias is pervasive across different imaging modalities and that current fairness methods offer limited mitigation. We further demonstrate a strong association between underlying bias sources and model disparities, emphasizing the need for holistic approaches that target all forms of bias. Notably, we find that fairness becomes increasingly difficult to maintain under distribution shifts, underscoring the limitations of existing solutions and the pressing need for more robust, equitable prognostic models.

## 1 Introduction

Machine learning (ML) algorithms trained on real-world medical images may inherently exhibit bias, leading to discrimination against certain social groups [57, 42]. This is especially concerning in the medical field, where biased algorithms can result in inequitable treatment recommendations, misdiagnoses, or unequal access to care [46]. Therefore, ensuring the fairness of ML models and identifying hidden biases within medical images is critical for advancing health equity [54].

Existing research on fairness in ML and medical image analysis has primarily focused on medical image diagnosis tasks, such as image classification and segmentation [82, 64]. These diagnostic tasks typically aim to determine the presence or absence of a condition. However, prognosis scenarios, which involve predicting the likely outcome or progression of a medical condition over time, have been

---

\*Equal contribution authors.
†Corresponding author.

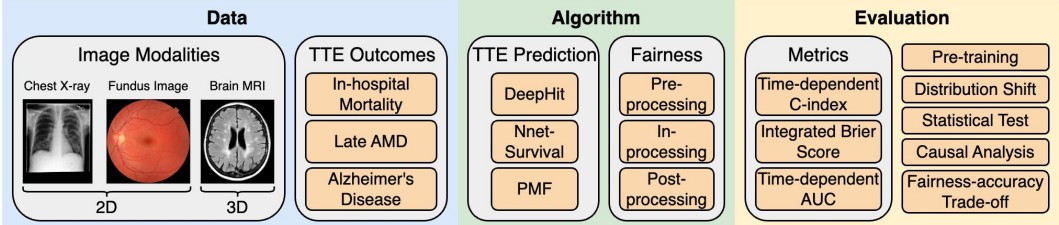

Figure 1: An overview of the FairTTE, a unified framework designed to investigate fairness in TTE prediction for medical image analysis.

largely overlooked. In ML, prognosis is framed as time-to-event (TTE) prediction or survival analysis, where the goal is to predict the time until a critical event. Unlike classification, TTE prediction provides a richer and more dynamic approach to modeling medical outcomes by accounting not only for the presence or absence of a condition but also for the timing and progression of key health events. This makes it especially valuable for predicting long-term outcomes such as survival or disease recurrence [21]. Despite its importance, there remains a significant gap in the literature regarding fairness in TTE prediction, particularly in the context of medical imaging (see Appendix A).

Developing fairness methods for TTE prediction in medical imaging presents several unique challenges compared to tasks like image classification or segmentation. First, there is a lack of public datasets that pair medical images with TTE outcome information. While many public datasets include sensitive attributes, they primarily focus on diagnostic tasks and do not provide the temporal data necessary for training TTE prediction models. Second, there is limited understanding of how biases in medical images specifically affect the fairness of TTE prediction models, complicating the development of fair algorithms. Third, the absence of a universally accepted fairness metric for TTE prediction poses a challenge in assessing and comparing fairness across studies [36, 53, 79]. Existing fairness metrics are often not well-suited to the complexities of medical applications (see Section 2.1). Furthermore, varying experimental designs and evaluation approaches in the literature hinder direct comparisons between fairness methods, leading to fragmentation in the field and impeding the establishment of best practices for fair TTE prediction in medical imaging.

To address the challenges of understanding fairness in TTE prediction for medical imaging, we introduce FairTTE, the first comprehensive framework specifically designed for this task (Figure 1). Our approach begins by applying causal reasoning to develop a general framework for understanding and quantifying bias in TTE prediction. This framework decomposes the data likelihood into distinct components, enabling a fine-grained analysis of various sources of bias, including disparities in image features, censoring rates, TTE labels, mutual information between images and TTE, and mutual information between images and censoring indicators. This principled decomposition allows us to explain why standard TTE prediction methods and fairness algorithms frequently fail to mitigate bias in practice (see Section 2.2). Using this framework, we characterize not only the presence but also the type and extent of bias across different datasets, offering a detailed understanding of fairness challenges in medical image prognosis. To complement the framework, we construct a collection of large-scale, publicly available medical image datasets tailored for fair TTE prediction. These datasets span a wide range of imaging modalities, including fundus images, chest X-rays, and brain MRIs, and cover diverse TTE outcomes, such as mortality, late age-related macular degeneration (AMD), and Alzheimer's disease. Finally, leveraging both the datasets and causal framework, we establish a realistic and comprehensive investigation study for fairness in TTE prediction. This investigation incorporates state-of-the-art (SOTA) algorithms across various learning strategies and, to our knowledge, represents the first attempt to provide a fine-grained analysis of fairness in TTE prediction, making a significant contribution to the growing field of fair ML in medical imaging.

Our work also explores fair TTE prediction methods across various settings, considering factors such as pre-training and distribution shifts that may influence model fairness. Building on our causal framework and through the training of over 20,000 models, we derive several key insights to inform future research in this area, as follows:

- Bias is prevalent across TTE prediction models trained on different imaging modalities, with consistent performance disparities observed between demographic groups.
- All medical image datasets examined in our study exhibit some sources of bias that adversely affect the fairness of TTE prediction models.

- Most SOTA fairness methods struggle to consistently mitigate bias—while they can improve fairness in some settings, these gains are often accompanied by reductions in predictive accuracy.
- Pre-training improves model accuracy but has minimal impact on fairness.
- Different types of distribution shifts affect fair TTE prediction in distinct ways, underscoring the growing difficulty of maintaining both fairness and utility under realistic clinical scenarios.

## 2 Unified Framework for Fair TTE Prediction

### 2.1 Fair TTE Prediction Setup

We first introduce the notations used throughout the paper and then formulate the fair TTE prediction task. In this study, we focus on analyzing methods designed for achieving group fairness in right-censored TTE prediction.

**Notations.** A TTE dataset contains observations for each individual along with their corresponding (right-censored) TTE outcomes. Specifically, each individual's data is represented by a tuple of random variables (RVs) $(X, Y, \Delta, A)$, where $X$ denotes a set of features, $Y$ represents an observed time, $\Delta$ is an event indicator, and $A$ is a sensitive attribute for the individual. If $\Delta = 1$ (indicating that the event has occurred), then $Y$ represents a true survival time; otherwise, $Y$ is a censoring time. We denote $T$ as a true (possibly unobserved) survival time associated with $X$, and $C$ as a true (possibly unobserved) censoring time associated with $X$. We use lowercase letters $x, y, \delta, a, t, c$ to denote the respective realizations of these random variables. It is important to note that for each individual, we do not observe both $T$ and $C$; instead, we observe exactly one of them. More precisely, $Y = \min\{T, C\}$ and $\Delta = \mathbb{1}\{T \leq C\}$, where $\mathbb{1}$ is an indicator function. Let $S(t'|x) = 1 - \int_0^{t'} P(t|x)dt$ represent the survival function at time $t$ given the feature $x$. In fair TTE prediction, our goal is to accurately predict $S(t'|x)$ while adhering to fairness constraints.

**Performance Metrics for TTE Prediction.** Consider a TTE prediction model $h : \mathcal{X} \to \mathcal{T}$ where $\mathcal{X}$ and $\mathcal{T}$ are the spaces of $x$ and $t$, we denote the performance metric for this model on the dataset $D = \{X_i, Y_i, \Delta_i, A_i\}_{i=1}^{|D|}$ as $\mathrm{Er}(f, h, D)$ where $f$ is the ground-truth labeling function (i.e., $f = P(t|x)$). Note that this notation is sufficiently flexible to encompass censoring data and different metrics in TTE prediction. Furthermore, while prediction error (smaller values indicate better performance, *e.g.*, Brier score [16]) is used as the metric in our theoretical analysis, prediction accuracy (larger values indicate better performance, *e.g.*, concordance index [19]) can also be applied.

**Fairness Metrics for TTE Prediction.** Various fairness metrics have been proposed recently for TTE prediction, which can be roughly classified into three categories based on their objectives: (i) ensuring similar predicted TTE outcomes for similar data points [36, 53, 77, 76, 72], (ii) ensuring similar predicted outcomes for data points from different groups [36, 53, 80], and (iii) ensuring similar predictive performance across different groups [11, 22, 78, 79]. Notably, some of these metrics are less applicable in the context of medical imaging. For instance, metrics in the first category require a well-defined similarity measure between data points, which is difficult to establish for medical images. Metrics in the second category may also be inappropriate when sensitive attributes are strong risk factors for the TTE outcome. For example, age is often a key predictor of various TTE outcomes, such as mortality. In this case, asking for similar predicted survival times for young and elderly individuals would be nonsensical.

Therefore, in this study, we focus on fairness metrics from the third category, which aim to ensure that the model maintains equal predictive performance across different groups. Specifically, given a performance metric $\mathrm{Er}$ for TTE prediction task, we define fairness metric $\mathcal{F}_{\mathrm{Er}}$ as follow:

$$\mathcal{F}_{\mathrm{Er}}(h) = \max_{a,a' \in \mathcal{A}} |\mathrm{Er}(f_a, h, D_a) - \mathrm{Er}(f_{a'}, h, D_{a'})| \tag{1}$$

where $\mathcal{A}$ is the set of groups considered in TTE prediction task, $D_a = \{X_i, Y_i, \Delta_i, A_i | A_i = a\}_{i=1}^{|D_a|}$ and $D_{a'} = \{X_i, Y_i, \Delta_i, A_i | A_i = a'\}_{i=1}^{|D_{a'}|}$ are the subsets containing data from groups $a$ and $a'$, and $f_a = P(t|x, a)$ and $f_{a'} = P(t|x, a')$ are the ground-truth labeling functions for $D_a$ and $D_{a'}$, respectively. If the labeling function $f$ is independent of the sensitive attribute, then $f = f_a = f_{a'}$. Note that the fairness metric in Eq. (1) is defined as the maximum performance gap between any two groups, and as such, it can be applied to any performance metric used in TTE prediction.

## 2.2 Causal Structure for Fair TTE Prediction

To conduct a fine-grained analysis to understand fairness in TTE prediction, we leverage the structural causal model (SCM) [48] to represent the data generation process underlying TTE data. Specifically, we examine how the sensitive attribute $A$ influences the TTE prediction model, which aims to approximate the data likelihood $P(t|x)$ using the SCM framework. To construct the causal graph, we introduce an unobserved underlying health condition of the patient, denoted as $Z$. In this context, the patient's feature set $X$ can be viewed as noisy and partially observed information derived from $Z$.

Following the approach in [25, 69], we partition $X$ into two components: $X_Z$, representing target-related features directly influenced by $Z$, and $X_A$, representing features related to the sensitive attribute, directly influenced by $A$. By construction, $X_A$ encodes sensitive information as it is predictive of the sensitive attribute $A$. Using these definitions, we construct causal graphs for TTE data under two settings: unbiased and biased scenarios, illustrated in Figure 2. In the unbiased scenario (Figure 2a), the sensitive attribute $A$ is irrelevant to the TTE outcome. It only influences $X_A$, and not any other variable in the graph. Therefore, capturing the in-

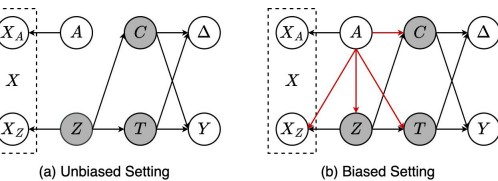

Figure 2: Causal structure in TTE prediction. Gray circles represent unobserved RVs. (a) Unbiased setting, where the sensitive attribute $A$ affects only $X_A$. (b) Biased setting, where the sensitive attribute $A$ may be correlated (red arrow) with other RVs in causal graph.

variant feature $X_Z$ across groups is sufficient for learning a fair model. Specifically, we have, $P(t|x_z) = P(t|x_z, a), \ \forall a \in \mathcal{A}$. In contrast, in the biased scenario (Figure 2b), the sensitive attribute $A$ influences additional variables in the causal graph beyond $X_A$, leading to dependency between $A$ and TTE outcome. This results in the conditional distribution of the outcome given $X_Z$ varies across groups $P(t|x_z, a) \neq P(t|x_z, a'), a, a' \in \mathcal{A}$ and/or the distribution of $X_Z$ differs across groups $P(x_z|a) \neq P(x_z|a')$. These discrepancies reflect the presence of bias in the data and challenge the assumption that learning invariant features alone is sufficient for fair TTE prediction.[1]

It is important to note that disparities in data distributions across groups, as reflected in the causal graph, do not necessarily indicate that the sensitive information captured during model training is spurious or inappropriate in every context. In some cases, genuine biological differences between groups may exist, making the sensitive attribute relevant for disease prediction. In such situations, incorporating group-specific information can be beneficial, as it allows the model to capture distinct disease mechanisms that reflect true underlying biological variation.

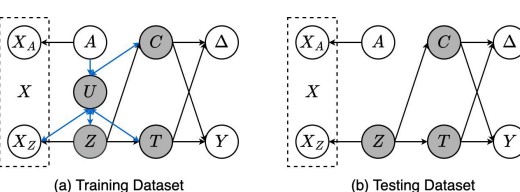

Figure 3: Causal structure in TTE prediction under distribution shift. We illustrate a scenario where an unfair causal pathways (blue arrows), induced by unobserved RV $U$, are present in train data (a) but absent in test data (b), leading to distribution shift. Bidirectional arrows indicate that the causal direction may vary depending on the specific context. Fair causal pathways (appear in both train and test data) may exists but are omitted for simplicity.

However, observed biases in the training data may arise from spurious correlations that reflect historical inequities in healthcare access, diagnosis, and treatment practices (illustrated by the blue arrows in Figure 3a). When models are trained on such data, they risk encoding and amplifying these biases, thereby perpetuating unfairness in clinical decision-making. From this perspective, fairness and distribution shift are intrinsically connected: achieving fairness requires models that can generalize to a test distribution in which unfair causal pathways are absent and only fair causal pathways remain (Figure 3b) [37, 71, 62].[2]

**Group-specific Distribution Disparity and Fairness.** As shown in Figure 2b, the effect of $A$ on other RVs can induce distribution disparity across groups. We now formalize how it affect fairness in Theorem 1 below.

---

[1]We acknowledge the possibility of unobserved RVs that may influence both sensitive attribute $A$ and other RVs (i.e., $X, Z, T, C$) in causal graph. However, since our primary objective is to characterize disparities in group-specific data distributions, we omit unobserved RVs from the causal graph for simplicity (see Appendix G.1).

[2]Real-world medical examples of fair and unfair causal pathways are provided in Appendix G.2.

**Theorem 1.** *Given a performance metric* Er *satisfying triangle inequality and symmetry properties, i.e.,* $|\mathrm{Er}(h, h', D) - \mathrm{Er}(h, h'', D)| \leq \mathrm{Er}(h', h'', D)$ *and* $\mathrm{Er}(h, h', D) = \mathrm{Er}(h', h, D)$*, then we have:*

$$\mathcal{F}_{\mathrm{Er}}(h) \leq \max_{a, a' \in \mathcal{A}} \left( \eta(\mathcal{H}, f_a, f_{a'}) + \mathcal{D}(\mathcal{H}, D_a, D_{a'}) \right), \quad \forall h \in \mathcal{H} \tag{2}$$

*with*

$$\eta(\mathcal{H}, f_a, f_{a'}) = \min_{h' \in \mathcal{H}} \left( \mathrm{Er}(f_a, h', D_a) + \mathrm{Er}(f_{a'}, h', D_{a'}) \right);$$

$$\mathcal{D}(\mathcal{H}, D_a, D_{a'}) = \max_{h', h'' \in \mathcal{H}} |\mathrm{Er}(h', h'', D_a) - \mathrm{Er}(h', h'', D_{a'})|$$

where $\mathcal{H}$ represents the hypothesis class of the TTE prediction model $h$, $\eta(\mathcal{H}, f_a, f_{a'})$ denotes the minimum joint prediction error on $D_a$ and $D_{a'}$, and $\mathcal{D}(\mathcal{H}, D_a, D_{a'})$ is the largest distance between two groups $a$ and $a'$, with respect to the hypothesis class $\mathcal{H}$. Note that the requirements on symmetric and triangle inequality properties are relatively mild and can be met by many performance metrics commonly used in practice (see Appendix B).

**Discussion of Theorem 1.** The term $\mathcal{D}(\mathcal{H}, D_a, D_{a'})$ aligns with the concept of subgroup separability—the ability to predict group membership from image features—in the group fairness literature [30]. Low subgroup separability implies that the image feature distributions are similar across groups, which corresponds to a small value of $\mathcal{D}(\mathcal{H}, D_a, D_{a'})$. Prior work has shown that unfairness tends to be less severe in datasets with low subgroup separability [30, 73]. Our results extend these findings by demonstrating that this relationship also holds in the context of TTE prediction, while further revealing that model fairness is influenced by additional factors beyond subgroup separability. We observe that while it is often possible to reduce $\mathcal{D}(\mathcal{H}, D_a, D_{a'})$ during training—for example, by learning fair representations across groups—minimizing the term $\eta(\mathcal{H}, f_a, f_{a'})$ may remain challenging. In particular, when the labeling functions $f_a$ and $f_{a'}$ differ significantly across groups (Figure 2b), even the optimal choice of hypothesis class $\mathcal{H}$ may not ensure a small $\eta$. In such cases, the upper bound on fairness error becomes large, implying that $\mathcal{F}_{\mathrm{Er}}(h)$ may also be large. Conversely, when $f_a$ and $f_{a'}$ are similar—as in Figure 2a or when the only causal pathway from $A$ is to $X$ in Figure 2b—the upper bound becomes small, indicating that $\mathcal{F}_{\mathrm{Er}}(h)$ can also be small. In these settings, fairness can be achieved by learning invariant representations across groups. We formally state this observation as follows.

**Proposition 2.** *Let* $g : \mathcal{X} \to \mathcal{Z}$ *be a mapping from the input space* $\mathcal{X}$ *to a representation space* $\mathcal{Z}$. *Assume the following conditions hold:*

*(i) Data distribution disparity across groups is a covariate shift, i.e.,* $P(t|x, a) = P(t|x), \forall a \in \mathcal{A}$.

*(ii) Representation* $\mathcal{Z}$ *is sufficient for an arbitrary group* $a \in \mathcal{A}$*, i.e.,* $I_a(Z, T) = I_a(X, T)$*, where* $I_a(\cdot, \cdot)$ *denotes the mutual information computed over the distribution* $D_a$.

*Then, distribution shift across groups w.r.t. representation* $\mathcal{Z}$ *is also a covariate shift, that is,*

$$P(t|z, a) = P(t|z), \quad \forall a \in \mathcal{A}.$$

**Discussion of Proposition 2.** This proposition suggests that under covariate shift $(i)$, fair TTE prediction can be achieved by learning a fair representation $Z$ across groups (i.e., $P(z|a) = P(z), \forall a \in \mathcal{A}$). More clearly, $P(z|a) = P(z), \forall a \in \mathcal{A}$ guarantees small $\mathcal{D}(\mathcal{H}, D_a, D_{a'})$ while $P(t|z, a) = P(t|z), \forall a \in \mathcal{A}$ indicates small $\eta(\mathcal{H}, f_a, f_{a'})$. We note that the sufficiency condition imposed on $Z$ $(ii)$ is practical: it only needs to hold for one group, and during training we have access to labeled data. Moreover, the dimension of $T$ is often smaller than that of $Z$, making the sufficiency assumption more attainable in practice.

To further investigate the influence of sensitive attributes on the fairness of TTE prediction models, we utilize our proposed causal framework to decompose the group-specific labeling functions. Following Bayes' theorem, we rewrite the labeling function $f_a$ w.r.t. each group $a$ as:

$$P(t|x, a) = \frac{P(x|t, a)P(t|a)}{P(x|a)} = \frac{P(x_a|x_z, t, a)}{P(x_a|x_z, a)} \cdot \frac{P(x_z|t, a)}{P(x_z|a)} \cdot P(t|a) = \frac{P(x_z|t, a)}{P(x_z|a)} \cdot P(t|a) \tag{3}$$

Note that in practical TTE prediction scenario, we observe $Y$ and $\Delta$ rather than $T$. Therefore, the TTE prediction model estimates $P(y, \delta|x)$ instead. Replacing $t$ with $y$ and $\delta$ in Eq. (3), we obtain:

$$P(y, \delta|x, a) = \frac{P(x_z|y, \delta, a)}{P(x_z|a)} \cdot P(y, \delta|a) = \underbrace{\frac{P(x_z|y, \delta, a)}{P(x_z|\delta, a)}}_{\mathrm{PMI}(x_z, y)} \cdot \underbrace{\frac{P(x_z|\delta, a)}{P(x_z|a)}}_{\mathrm{PMI}(x_z, \delta)} \cdot \underbrace{P(y|\delta, a)}_{\mathrm{TTE}} \cdot \underbrace{P(\delta|a)}_{\mathrm{censoring}} \tag{4}$$

where the first and second terms represent the pointwise mutual information (PMI) between $X_Z$ and $Y$ (conditioned on $\Delta$), and between $X_Z$ and $\Delta$, respectively. The third term corresponds to the TTE distribution, while the fourth term accounts for the censoring rate. As shown in Figure 2b, when $A$ is correlated with other random variables in the causal graph, it alters these four terms, resulting in $P(y, \delta|x, a) \neq P(y, \delta|x, a')$.

**Sources of Bias**[3] **in Fair TTE Prediction.** Based on Theorem 1 and the decomposition formula in Eq. (4), we can identify five primary sources of bias across groups in fair TTE prediction: 1) disparity in image feature distributions, 2) disparity in mutual information between $X_Z$ and $Y$, 3) disparity in mutual information between $X_Z$ and $\Delta$, 4) disparity in the TTE distributions, and 5) disparity in the censoring rates. In practice, we observe that TTE datasets often contain multiple sources of bias rather than just one. These five cases represent the fundamental sources of bias and are crucial for understanding the complex biases present in real-world TTE data.

## 3 Experimental Setup for Fair TTE Prediction

We propose FairTTE, a reproducible and user-friendly framework for evaluating fairness algorithms in TTE prediction within medical imaging. Our framework includes large-scale experiments conducted on three real-world medical image datasets, covering diverse imaging modalities and TTE outcomes, with up to three sensitive attributes considered for each dataset. Using these datasets, we evaluate three TTE prediction models and five fairness algorithms in the context of fair TTE prediction.

**Datasets.** FairTTE includes **MIMIC-CXR** [27] for predicting in-hospital mortality from chest X-ray images, **ADNI** [49] for predicting Alzheimer's disease from brain MRI images, and **AREDS** [14] for predicting late AMD from color fundus images. We selected these datasets for our benchmark based on several key criteria: the availability of temporal information to derive TTE outcomes, the diversity of medical imaging modalities, the presence of various potential sources of bias, the availability of sensitive attributes, and the range of dataset sizes. Table A1 provides basic information about these datasets, with additional details on data access and TTE label construction provided in Appendix C.

**Algorithms.** TTE prediction models: We employ **DeepHit** [43], **Nnet-survival** [15], and **PMF** [40] as the base models for TTE prediction to investigate the fairness issue in medical image prognosis. Fairness algorithms: We incorporate five SOTA fairness algorithms and adapt them to the TTE prediction setting to promote equitable predictions. These algorithms span a wide range of learning strategies and are categorized into three main groups: 1) pre-processing: **subgroup rebalancing (SR)** [31], 2) in-processing: **domain independence (DI)** [71], **fair representation learning (FRL)** [70], **distributionally robust optimization (DRO)** [22], and post-processing: **controlling for sensitive attributes (CSA)** [51]. To the best of our knowledge, this is the first study to comprehensively evaluate such a diverse set of fairness algorithms for TTE prediction in medical imaging. Detailed descriptions of each algorithm are provided in Appendix D.

**Evaluation Metrics.** Compared to classification tasks, evaluating TTE prediction, particularly in medical contexts, requires careful consideration. The primary challenge in assessing accuracy in TTE prediction is **censoring**, which complicates the use of standard evaluation metrics. As a result, no single evaluation metric is universally ideal for all TTE prediction scenarios.

To address these challenges, we adopt multiple evaluation metrics to ensure a comprehensive assessment of TTE predictive performance. These metrics can be grouped into two categories: 1) ranking-based metrics which evaluate the ranking of patients in the dataset based on their corresponding survival times: **time-dependent C-index** ($C^{td}$) [1], **time-dependent AUC** ($AUC^{td}$) [67] and 2) squared error which measures the error between estimated survival times and ground-truth values: **Integrated Brier score** ($IBS$) [16]. For each performance metric, we consider a corresponding fairness metric. A detailed description of all metrics can be found in Appendix E.

**Model Selection.** Prioritizing fairness often involves a trade-off with utility, as the model's objective shifts from utility to balancing both utility and fairness. Consequently, model selection becomes critical in determining the optimal trade-off. In our setting, we perform a hyperparameter search

---

[3]In our context, sources of bias refer to disparities in the data probability distributions across groups, which can be quantified from observed data. However, identifying causal pathways between the sensitive attribute and other variables in the causal graph (Figure 2b) remains challenging and requires deep clinical insight.

and select the best fair TTE prediction models based on fairness performance on the validation set, allowing for up to a 5% reduction in predictive performance compared to the base TTE models.

**Implementation Details.** We use a 2D EfficientNet [61] backbone for the AREDS and MIMIC-CXR datasets, and a 3D ResNet-18 backbone [65] for the ADNI dataset. These lightweight backbones are chosen to mitigate overfitting. In addition to training models from scratch, we also explore pre-training using weights from models trained on the ImageNet [10] and Kinetics [35] datasets. To ensure stability across random initializations, we perform a hyperparameter search with 10 random seeds for each combination of dataset, sensitive attribute, algorithm, and evaluation metric. More implementation details can be found in Appendix E.2 and in our code repository. [4]

## 4 Experiment and Result

**Bias Across TTE Prediction Models in Diverse Imaging Modalities and Outcomes.** We first train TTE prediction models, including DeepHit, Nnet-survival, and PMF, on various datasets and sensitive attributes, then select the best models based on their performance on the validation sets. For each dataset and sensitive attribute pair, we report predictive performance using metrics including $C^{td}$, $AUC^{td}$, and $IBS$, along with fairness as the performance gap between the best and worst groups. As shown in Figure 4, these performance gaps (measured by $C^{td}$) are prevalent across all datasets and sensitive attributes. From this figure, we also observe that these gaps are more pronounced for age and race compared to sex across all datasets. While bias in model predictions has been extensively discussed in the context of medical classification and segmentation, as well as TTE prediction for tabular data, it has not been systematically quantified for TTE prediction in medical imaging. This study provides the first comprehensive analysis across a wide range of imaging modalities, TTE outcomes, and sensitive attributes.

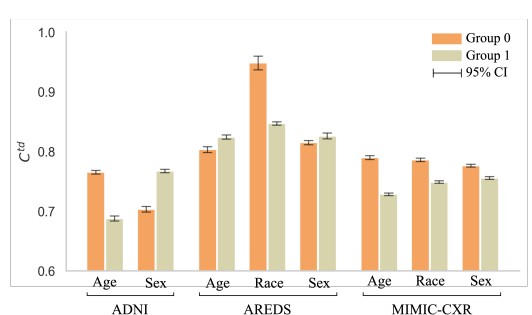

Figure 4: Per-group performance ($C^{td}$) of TTE prediction models across various datasets and sensitive attribute combinations. The visualized performances correspond to the best models determined by model selection conducted on validation sets. The 95% confidence intervals (CIs) are calculated using bootstrapping over test sets. Definitions of groups 0 and 1 are provided in Appendix F.2.

**Statistical Tests.** We conduct statistical tests to verify the robustness of our findings against the variability in TTE prediction models and hyperparameters. Specifically, for each combination of dataset, sensitive attribute, and evaluation metric, we perform two-sided Wilcoxon signed-rank test [6] on the results across all TTE prediction models and hyperparameters to identify significant differences in predictive performance between groups (p-value $< 0.05$). As illustrated in Figure A7, our analysis reveals significant performance disparities between groups across all experimental settings, highlighting the importance of considering fairness in TTE prediction tasks.

**Quantifying Sources of Bias.** To better understand the unfair behavior of TTE prediction models, we quantify the degree of each source of bias in various datasets and sensitive attribute settings. Specifically, to estimate disparities in PMI between RVs across groups, we compute normalized mutual information scores. To evaluate disparities in image features and TTE outcomes across groups, we calculate the Wasserstein distance [68]. Lastly, disparities in censor-

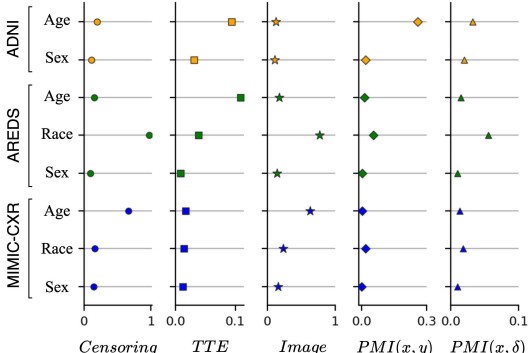

Figure 5: Quantification of the degree of various sources of bias across all datasets and sensitive attributes. Bias degrees range from 0 to 1, where 0 indicates no bias and 1 represents maximum bias within the datasets.

---

[4] https://github.com/pth1993/FairTTE

ing rates are quantified by examining differences in the ratio of censored data across groups. Detailed methods for quantifying sources of bias can be found in Appendix F.4.

Figure 5 illustrates the degree of bias across these five sources for each dataset and sensitive attribute setting. The results reveal a correlation between the degree of bias and the fairness performance of TTE prediction models. For instance, in the AREDS dataset, the degree of bias in terms of disparities in censoring rates, in PMI between $X_Z$ and $Y$, and in PMI between $X_Z$ and $\Delta$ is significantly higher when considering race as the sensitive attribute compared to sex or age. This finding aligns with Figure 4, where the performance gap between racial groups is considerably larger than the gaps between sex or age groups. Additionally, we observe that settings with greater disparities in image features across groups exhibit larger performance gaps. This observation aligns with prior findings in medical image classification tasks, where higher subgroup separability is associated with greater performance disparities [30, 73].

**The Role of Pre-Training.** Pre-training on large datasets has been proven effective in many applications [2]. In this study, we investigate the impact of pre-training on both the accuracy and fairness of TTE prediction models. As shown in Figure 6, pre-training outperforms training models from scratch in terms of predictive performance. This improvement is particularly notable for the ADNI dataset, which is relatively small and contains more complex data structures (3D images) compared to the AREDS and MIMIC-CXR datasets (2D images). However, in terms of fairness, we do not observe a significant improvement with pre-training compared to training from scratch, as shown in Figure A9. Specifically, the p-values from one-sided Wilcoxon signed-rank tests are larger than 0.05 in 18 out of 24 settings, suggesting that pre-training does not lead to more equitable predictions in most cases. These results imply that while pre-training enhances accuracy, combining it with fairness algorithms may be necessary to achieve both fairer and more accurate TTE prediction.

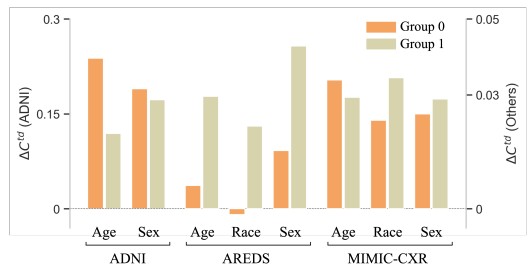

Figure 6: Per-group average performance gap ($\Delta C^{td}$) for TTE prediction models using a pre-training strategy compared to training from scratch across various datasets and sensitive attribute combinations. A positive $\Delta C^{td}$ indicates that the pre-training strategy enhances predictive performance relative to training from scratch.

**Performance of Fairness Algorithms in TTE Prediction.** We conduct an experiment to evaluate the effectiveness of fairness algorithms in reducing disparities between groups in TTE prediction. In this experiment, we use DeepHit as the base TTE prediction model and integrate various fairness algorithms to achieve equitable predictions. To ensure a comprehensive evaluation, we select algorithms from pre-processing (i.e., SR), in-processing (i.e., DI, FRL, DRO), and post-processing approaches (i.e., CSA). As shown in Table 1, existing fairness methods reduce performance gaps between groups compared to DeepHit in settings where bias sources are substantial (e.g., ADNI with age, AREDS with race, and MIMIC-CXR with age). However, when bias sources are small, enforcing fairness can sometimes exacerbate performance disparities. Moreover, no method consistently outperforms DeepHit across all settings. To verify the significance of our observations, we perform a Friedman test followed by a Nemenyi post-hoc test [9] across all algorithms, datasets, and sensitive attribute settings. As shown in Figure A10, the results confirm that current fairness algorithms do not significantly outperform DeepHit in mitigating bias. Additionally, we observe trade-offs in predictive performance in most cases where fairness interventions reduce group disparities: while certain sources of bias (e.g., disparities in image features) are mitigated, others often persist. These findings highlight the need for developing new fairness algorithms capable of addressing multiple sources of bias without compromising predictive performance.

**Fair TTE Prediction Under Distribution Shift.** Fairness algorithms are known to lose effectiveness under distribution shifts, particularly in medical image diagnosis tasks [73, 29]. Building on this insight, we investigate the impact of distribution shifts on fair TTE prediction. As illustrated in Figure 3, we define distribution shifts as scenarios where correlations between the sensitive attribute and other random variables in the causal graph exist in the training dataset but disappear in the testing dataset. To simulate such shifts, we manipulate the data generation process by corrupting images, TTE labels, or censoring indicators for one group while keeping the other group unchanged. The

Table 1: Predictive and fairness performances of fairness algorithms across all dataset and sensitive attribute combinations. We report the actual predictive performance as percentages for DeepHit. For fair algorithms, relative changes in each metric compared to DeepHit are shown. The reported performances correspond to the best models selected via model selection conducted on the validation sets. Blue indicates positive changes, red indicates negative changes. Metrics with ↓ are better when lower, and with ↑ when higher. Best fair algorithm performances are highlighted with gray cells.

| Model | Accuracy | | | Fairness | | | Accuracy | | | Fairness | | |
|---|---|---|---|---|---|---|---|---|---|---|---|---|
| | $AUC^{td}\uparrow$ | $IBS\downarrow$ | $C^{td}\uparrow$ | $\mathcal{F}_{AUC^{td}}\downarrow$ | $\mathcal{F}_{IBS}\downarrow$ | $\mathcal{F}_{C^{td}}\downarrow$ | $AUC^{td}\uparrow$ | $IBS\downarrow$ | $C^{td}\uparrow$ | $\mathcal{F}_{AUC^{td}}\downarrow$ | $\mathcal{F}_{IBS}\downarrow$ | $\mathcal{F}_{C^{td}}\downarrow$ |
| | ADNI - Age | | | | | | ADNI - Sex | | | | | |
| DeepHit | 82.02 | 24.00 | 74.20 | 14.19 | 16.39 | 7.74 | 82.02 | 24.00 | 73.84 | 7.02 | 2.93 | 15.82 |
| DRO | 1.46% | -13.99% | -6.40% | 0.84% | -46.61% | -18.26% | -2.65% | -18.57% | -1.69% | 23.56% | -17.17% | -74.88% |
| SR | -5.76% | 0.90% | 6.73% | -49.81% | -39.84% | -26.50% | -3.54% | 1.99% | -1.67% | 388.82% | -71.74% | -24.78% |
| FRL | -6.00% | -18.63% | -2.55% | 53.91% | -60.63% | -35.86% | -1.75% | -12.58% | -1.98% | -2.95% | 8.74% | -35.60% |
| DI | 1.00% | -14.48% | -2.19% | 55.54% | -37.51% | -66.22% | -15.81% | 5.10% | -5.57% | 14.09% | 193.83% | 7.49% |
| CSA | -4.18% | -11.44% | -5.79% | -32.44% | -42.55% | -26.01% | -3.89% | -4.99% | -10.38% | 59.28% | -0.51% | -50.43% |
| | AREDS - Age | | | | | | AREDS - Race | | | | | |
| DeepHit | 78.41 | 15.37 | 81.30 | 1.58 | 12.56 | 2.20 | 81.78 | 11.74 | 84.53 | 14.00 | 10.14 | 11.09 |
| DRO | 0.88% | -4.37% | 0.08% | 19.19% | -20.08% | -5.03% | 1.78% | -2.28% | -1.41% | -34.89% | -10.93% | -37.21% |
| SR | 1.71% | -2.52% | 0.32% | -1.93% | -4.59% | 15.44% | -0.58% | -3.73% | -0.69% | -24.63% | -11.97% | -5.65% |
| FRL | 0.80% | -0.70% | 0.20% | -25.41% | -8.21% | -35.01% | 1.75% | 4.41% | -0.21% | -33.49% | -16.10% | -11.51% |
| DI | 2.12% | -1.74% | 0.73% | 83.73% | -6.53% | 9.89% | 2.47% | -0.67% | -0.84% | -8.58% | -9.64% | -26.11% |
| CSA | 0.89% | -5.94% | 0.11% | -1.95% | -29.56% | 4.60% | 1.37% | -1.32% | -0.23% | -10.74% | -8.14% | -21.97% |
| | AREDS - Sex | | | | | | MIMIC-CXR - Age | | | | | |
| DeepHit | 79.08 | 15.36 | 81.77 | 0.76 | 3.84 | 1.32 | 78.61 | 20.38 | 76.21 | 3.06 | 1.49 | 5.93 |
| DRO | 1.21% | -3.62% | -0.54% | 12.75% | -6.52% | -95.74% | -1.55% | 13.50% | -1.61% | -49.46% | 58.14% | -17.81% |
| SR | -0.07% | -4.02% | -0.14% | -47.46% | -4.14% | -85.27% | -0.96% | 2.75% | -0.37% | -82.53% | -8.31% | 2.86% |
| FRL | 1.76% | -5.37% | -0.54% | -51.07% | -1.20% | 23.24% | -4.28% | 5.26% | -2.42% | -61.41% | -60.79% | -21.96% |
| DI | 0.18% | -5.84% | 1.31% | -13.82% | 16.12% | -65.70% | 0.41% | 5.29% | -0.63% | -52.21% | -44.27% | -10.24% |
| CSA | -0.18% | -8.23% | 0.46% | -28.46% | 17.58% | -71.42% | 0.36% | -3.19% | -0.97% | -66.46% | -29.21% | -13.72% |
| | MIMIC-CXR - Race | | | | | | MIMIC-CXR - Sex | | | | | |
| DeepHit | 78.61 | 21.10 | 76.21 | 2.89 | 0.87 | 4.03 | 78.61 | 20.38 | 76.21 | 4.77 | 1.23 | 2.09 |
| DRO | -0.72% | 3.13% | -0.78% | -5.69% | 31.34% | -4.80% | -0.35% | 4.51% | -1.20% | 12.28% | 92.27% | -9.61% |
| SR | -0.11% | -0.90% | -0.41% | 10.28% | -74.37% | -0.80% | -0.12% | 2.31% | -0.19% | 17.02% | 25.35% | -1.37% |
| FRL | -0.61% | -2.44% | 0.01% | 36.89% | -19.46% | 0.34% | -2.24% | 0.21% | -0.57% | 15.69% | 10.97% | -4.16% |
| DI | -0.44% | 0.00% | -0.52% | 37.80% | 30.77% | -0.43% | -1.94% | 1.21% | -0.34% | 14.26% | -8.22% | -13.76% |
| CSA | -0.42% | -4.61% | -0.96% | -14.83% | 36.14% | 0.44% | -0.47% | -2.58% | -0.63% | 10.63% | -49.08% | -0.46% |

real-world motivations and detailed descriptions of these causal distribution shifts are provided in Appendices G.2 and F.5 .

Figure 7 compares the predictive performance of models under in-distribution and distribution shift learning scenarios for the MIMIC-CXR dataset, where sex is considered the sensitive attribute. The complete results for all settings are provided in Appendix H.6. Our findings indicate that different types of distribution shifts impact model performance in distinct ways. Specifically, adding noise to TTE labels (shift in $Y$) leads to a significant decline in $IBS$, as the increased label uncertainty makes accurate predictions more challenging. Flipping censoring indicators (shift in $\Delta$), on the other hand, severely degrades ranking-based metrics such as $C^{td}$ and $AUC^{td}$, as it reduces the availability of comparable pairs during training, making it harder for the model to learn an effective ranking function. Additionally, introducing noise to medical images (shift in $X$) negatively affects all performance metrics, as degraded image quality limits the model's ability to extract meaningful features. These observations align with our expectations—corrupting TTE labels directly impacts metrics measuring error between predicted and ground-truth TTE, while censoring flips disrupt ranking-based evaluations by impairing the model's ability to differentiate survival times.

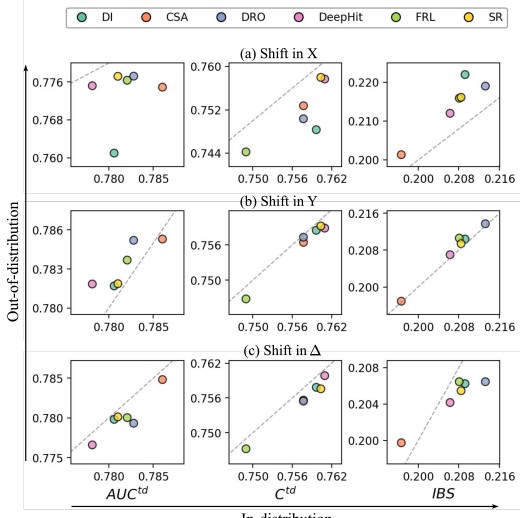

Figure 7: Predictive performance comparison of fair TTE prediction models in ID vs. OOD settings on the MIMIC-CXR dataset with sex as the sensitive attribute. Results are averaged across all random seeds. Points on the dashed line indicate equal performance in both settings. For $AUC^{td}$ and $C^{td}$ (resp. $IBS$), points below (resp. above) the line indicate degraded performance in OOD.

Regarding fairness, our observations reveal mixed results. In some cases, distribution shifts negatively impact fairness, while in others, they lead to improvements. Notably, adding noise to the data of one group generally degrades predictive performance on that group. However, we also observe that this noise can similarly affect the other group—sometimes even more severely—thereby reducing the performance gap between groups and unintentionally improving fairness. These findings underscore the complexities of achieving fairness in the presence of distribution shifts and highlight the need for developing TTE algorithms that are both fair and robust in such challenging scenarios.

## 5 Limitation and Broader Impact

While FairTTE represents a significant step forward in advancing fairness research in medical image prognosis by providing the first comprehensive benchmark across multiple datasets and fairness algorithms, we acknowledge several limitations in the current study. Specifically, our analysis focuses on the standard TTE prediction setting, which assumes a single clinical risk and non-informative right censoring. In real-world clinical applications, however, patients often face multiple competing risks and may drop out of studies for reasons that introduce informative censoring, making the fairness landscape considerably more complex. Addressing fairness in such settings remains an important direction for future research. Additionally, although we adopt group fairness definitions (i.e., minimizing performance gaps across subgroups), we recognize that strictly enforcing these criteria can, in some clinical scenarios, degrade overall utility or harm performance for all groups. In such scenarios, alternative fairness notions may be more appropriate, depending on the clinical context and ethical objectives.

Selecting an appropriate fairness criterion in the medical domain requires careful consideration of the clinical context, ethical principles, and statistical validity. An effective fairness notion should align with the model's intended use, its potential impact on different patient subgroups, and real-world constraints. In our study, we focus on statistical group fairness (i.e., predictive performance gaps across subgroups) which is widely examined in the medical image analysis literature. This choice is grounded in the assumption that any causal pathway from sensitive attributes represents an unfair influence in the causal graph and should be mitigated through fairness constraints. However, we acknowledge that in practical clinical scenarios, such causal pathways may reflect fair and clinically meaningful relationships. Enforcing group fairness in these cases may inadvertently remove relevant information and degrade predictive performance.

Moving forward, we suggest several directions for fairness research in medical image analysis: (1) Identifying the causal nature of bias to distinguish between fair and unfair sources, enabling models to address specific pathways appropriately; (2) Developing fairness metrics and mitigation strategies that preserve clinically relevant (fair) pathways while minimizing the effect of unfair ones; and (3) Collaborating with clinicians and domain experts to define context-specific fairness objectives that are aligned with both clinical utility and ethical standards.

## 6 Conclusion

We systematically investigate fair TTE prediction in medical imaging, introducing a unified framework to define and quantify bias sources and establishing a comprehensive and realistic evaluation. Our framework includes three TTE prediction methods, five fairness algorithms, and three large-scale public datasets spanning diverse modalities and outcomes. Extensive experiments reveal critical insights, including the link between bias sources and model unfairness, the inconsistent effectiveness of existing fairness methods, the role of pre-training strategies, and the challenges of causal distribution shifts. We hope our fine-grained analysis encourages more rigorous evaluations and drives the development of new fairness algorithms for TTE prediction.

## Acknowledgements

This work was funded in part by the National Science Foundation under award number IIS-2145625 and by the National Institutes of Health under awards number R01AI188576 and R01CA301579.

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

## NeurIPS Paper Checklist

1. **Claims**

   Question: Do the main claims made in the abstract and introduction accurately reflect the paper's contributions and scope?

   Answer: [Yes]

   Justification: The main claims presented in the abstract and introduction accurately reflect the paper's contributions and scope. They are supported by both theoretical results—formally stated and proven—and comprehensive experimental evaluations across multiple datasets, metrics, and fairness algorithms. We clearly distinguish between established findings and broader motivations, and we explicitly state assumptions and limitations to ensure that the claims are appropriately contextualized and not overstated.

   Guidelines:

   - The answer NA means that the abstract and introduction do not include the claims made in the paper.
   - The abstract and/or introduction should clearly state the claims made, including the contributions made in the paper and important assumptions and limitations. A No or NA answer to this question will not be perceived well by the reviewers.
   - The claims made should match theoretical and experimental results, and reflect how much the results can be expected to generalize to other settings.
   - It is fine to include aspirational goals as motivation as long as it is clear that these goals are not attained by the paper.

2. **Limitations**

   Question: Does the paper discuss the limitations of the work performed by the authors?

   Answer: [Yes]

   Justification: We explicitly discuss the assumptions made in our theoretical framework—such as triangle inequality and symmetric properties for evaluation metric—and their potential violations in real-world scenarios. We also acknowledge limitations in generalizability, as our experiments are conducted on a finite set of publicly available medical imaging datasets. These limitations help contextualize our contributions and inform directions for future research.

   Guidelines:

   - The answer NA means that the paper has no limitation while the answer No means that the paper has limitations, but those are not discussed in the paper.
   - The authors are encouraged to create a separate "Limitations" section in their paper.
   - The paper should point out any strong assumptions and how robust the results are to violations of these assumptions (e.g., independence assumptions, noiseless settings, model well-specification, asymptotic approximations only holding locally). The authors should reflect on how these assumptions might be violated in practice and what the implications would be.
   - The authors should reflect on the scope of the claims made, e.g., if the approach was only tested on a few datasets or with a few runs. In general, empirical results often depend on implicit assumptions, which should be articulated.
   - The authors should reflect on the factors that influence the performance of the approach. For example, a facial recognition algorithm may perform poorly when image resolution is low or images are taken in low lighting. Or a speech-to-text system might not be used reliably to provide closed captions for online lectures because it fails to handle technical jargon.
   - The authors should discuss the computational efficiency of the proposed algorithms and how they scale with dataset size.

- If applicable, the authors should discuss possible limitations of their approach to address problems of privacy and fairness.
- While the authors might fear that complete honesty about limitations might be used by reviewers as grounds for rejection, a worse outcome might be that reviewers discover limitations that aren't acknowledged in the paper. The authors should use their best judgment and recognize that individual actions in favor of transparency play an important role in developing norms that preserve the integrity of the community. Reviewers will be specifically instructed to not penalize honesty concerning limitations.

3. **Theory assumptions and proofs**

Question: For each theoretical result, does the paper provide the full set of assumptions and a complete (and correct) proof?

Answer: [Yes]

Justification: The assumptions for Theorem 1 and Proposition 2 are explicitly stated in Section 2.2, and complete, formal proofs are provided in Appendix B. All theoretical results are clearly numbered, cross-referenced, and presented with the necessary assumptions to ensure clarity and correctness. Where appropriate, we also include intuitive explanations in the main text to aid understanding.

Guidelines:

- The answer NA means that the paper does not include theoretical results.
- All the theorems, formulas, and proofs in the paper should be numbered and cross-referenced.
- All assumptions should be clearly stated or referenced in the statement of any theorems.
- The proofs can either appear in the main paper or the supplemental material, but if they appear in the supplemental material, the authors are encouraged to provide a short proof sketch to provide intuition.
- Inversely, any informal proof provided in the core of the paper should be complemented by formal proofs provided in appendix or supplemental material.
- Theorems and Lemmas that the proof relies upon should be properly referenced.

4. **Experimental result reproducibility**

Question: Does the paper fully disclose all the information needed to reproduce the main experimental results of the paper to the extent that it affects the main claims and/or conclusions of the paper (regardless of whether the code and data are provided or not)?

Answer: [Yes]

Justification: The experimental design and implementation details necessary to reproduce the main results are thoroughly documented in Section 3 and Appendix F. This includes dataset descriptions, preprocessing steps, model architectures, training protocols, hyperparameters, and evaluation metrics. We also describe our experimental setup across all dataset and sensitive attribute combinations. These details are sufficient for reproducing our findings, even without access to the code or data at submission time. Code and documentation will be released publicly to further facilitate reproducibility upon publication.

Guidelines:

- The answer NA means that the paper does not include experiments.
- If the paper includes experiments, a No answer to this question will not be perceived well by the reviewers: Making the paper reproducible is important, regardless of whether the code and data are provided or not.
- If the contribution is a dataset and/or model, the authors should describe the steps taken to make their results reproducible or verifiable.
- Depending on the contribution, reproducibility can be accomplished in various ways. For example, if the contribution is a novel architecture, describing the architecture fully might suffice, or if the contribution is a specific model and empirical evaluation, it may be necessary to either make it possible for others to replicate the model with the same dataset, or provide access to the model. In general. releasing code and data is often one good way to accomplish this, but reproducibility can also be provided via detailed

instructions for how to replicate the results, access to a hosted model (e.g., in the case of a large language model), releasing of a model checkpoint, or other means that are appropriate to the research performed.

- While NeurIPS does not require releasing code, the conference does require all submissions to provide some reasonable avenue for reproducibility, which may depend on the nature of the contribution. For example

    (a) If the contribution is primarily a new algorithm, the paper should make it clear how to reproduce that algorithm.

    (b) If the contribution is primarily a new model architecture, the paper should describe the architecture clearly and fully.

    (c) If the contribution is a new model (e.g., a large language model), then there should either be a way to access this model for reproducing the results or a way to reproduce the model (e.g., with an open-source dataset or instructions for how to construct the dataset).

    (d) We recognize that reproducibility may be tricky in some cases, in which case authors are welcome to describe the particular way they provide for reproducibility. In the case of closed-source models, it may be that access to the model is limited in some way (e.g., to registered users), but it should be possible for other researchers to have some path to reproducing or verifying the results.

5. **Open access to data and code**

    Question: Does the paper provide open access to the data and code, with sufficient instructions to faithfully reproduce the main experimental results, as described in supplemental material?

    Answer: [Yes]

    Justification: We provide open access to all code, documentation, and scripts necessary to reproduce our main experimental results. Detailed instructions for setting up the environment, accessing and preprocessing the datasets (including MIMIC-CXR, ADNI, and AREDS), and running the experiments are included in Appendix F and will be made publicly available upon publication. For datasets with access restrictions, we provide clear guidance on how to obtain access through the appropriate data use agreements in Appendix C. Our released code includes scripts for training, evaluation, and statistical analysis, along with configuration files for all reported experiments to ensure faithful reproduction of our results.

    Guidelines:

    - The answer NA means that paper does not include experiments requiring code.
    - Please see the NeurIPS code and data submission guidelines (`https://nips.cc/public/guides/CodeSubmissionPolicy`) for more details.
    - While we encourage the release of code and data, we understand that this might not be possible, so "No" is an acceptable answer. Papers cannot be rejected simply for not including code, unless this is central to the contribution (e.g., for a new open-source benchmark).
    - The instructions should contain the exact command and environment needed to run to reproduce the results. See the NeurIPS code and data submission guidelines (`https://nips.cc/public/guides/CodeSubmissionPolicy`) for more details.
    - The authors should provide instructions on data access and preparation, including how to access the raw data, preprocessed data, intermediate data, and generated data, etc.
    - The authors should provide scripts to reproduce all experimental results for the new proposed method and baselines. If only a subset of experiments are reproducible, they should state which ones are omitted from the script and why.
    - At submission time, to preserve anonymity, the authors should release anonymized versions (if applicable).
    - Providing as much information as possible in supplemental material (appended to the paper) is recommended, but including URLs to data and code is permitted.

6. **Experimental setting/details**

    Question: Does the paper specify all the training and test details (e.g., data splits, hyperparameters, how they were chosen, type of optimizer, etc.) necessary to understand the results?

Answer: [Yes]

Justification: The experimental settings, including data splits, hyperparameter configurations, optimization strategies, and model selection procedures, are comprehensively detailed in Appendix F. These include descriptions of how hyperparameters were tuned, the choice of optimizer, training schedules, and evaluation protocols. This information is sufficient to understand and contextualize the reported results.

Guidelines:

- The answer NA means that the paper does not include experiments.
- The experimental setting should be presented in the core of the paper to a level of detail that is necessary to appreciate the results and make sense of them.
- The full details can be provided either with the code, in appendix, or as supplemental material.

7. **Experiment statistical significance**

Question: Does the paper report error bars suitably and correctly defined or other appropriate information about the statistical significance of the experiments?

Answer: [Yes]

Justification: We report error bars using bootstrapping across multiple runs to capture variability due to model initialization, train/validation splits, and random seeds. Specifically, we calculate 95% confidence intervals based on bootstrapped samples to quantify uncertainty in model performance. Additionally, we assess statistical significance using well-established non-parametric tests, including the Wilcoxon signed-rank test for pairwise comparisons, and the Friedman test followed by the Nemenyi post-hoc test for comparing multiple methods across datasets and sensitive attributes.

Guidelines:

- The answer NA means that the paper does not include experiments.
- The authors should answer "Yes" if the results are accompanied by error bars, confidence intervals, or statistical significance tests, at least for the experiments that support the main claims of the paper.
- The factors of variability that the error bars are capturing should be clearly stated (for example, train/test split, initialization, random drawing of some parameter, or overall run with given experimental conditions).
- The method for calculating the error bars should be explained (closed form formula, call to a library function, bootstrap, etc.)
- The assumptions made should be given (e.g., Normally distributed errors).
- It should be clear whether the error bar is the standard deviation or the standard error of the mean.
- It is OK to report 1-sigma error bars, but one should state it. The authors should preferably report a 2-sigma error bar than state that they have a 96% CI, if the hypothesis of Normality of errors is not verified.
- For asymmetric distributions, the authors should be careful not to show in tables or figures symmetric error bars that would yield results that are out of range (e.g. negative error rates).
- If error bars are reported in tables or plots, The authors should explain in the text how they were calculated and reference the corresponding figures or tables in the text.

8. **Experiments compute resources**

Question: For each experiment, does the paper provide sufficient information on the computer resources (type of compute workers, memory, time of execution) needed to reproduce the experiments?

Answer: [Yes]

Justification: Details on compute resources used for all experiments are provided in Appendix F.1. This includes the type of hardware, memory specifications, and estimated runtime for each experimental run. This information is sufficient to support reproducibility and assess the computational requirements of our work.

Guidelines:

- The answer NA means that the paper does not include experiments.
- The paper should indicate the type of compute workers CPU or GPU, internal cluster, or cloud provider, including relevant memory and storage.
- The paper should provide the amount of compute required for each of the individual experimental runs as well as estimate the total compute.
- The paper should disclose whether the full research project required more compute than the experiments reported in the paper (e.g., preliminary or failed experiments that didn't make it into the paper).

9. **Code of ethics**

Question: Does the research conducted in the paper conform, in every respect, with the NeurIPS Code of Ethics `https://neurips.cc/public/EthicsGuidelines`?

Answer: [Yes]

Justification: We have thoroughly reviewed the NeurIPS Code of Ethics and confirm that our research adheres to all applicable guidelines. Our study involves no human subjects, privacy risks, or potentially harmful applications, and we have taken appropriate steps to ensure transparency, reproducibility, and responsible use of data and models.

Guidelines:

- The answer NA means that the authors have not reviewed the NeurIPS Code of Ethics.
- If the authors answer No, they should explain the special circumstances that require a deviation from the Code of Ethics.
- The authors should make sure to preserve anonymity (e.g., if there is a special consideration due to laws or regulations in their jurisdiction).

10. **Broader impacts**

Question: Does the paper discuss both potential positive societal impacts and negative societal impacts of the work performed?

Answer: [NA]

Justification: This paper presents foundational research on the methodological evaluation of fairness in TTE prediction using medical imaging. The work is primarily intended to inform academic research and improve scientific understanding of bias in prognostic modeling. It does not introduce new systems intended for deployment, nor does it involve the creation or use of data or models with direct societal impact. As such, we consider the broader societal implications beyond the scope of this work and have not included a dedicated discussion.

Guidelines:

- The answer NA means that there is no societal impact of the work performed.
- If the authors answer NA or No, they should explain why their work has no societal impact or why the paper does not address societal impact.
- Examples of negative societal impacts include potential malicious or unintended uses (e.g., disinformation, generating fake profiles, surveillance), fairness considerations (e.g., deployment of technologies that could make decisions that unfairly impact specific groups), privacy considerations, and security considerations.
- The conference expects that many papers will be foundational research and not tied to particular applications, let alone deployments. However, if there is a direct path to any negative applications, the authors should point it out. For example, it is legitimate to point out that an improvement in the quality of generative models could be used to generate deepfakes for disinformation. On the other hand, it is not needed to point out that a generic algorithm for optimizing neural networks could enable people to train models that generate Deepfakes faster.
- The authors should consider possible harms that could arise when the technology is being used as intended and functioning correctly, harms that could arise when the technology is being used as intended but gives incorrect results, and harms following from (intentional or unintentional) misuse of the technology.

- If there are negative societal impacts, the authors could also discuss possible mitigation strategies (e.g., gated release of models, providing defenses in addition to attacks, mechanisms for monitoring misuse, mechanisms to monitor how a system learns from feedback over time, improving the efficiency and accessibility of ML).

11. **Safeguards**

Question: Does the paper describe safeguards that have been put in place for responsible release of data or models that have a high risk for misuse (e.g., pretrained language models, image generators, or scraped datasets)?

Answer: [NA]

Justification: The paper poses no such risks.

Guidelines:

- The answer NA means that the paper poses no such risks.
- Released models that have a high risk for misuse or dual-use should be released with necessary safeguards to allow for controlled use of the model, for example by requiring that users adhere to usage guidelines or restrictions to access the model or implementing safety filters.
- Datasets that have been scraped from the Internet could pose safety risks. The authors should describe how they avoided releasing unsafe images.
- We recognize that providing effective safeguards is challenging, and many papers do not require this, but we encourage authors to take this into account and make a best faith effort.

12. **Licenses for existing assets**

Question: Are the creators or original owners of assets (e.g., code, data, models), used in the paper, properly credited and are the license and terms of use explicitly mentioned and properly respected?

Answer: [Yes]

Justification: All external assets used in this paper—including datasets (MIMIC-CXR, ADNI, and AREDS) and pretrained models (e.g., RETFound and EfficientNet)—are properly cited with their corresponding publications. We use publicly available versions of these datasets and models in accordance with their respective licenses and terms of use. We include dataset description and access URLs in the Appendix C.

Guidelines:

- The answer NA means that the paper does not use existing assets.
- The authors should cite the original paper that produced the code package or dataset.
- The authors should state which version of the asset is used and, if possible, include a URL.
- The name of the license (e.g., CC-BY 4.0) should be included for each asset.
- For scraped data from a particular source (e.g., website), the copyright and terms of service of that source should be provided.
- If assets are released, the license, copyright information, and terms of use in the package should be provided. For popular datasets, `paperswithcode.com/datasets` has curated licenses for some datasets. Their licensing guide can help determine the license of a dataset.
- For existing datasets that are re-packaged, both the original license and the license of the derived asset (if it has changed) should be provided.
- If this information is not available online, the authors are encouraged to reach out to the asset's creators.

13. **New assets**

Question: Are new assets introduced in the paper well documented and is the documentation provided alongside the assets?

Answer: [Yes]

Justification: We release new assets as part of this work, including benchmark code, evaluation scripts, and structured documentation for fair TTE prediction across multiple medical imaging datasets. These assets are well-documented and include detailed instructions for dataset preprocessing, model training, evaluation, and reproduction of results. All assets are anonymized for the review process and will be made publicly available upon publication. No personally identifiable information or private data is included, and all released components comply with the data use agreements of the underlying datasets.

Guidelines:

- The answer NA means that the paper does not release new assets.
- Researchers should communicate the details of the dataset/code/model as part of their submissions via structured templates. This includes details about training, license, limitations, etc.
- The paper should discuss whether and how consent was obtained from people whose asset is used.
- At submission time, remember to anonymize your assets (if applicable). You can either create an anonymized URL or include an anonymized zip file.

14. **Crowdsourcing and research with human subjects**

Question: For crowdsourcing experiments and research with human subjects, does the paper include the full text of instructions given to participants and screenshots, if applicable, as well as details about compensation (if any)?

Answer: [NA]

Justification: The paper does not involve crowdsourcing nor research with human subjects.

Guidelines:

- The answer NA means that the paper does not involve crowdsourcing nor research with human subjects.
- Including this information in the supplemental material is fine, but if the main contribution of the paper involves human subjects, then as much detail as possible should be included in the main paper.
- According to the NeurIPS Code of Ethics, workers involved in data collection, curation, or other labor should be paid at least the minimum wage in the country of the data collector.

15. **Institutional review board (IRB) approvals or equivalent for research with human subjects**

Question: Does the paper describe potential risks incurred by study participants, whether such risks were disclosed to the subjects, and whether Institutional Review Board (IRB) approvals (or an equivalent approval/review based on the requirements of your country or institution) were obtained?

Answer: [NA]

Justification: This study does not involve crowdsourcing or research with human subjects; therefore, IRB approval is not required.

Guidelines:

- The answer NA means that the paper does not involve crowdsourcing nor research with human subjects.
- Depending on the country in which research is conducted, IRB approval (or equivalent) may be required for any human subjects research. If you obtained IRB approval, you should clearly state this in the paper.
- We recognize that the procedures for this may vary significantly between institutions and locations, and we expect authors to adhere to the NeurIPS Code of Ethics and the guidelines for their institution.
- For initial submissions, do not include any information that would break anonymity (if applicable), such as the institution conducting the review.

16. **Declaration of LLM usage**

Question: Does the paper describe the usage of LLMs if it is an important, original, or non-standard component of the core methods in this research? Note that if the LLM is used only for writing, editing, or formatting purposes and does not impact the core methodology, scientific rigorousness, or originality of the research, declaration is not required.

Answer: [NA]

Justification: The core method development in this research does not involve LLMs as any important, original, or non-standard component; therefore, a declaration of LLM usage is not applicable.

Guidelines:

- The answer NA means that the core method development in this research does not involve LLMs as any important, original, or non-standard components.
- Please refer to our LLM policy (`https://neurips.cc/Conferences/2025/LLM`) for what should or should not be described.

## Appendix Contents

# A Related Works

**TTE Prediction.** TTE prediction models can generally be classified into two categories: continuous-time and discrete-time models, each with distinct approaches for handling event timing. Continuous-time models treat time as a continuous variable and often extend traditional models like Cox regression. For example, DeepSurv [34] extends the Cox regression by using a deep neural network with non-linear activation functions in hidden layers. Cox-Time [41] further builds on DeepSurv, introducing time-dependent predictors that allow for the estimation of time-varying effects. In contrast, discrete-time models treat time as a series of distinct intervals and typically use classification techniques. DeepHit [43] learns survival times directly without assuming a specific underlying stochastic process, parameterizing the discrete probability mass function. Another method, Nnet-survival [15], parametrizes the discrete hazard function using a neural network and optimizes the negative log-likelihood loss.

**Fairness in Machine Learning.** Fairness in machine learning has gained significant attention in recent years, with a focus on ensuring models are unbiased and equitable across individuals and groups. *Fairness metrics.* Fairness metrics can be broadly categorized into two types: group fairness [13, 7, 74, 18] and individual fairness [13, 59]. Group fairness ensures that models are fair across different demographic groups, while individual fairness emphasizes that similar individuals should be treated similarly. *Fairness algorithms.* To address fairness and bias issues, bias mitigation methods are generally classified into three approaches: pre-processing, which focuses on modifying the input data before model training [31, 4]; in-processing, which incorporates fairness constraints during model training [32, 75, 45, 60]; and post-processing, which adjusts model outputs to improve fairness [18, 24].

**Fairness in Medical Imaging.** In medical image analysis, machine learning (ML) models have been shown to exhibit systematic biases related to various attributes such as race, gender, and age [57, 42]. These biases are prevalent across different medical imaging modalities, including chest X-rays [56], CT scans [81], and skin dermatology images [38]. While several efforts have been made to benchmark fairness algorithms on medical images, existing datasets [23, 39, 66, 17] and benchmarks [82, 64] primarily focus on diagnostic tasks like image classification and segmentation. Unfortunately, they often overlook the crucial domain of medical prognosis, which involves predicting TTE outcomes.

**Fairness in TTE Prediction.** Despite significant advances in TTE prediction, research on fairness in this area remains limited. *Fairness metrics for TTE prediction.* Fairness metrics for TTE prediction have only recently been defined. These metrics can be roughly classified into three categories based on their objectives: (i) ensuring similar predicted TTE outcomes for similar data points [36, 53, 77, 76, 72], (ii) ensuring similar predicted outcomes for data points from different groups [36, 53, 80], and (iii) ensuring similar predictive performance across different groups [11, 22, 78, 79]. *Fairness algorithms for TTE prediction.* Building on these metrics, several methods have been proposed to achieve fair TTE prediction. One approach incorporates fairness as a regularization term during model training [36, 53, 11], ensuring that the model accounts for fairness constraints throughout its optimization process. Another approach focuses on improving worst-group accuracy by leveraging distributionally robust optimization techniques [22, 20, 55], which aim to enhance performance for underrepresented or disadvantaged groups. In addition to these in-processing methods, recent work has also explored pre- and post-processing strategies to address fairness in TTE prediction [80]. However, these efforts are limited to tabular data and fail to consider medical images, which are essential and pervasive in medical prognosis tasks.

## B   Missing Proof

**Proof for Theorem 1**   For any $a, a' \in \mathcal{A}$, we have:

$$\mathrm{Er}(f_{a'}, h, D_{a'}) \leq \mathrm{Er}(f_a, h, D_a) + |\mathrm{Er}(f_a, h, D_a) - \mathrm{Er}(f_{a'}, h, D_{a'})|$$

$$\overset{(1)}{\leq} \mathrm{Er}(f_a, h, D_a) + |\mathrm{Er}(f_a, h, D_a) - \mathrm{Er}(h, h^*, D_a)|$$
$$+ |\mathrm{Er}(h, h^*, D_a) - \mathrm{Er}(h, h^*, D_{a'})| + |\mathrm{Er}(h, h^*, D_{a'}) - \mathrm{Er}(f_{a'}, h, D_{a'})|$$

$$\overset{(2)}{\leq} \mathrm{Er}(f_a, h, D_a) + \mathrm{Er}(f_a, h^*, D_a) + \mathcal{D}(\mathcal{H}, D_a, D_{a'}) + \mathrm{Er}(f_{a'}, h^*, D_{a'})$$

$$\overset{(3)}{=} \mathrm{Er}(f_a, h, D_a) + \eta(\mathcal{H}, f_a, f_{a'}) + \mathcal{D}(\mathcal{H}, D_a, D_{a'})$$

where $h^* = \arg\min_{h' \in \mathcal{H}} (\mathrm{Er}(f_a, h', D_a) + \mathrm{Er}(f_{a'}, h', D_{a'}))$. We have $\overset{(1)}{\leq}$ by using inequality $|a + b| \leq |a| + |b|$; $\overset{(2)}{\leq}$ by using triangle inequality for Er metric and $|\mathrm{Er}(h, h^*, D_a) - \mathrm{Er}(h, h^*, D_{a'})| \leq \max_{h', h'' \in \mathcal{H}} |\mathrm{Er}(h', h'', D_a) - \mathrm{Er}(h', h'', D_{a'})| = \mathcal{D}(\mathcal{H}, D_a, D_{a'})$; $\overset{(3)}{=}$ because $\eta(\mathcal{H}, f_a, f_{a'}) = (\mathrm{Er}(f_a, h^*, D_a) + \mathrm{Er}(f_{a'}, h^*, D_{a'}))$ by definition. Subtracting $\mathrm{Er}(f_a, h, D_a)$ from both sides and taking $\max$ operator, we have:

$$\mathcal{F}_{\mathrm{Er}}(h) = \max_{a, a' \in \mathcal{A}} |\mathrm{Er}(f_a, h, D_a) - \mathrm{Er}(f_{a'}, h, D_{a'})| \leq \max_{a, a' \in \mathcal{A}} (\eta(\mathcal{H}, f_a, f_{a'}) + \mathcal{D}(\mathcal{H}, D_a, D_{a'}))$$

**Discussion on the assumption of performance metric.**   The proof of Theorem 1 relies on the assumption that the performance metric Er satisfies the properties of triangle inequality and symmetry. This assumption is relatively mild and holds for many commonly used performance metrics. For instance, [58] introduced the symmetric discordance index (SDI), a ranking-based metric that adheres to these properties, demonstrating the practical applicability of this assumption in practice.

**Proof of Proposition 2**   For $a \in \mathcal{A}$ with $I_a(Z, T) = I_a(X, T)$, we have:

$$\log P(t|x, a) = \log \left( \int P(t, z|x, a) dz \right)$$

$$= \log \left( \int P(t|z, a) P(z|x, a) dz \right)$$

$$= \log \left( \mathbb{E}_{P(z|x)} [P(t|z, a)] \right)$$

$$\overset{(1)}{\geq} \mathbb{E}_{P(z|x)} [\log p(t|z, a)] \tag{5}$$

We have $\overset{(1)}{\geq}$ by using Jensen's inequality. $\forall a' \in \mathcal{A}$, taking expectation w.r.t. $P(x, t|a')$ over both sides, we have:

$$\mathbb{E}_{P(x, t|a')} \left[ \log P(t|x, a) - \mathbb{E}_{P(z|x)} [\log P(t|z, a)] \right]$$

$$= \int \int \left( \log P(t|x, a) - \mathbb{E}_{P(z|x)} [\log P(t|z, a)] \right) P(x, t|a') dx dt$$

$$= \int \int \left( \log P(t|x, a) - \mathbb{E}_{P(z|x)} [\log P(t|z, a)] \right) P(x, t|a) \frac{P(x, t|a')}{P(x, t|a)} dx dt$$

$$= \mathbb{E}_{P(x, t|a)} \left[ \left( \log P(t|x, a) - \mathbb{E}_{P(z|x)} [\log P(t|z, a)] \right) \frac{P(x, t|a')}{P(x, t|a)} \right]$$

$$\overset{(1)}{\leq} \left( \max_{x,t} \frac{P(x,t|a')}{P(x,t|a)} \right) \mathbb{E}_{P(x,t|a)} \left[ \log P(t|x,a) - \mathbb{E}_{P(z|x)} \left[ \log P(t|z,a) \right] \right]$$

$$= \left( \max_{x,t} \frac{P(x,t|a')}{P(x,t|a)} \right) \left( \mathbb{E}_{P(x,t|a)} \left[ \log P(t|x,a) \right] - \mathbb{E}_{P(z,t|a)} \left[ \log P(t|z,a) \right] \right)$$

$$= \left( \max_{x,t} \frac{P(x,t|a')}{P(x,t|a)} \right) (H_a(T,X) - H_{a'}(T,Z))$$

$$= \left( \max_{x,t} \frac{P(x,t|a')}{P(x,t|a)} \right) ((H_a(T) - H_a(T,Z)) - (H_a(T) - H_a(T,X)))$$

$$= \left( \max_{x,t} \frac{P(x,t|a')}{P(x,t|a)} \right) (I_a(T,Z) - I_a(T,X))$$

$$\overset{(2)}{=} 0 \tag{6}$$

We have $\overset{(1)}{\leq}$ because $\log P(t|x,a) - \mathbb{E}_{P(z|x)} \left[ \log P(t|z,a) \right] \geq 0$ according to Eq. (5); $\overset{(2)}{=}$ because $I_a(T,Z) = I_a(T,X)$. Based on Eq. (6), we have:

$$\mathbb{E}_{P(x,t|a')} \left[ \log P(t|x,a) \right] = \mathbb{E}_{P(x,t|a')} \left[ \mathbb{E}_{P(z|x)} \left[ \log P(t|z,a) \right] \right]$$
$$= \mathbb{E}_{P(t,z|a')} \left[ \log P(t|z,a) \right] \tag{7}$$

We also have:

$$\mathbb{E}_{P(t,z|a')} \left[ \log P(t|z,a') \right] = -H_{a'}(T|Z)$$
$$= I_{a'}(T,Z) - H_{a'}(T)$$
$$\overset{(1)}{\leq} I_{a'}(T,X) - H_{a'}(T)$$
$$= -H_{a'}(T|X)$$
$$= \mathbb{E}_{P(x,t|a')} \left[ \log P(t|x,a') \right] \tag{8}$$

We have $\overset{(1)}{\leq}$ by using data processing inequality. Finally, we have:

$$\mathbb{E}_{P(z|a')} \left[ \mathcal{D}_{KL} \left( P(t|z,a') \parallel P(t|z,a) \right) \right]$$
$$\overset{(1)}{=} \mathbb{E}_{P(z|a')} \left[ \mathcal{D}_{KL} \left( P(t|z,a') \parallel P(t|z,a) \right) \right] - \mathbb{E}_{P(x|a')} \left[ \mathcal{D}_{KL} \left( P(t|x,a') \parallel P(t|x,a) \right) \right]$$
$$= \mathbb{E}_{P(t,z|a')} \left[ \log P(t|z,a') - \log P(t|z,a) \right] - \mathbb{E}_{P(x,t|a')} \left[ \log P(t|x,a') - \log P(t|x,a) \right]$$
$$= \left( \mathbb{E}_{P(t,z|a')} \left[ \log P(t|z,a') \right] - \mathbb{E}_{P(x,t|a')} \left[ \log P(t|x,a') \right] \right)$$
$$+ \left( \mathbb{E}_{P(x,t|a')} \left[ \log P(t|x,a) \right] - \mathbb{E}_{P(t,z|a')} \left[ \log P(t|z,a) \right] \right)$$
$$\overset{(2)}{=} 0 \tag{9}$$

We have $\overset{(1)}{=}$ because the shift between two domains w.r.t. input space $\mathcal{X}$ is covariate shift; $\overset{(2)}{=}$ by using Eq. (7) and Eq. (8) and the fact that KL-divergence is non-negative. Note that Eq. (9) implies that the shift between these two domains w.r.t. representation space $\mathcal{Z}$ is also covariate shift (i.e., $P(t|z,a) = P(t|z,a') = P(t|z), \forall a, a' \in \mathcal{A}$).

Table A1: Overview of medical image datasets for fair TTE prediction evaluation.

| Dataset | Prediction Task | Modality | Subgroup | Attribute | # images | Censoring rate | Mean TTE |
|---|---|---|---|---|---|---|---|
| AREDS | Late AMD | Retinal Fundus | | Total | 129708 | 83.9% | 4.4 (years) |
| | | | Age | ≤70 | 44224 | 83.1% | 5.1 |
| | | | | >70 | 85484 | 84.4% | 3.9 |
| | | | Sex | Female | 71837 | 83.0% | 4.4 |
| | | | | Male | 57871 | 85.1% | 4.3 |
| | | | Race | Non-white | 4888 | 99.2% | 3.1 |
| | | | | White | 124820 | 83.3% | 4.4 |
| MIMIC-CXR | In-hospital Mortality | Chest X-ray | | Total | 269360 | 61.7% | 488.6 (days) |
| | | | Age | ≤60 | 103437 | 77.3% | 503.3 |
| | | | | >60 | 165923 | 52.0% | 484.2 |
| | | | Sex | Female | 125742 | 63.9% | 514.4 |
| | | | | Male | 143618 | 59.7% | 468.4 |
| | | | Race | Non-white | 83234 | 66.7% | 487.2 |
| | | | | White | 186126 | 59.4% | 489.1 |
| ADNI | Alzheimer's Disease | Brain MRI | | Total | 2227 | 63.2% | 35.9 (months) |
| | | | Age | ≤80 | 1597 | 62.2% | 37.1 |
| | | | | >80 | 630 | 65.7% | 32.6 |
| | | | Sex | Female | 986 | 64.7% | 38.5 |
| | | | | Male | 1241 | 62.0% | 33.9 |

# C  Dataset Details

## C.1  AREDS

### C.1.1  Dataset Description

The Age-Related Eye Disease Study (AREDS) [14] was a clinical trial conducted between 1992 and 2001 across 11 retinal specialty clinics in the United States. The primary objective was to study the risk factors for age-related macular degeneration (AMD) and the impact of dietary supplements on AMD progression. The study followed 4,757 participants, aged 55–80 at enrollment, for a median of 6.5 years. Participants were selected with a broad range of AMD severity, from no AMD to late-stage AMD in one eye. At each visit, certified technicians captured color fundus photography images using a standardized imaging protocol, although adherence to the protocol varied, leading to visits at irregular intervals. AMD severity scores were determined by expert graders at the University of Wisconsin Fundus Photograph Reading Center, with late AMD defined as the presence of neovascular AMD or atrophic AMD with geographic atrophy (severity scores from 10 to 12 using severity scale [8]). For this study, we include images from both the left and right eyes of each participant and make predictions separately for each eye. In cases where multiple images were captured per eye during a visit, we select only one image for analysis. Demographic information, including age, sex, and race, is available in the dataset and is used as sensitive attributes in our study.

### C.1.2  TTE Outcome Construction

In our study, TTE outcomes for fundus images are defined as the duration, in years, from the date an image was captured to the first recorded diagnosis of late AMD in the corresponding eye. For eyes without a recorded diagnosis of late AMD, the censoring dates are set to the time of the last imaging visit. To ensure that the model forecasts the future risk of developing late AMD, all images taken during the final visit for a given eye were excluded from the dataset, as this visit was used solely to determine the TTE outcome. By removing the final visit, we ensured that no images were included with late AMD already present at the time of acquisition. This process resulted in a final dataset of 129,708 fundus images, each paired with corresponding TTE information, enabling a robust analysis of the TTE prediction for AMD progression.

### C.1.3  Data Access

AREDS is a publicly available dataset hosted in the National Center for Biotechnology Information (NCBI) database of Genotypes and Phenotypes (dbGAP) through controlled access. Researchers can request access to this dataset at dbGAP. Once the application is approved, researchers can access data

at the following address: `https://www.ncbi.nlm.nih.gov/projects/gap/cgi-bin/study.cgi?study_id=phs000001.v3.p1`

## C.2  MIMIC-CXR

### C.2.1  Dataset Description

The MIMIC-CXR dataset [27] is a comprehensive, publicly available collection of chest X-ray images, along with associated clinical data, from the larger MIMIC-IV [26] database. It includes over 370,000 chest X-ray images from more than 65,000 patients, annotated with both structured and unstructured clinical information, such as patient demographics, diagnoses, and other relevant clinical details. For our study, we utilize the MIMIC-CXR-JPG [27], a processed version of the MIMIC-CXR dataset, which provides images in JPG format derived from the original DICOM files. Additionally, we link this dataset to the broader MIMIC database to access patient demographic information, including age, sex, and race, which we incorporate as sensitive attributes in our fairness benchmarking framework.

### C.2.2  TTE outcome construction

To construct TTE outcomes for chest X-ray images, we extract in-hospital mortality events from the MIMIC-IV database. For patients without a recorded date of mortality, the censoring dates are set as 1 year after their last recorded discharge date. We exclude any images that do not have a matching record in the MIMIC-IV patient table, images taken after the latest discharge date, and images taken after a recorded date of mortality. TTE is then calculated as the number of days from the image study date to either the date of mortality or the censoring date. This process results in a final dataset of 269,360 chest X-ray images, each paired with corresponding TTE information, enabling a robust analysis of the TTE prediction for in-hospital mortality.

### C.2.3  Data Access

MIMIC-CXR, MIMIC-CXR-JPG, and MIMIC-IV are a publicly available datasets hosted by PhysioNet, which is a platform providing access to medical data. To access the dataset, researchers first need to create an account on PhysioNet and then complete the required training (CITI training). Once researchers have completed the CITI training, they will need to request access to the dataset at the following address:

- MIMIC-IV: `https://physionet.org/content/mimiciv/3.1/`
- MIMIC-CXR-JPG: `https://physionet.org/content/mimic-cxr-jpg/2.1.0/`

## C.3  ADNI

### C.3.1  Dataset Description

The Alzheimer's Disease Neuroimaging Initiative (ADNI) [49] is a large, longitudinal study aimed at identifying biomarkers for Alzheimer's disease (AD) and tracking the progression of the disease over time. Launched in 2004, ADNI is one of the most comprehensive datasets for studying Alzheimer's disease and other neurodegenerative disorders. It includes a wide range of data types, including clinical assessments, neuroimaging (MRI, PET), genetic data, and fluid biomarkers (e.g., cerebrospinal fluid and blood samples) from over 1,700 participants. These participants are categorized into different diagnostic groups, including cognitively normal individuals, those with mild cognitive impairment (MCI), and individuals with Alzheimer's disease. For our study, we focus on neuroimaging data, specifically MRI scans, and consider age and sex as sensitive attributes in our analysis.

### C.3.2  TTE outcome construction

In our study, TTE outcomes for brain MRI scans are defined as the duration, measured in 6-month intervals, from the date an MRI scan was captured to the first recorded diagnosis of Alzheimer's disease. For participants without a recorded diagnosis of Alzheimer's disease, censoring dates are set to the time of their last imaging visit. To ensure that the model predicts the future risk of developing Alzheimer's disease, we excluded all MRI scans taken during the final visit, as this visit was used

solely for determining the TTE outcome. By removing the final visit, we ensured that no MRI scans were included for participants who had already been diagnosed with Alzheimer's disease at the time of acquisition. This process resulted in a final dataset of 2,227 brain MRI scans, each paired with corresponding TTE information, enabling a comprehensive analysis of TTE prediction for Alzheimer's disease progression.

### C.3.3 Data Access

ADNI is a publicly available dataset hosted in the Image and Data Archive (IDA), a secure online resource for archiving, exploring and sharing neuroscience data. Access to the ADNI dataset requires that researchers register for an IDA account. Once the account is created and the ADNI Data Use Agreement is completed, they can access data at the following address: `https://adni.loni.usc.edu`

## D  Algorithms Details

### D.1  TTE Prediction Models

#### D.1.1  PMF

PMF [40] is a discrete-time model designed for TTE prediction tasks. It represents the survival time PMF $P(t|x)$ using a neural network $f(\cdot; \theta) : \mathcal{X} \to [0, 1]^L$ where $\theta$ denotes the model parameters and $L$ represents the number of time intervals. The model parameters $\theta$ are estimated by minimizing the negative log-likelihood, averaged over the training data, as follows:

$$\mathcal{L}_{PMF}(\theta) = -\frac{1}{n} \sum_{i=1}^{n} \left\{ \delta_i \log \left( f_{\kappa(y_i)}(x_i; \theta) \right) + (1 - \delta_i) \log \left( \sum_{m=\kappa(y_i)+1}^{L} f_m(x_i; \theta) \right) \right\}$$

where $n$ is the number of training samples, $\kappa(y_i)$ denotes the specific time interval (from $1, 2, \cdots, L$) that corresponds to time $y_i$, and $f_m$ represents the predicted probability that the TTE falls within the $m - th$ interval.

#### D.1.2  DeepHit

DeepHit [43] extends the PMF model by incorporating a ranking loss function alongside the negative log-likelihood. Given a comparable set defined as $\mathcal{E} := \{(i, j) \in [n] \times [n] : \delta_i = 1, y_i < y_j\}$, this ranking loss is calculated over the training data, as follows:

$$\mathcal{L}_{DeepHit}^{rank}(\theta) = \sum_{(i,j) \in \mathcal{E}} \exp \left( \frac{- \left( \sum_{m=1}^{\kappa(y_i)} f_m(x_i; \theta) - \sum_{m'=1}^{\kappa(y_j)} f_{m'}(x_j; \theta) \right)}{\sigma} \right)$$

where $n$ is the number of training samples, $\kappa(y_i)$ denotes the specific time interval (from $1, 2, \cdots, L$) that corresponds to time $y_i$, and $f_m$ represents the predicted probability that the TTE falls within the $m - th$ interval.

#### D.1.3  Nnet-survival

Nnet-survival [15] is another discrete-time model designed for TTE prediction tasks. It represents the hazard function $h(\cdot|x)$ using a neural network $f(\cdot; \theta) : \mathcal{X} \to \mathbb{R}^L$ where $\theta$ denotes the model parameters and $L$ represents the number of time intervals. Specifically, Nnet-survival sets the hazard function equal to:

$$h[\ell | x; \theta] := \frac{1}{1 + e^{-f_\ell(x; \theta)}} \quad \text{for } \ell \in [L], x \in \mathcal{X}$$

where $f_m(x; \theta)$ is the $m - th$ output of the neural network. The model parameters $\theta$ are estimated by minimizing the negative log-likelihood, averaged over the training data, as follows:

$$\mathcal{L}_{Nnet-survival}(\theta) = \frac{1}{n} \sum_{i=1}^{n} \left\{ \delta_i \log \left( 1 + e^{-f_{\kappa(y_i)}(x_i; \theta)} \right) + (1 - \delta_i) \log \left( 1 + e^{f_{\kappa(y_i)}(x_i; \theta)} \right) \right.$$
$$\left. + \sum_{m=1}^{\kappa(y_i)-1} \log \left( 1 + e^{f_m(x_i; \theta)} \right) \right\}$$

where $n$ is the number of training samples and $\kappa(y_i)$ denotes the specific time interval (from $1, 2, \cdots, L$) that corresponds to time $y_i$.

## D.2 Fairness Algorithms

### D.2.1 Distributional Group Optimization

Distributional Group Optimization [55, 22] is an optimization technique designed to enhance model robustness by minimizing the worst-case training loss across different groups. Instead of optimizing for the average performance, GroupDRO focuses on the group with the highest loss, ensuring that the model does not disproportionately underperform on any particular group. By incorporating increased regularization, this approach helps mitigate disparities in model performance across diverse subpopulations, making it particularly useful in settings where fairness and reliability across groups are critical.

### D.2.2 Subgroup Rebalancing

This resampling method [31] addresses class imbalance by upsampling minority groups, ensuring that all groups have equal representation during training. By increasing the frequency of underrepresented samples, the model is exposed to a more balanced dataset, reducing bias and improving fairness. This approach helps prevent the model from being overly influenced by the majority group, leading to more equitable predictions across all groups.

### D.2.3 Fair Representation Learning

A common approach to promoting fairness in machine learning is through fair representation learning, which aims to obfuscate sensitive group membership information in the learned representations. By ensuring that the model's latent features do not encode discriminatory patterns, this method helps mitigate bias in downstream predictions. Following [70], we incorporate fairness constraints into representation learning by leveraging kernel-based distribution matching via Maximum Mean Discrepancy. This technique enforces similarity in feature distributions across different groups, reducing disparities while preserving task-relevant information.

### D.2.4 Domain Independence

The Domain Independence [71] is a method that trains separate classifiers for different groups while utilizing a shared encoder. This approach allows the model to capture group-specific patterns through distinct classifiers while maintaining a common feature representation across all groups. By leveraging a shared encoder, DomainInd enhances generalization and reduces the risk of overfitting to individual groups, ultimately improving the model's robustness and fairness in diverse domains.

### D.2.5 Controlled for Sensitive Attribute

This approach [80, 51], similar to Domain Independence, involves training separate models for each group based on the values of a sensitive attribute. By doing so, the method captures group-specific patterns while maintaining model flexibility. During the prediction phase, the outputs of the fitted models are averaged across all groups in the population. This averaging process ensures that no single group disproportionately influences the predictions, promoting fairness and reducing bias in the model's outcomes.

### D.3 Model Architecture Details

Each (fair) TTE prediction model comprises two key networks: an image encoder and a classifier. The image encoder transforms images into representation vectors, while the classifier predicts survival time intervals based on these learned representations. In our study, we employ a 2D EfficientNet [61] backbone as the image encoder for the AREDS and MIMIC-CXR datasets, and a 3D ResNet-18 backbone [65] for the ADNI dataset. To adapt these models for our task, we replace the original fully connected layers with new layers that map images into the representation space. The architectural details of our 2D and 3D image encoders, along with the classifier, are presented in Table A2.

Table A2: Architecture details of (fair) TTE prediction models. In our experiments, we set **n_channel** = 3 for both 2D and 3D images by duplicating grayscale chest X-ray and brain MRI images to obtain three-channel inputs. **feature_dim** is set to 64, and **hidden_dim** is set to 16. **n_class** is determined using the 'equidistant' discretization method, with values of 14 for AREDS, 28 for ADNI, and 128 for MIMIC-CXR.

| Networks | Layers | |
|---|---|---|
| Image Encoder | 2D image | Conv2d(input channel = **n_channel**, output channel = 32, kernel = 3) 
 MBConv1(input channel = 32, output channel = 16, kernel = 3) 
 MBConv6(input channel = 16, output channel = 24, kernel = 3) * 2 
 MBConv6(input channel = 24, output channel = 40, kernel = 5) * 2 
 MBConv6(input channel = 40, output channel = 80, kernel = 3) * 3 
 MBConv6(input channel = 80, output channel = 112, kernel = 5) * 3 
 MBConv6(input channel = 112, output channel = 192, kernel = 5) * 4 
 MBConv6(input channel = 192, output channel = 320, kernel = 3) 
 Conv2d(input channel = 320, output channel = 1280, kernel = 1) 
 AdaptiveAvgPool2d(output_size = 1) 
 Linear(input dim = 1280, output dim = feature_dim) 
 Dropout(p=0.5) |
| | 3D image | Conv3d(input channel = **n_channel**, output channel = 64, kernel = $3 \times 7 \times 7$) 
 Conv3d(input channel = 64, output channel = 64, kernel = $3 \times 3 \times 3$) * 4 
 Conv3d(input channel = 64, output channel = 128, kernel = $3 \times 3 \times 3$)* 4 
 Conv3d(input channel = 128, output channel = 256, kernel = $3 \times 3 \times 3$)* 4 
 Conv3d(input channel = 256, output channel = 512, kernel = $3 \times 3 \times 3$)* 4 
 AdaptiveAvgPool3d(output_size=(1,1,1)) 
 Linear(input dim = 512, output dim = **feature_dim**) 
 Dropout(p=0.5) |
| Classifier | Linear(input dim = **feature_dim**, output dim = **hidden_dim**) 
 Linear(input dim = **hidden_dim**, output dim = **n_classes**) | |

## E   Evaluation Metrics

### E.1   Performance Metrics

The performance metrics used in our study are defined based on [5].

#### E.1.1   Time-dependent concordance index

Harrell's concordance index (C-index) is one of the most widely used accuracy metrics in TTE prediction. It quantifies the fraction of data point pairs that are correctly ranked by a prediction model among those that can be unambiguously ranked. The C-index values range from 0 to 1, with 1 indicating perfect ranking accuracy. However, a notable limitation of the C-index is its dependence on a risk score function for ranking, which many TTE prediction models do not explicitly learn. To address this limitation, [1] introduced a time-dependent concordance index ($C^{td}$), which leverages predicted survival functions, $\hat{S}(\cdot|x)$, to assess model performance more effectively. The $C^{td}$ is computed as follows.

**Definition 3.** Suppose that we have a survival function estimate $\hat{S}(\cdot|x)$ for any $x \in \mathcal{X}$. Then using the set of comparable pairs $\mathcal{E} := \{(i, j) \in [n] \times [n] : \delta_i = 1, y_i < y_j\}$, we define the $C^{td}$ metric as:

$$C^{td} := \frac{1}{\mathcal{E}} \sum_{(i,j) \in \mathcal{E}} \mathbb{1}\left\{\hat{S}(y_i|x_i) < \hat{S}(y_i|x_j)\right\}$$

which is between 0 and 1. Higher scores are better.

### E.1.2 Time-dependent AUC

While the C-index and $C^{td}$ scores provide valuable single-number summaries of predictive accuracy in TTE prediction, they lack the ability to evaluate accuracy at a specific user-defined time, $t$. To address this limitation, [67] introduced time-dependent AUC scores ($AUC^{td}$), which explicitly depend on the chosen time point $t$. The core idea behind $AUC^{td}$ is to frame a binary classification problem for a fixed time $t$, where the "positive" class consists of data points that experienced the event no later than $t$, and the "negative" class includes those that survived beyond $t$. The survival function $\hat{S}(t|\cdot) : \mathcal{X} \to [0, 1]$ serves as the probabilistic classifier, predicting survival probabilities. A lower predicted survival probability for a given point $x$ implies a higher likelihood of belonging to the positive class. The $AUC^{td}$ score quantifies the classifier's ability to distinguish between these two classes at time $t$, offering a time-specific accuracy assessment of the model's predictions. The $AUC^{td}$ is computed as follows.

**Definition 4.** Suppose that we have a survival function estimate $\hat{S}(\cdot|x)$ for any $x \in \mathcal{X}$. Then for any $t > 0$, using the set of comparable pairs $\mathcal{E}(t) := \{(i, j) \in [n] \times [n] : \delta_i = 1, y_i \leq t, y_j > t\}$, we define the $AUC^{td}(t)$ (the $AUC^{td}$ at time $t$) as:

$$AUC^{td}(t) := \frac{\sum_{(i,j) \in \mathcal{E}(t)} w_i \mathbb{1}\left\{\hat{S}(t|x_i) < \hat{S}(t|x_j)\right\}}{\sum_{(i,j) \in \mathcal{E}(t)} w_i}$$

where $w_1, w_2, \cdots, w_n \in [0, \infty)$ are inverse probability of censoring weights to be defined as $w_i := 1/(\hat{S}_{censor}(y_i)\hat{S}_{censor}(t))$, and $\hat{S}_{censor}(t)$ is an estimation of $S_{censor}(t) := P(C > t)$ using Kaplan-Meier estimator [33]. $AUC^{td}(t)$ is between 0 and 1 and higher scores are better. Finally, we can get $AUC^{td}$ as

$$AUC^{td} := \frac{1}{t_{\max} - t_{\min}} \int_{t_{\min}}^{t_{\max}} AUC^{td}(u)du$$

where $t_{\min}$ and $t_{\min}$ are user-specified lower and upper limits of integration.

### E.1.3 Integrated Brier Score

The Integrated Brier Score ($IBS$) is a performance metric that directly evaluates the error of an estimated survival function $\hat{S}(\cdot|x)$ without relying on ranking. The $IBS$ is calculated as follows.

**Definition 5.** Suppose that we have a survival function estimate $\hat{S}(\cdot|x)$ for any $x \in \mathcal{X}$. Then for any $t > 0$, we define the $BS(t)$ (the $IBS$ at time $t$) as:

$$BS(t) := \frac{1}{N} \sum_{i=1}^{n} \left( \frac{\hat{S}(t|x_i)^2 \delta_i \mathbb{1}\{y_i \leq t\}}{\hat{S}_{censor}(y_i)} + \frac{\left(1 - \hat{S}(t|x_i)^2\right) \mathbb{1}\{y_i > t\}}{\hat{S}_{censor}(t)} \right)$$

which is nonnegative. Lower scores are better. Finally, we can get $IBS$ as

$$IBS := \frac{1}{t_{\max} - t_{\min}} \int_{t_{\min}}^{t_{\max}} BS(u)du$$

where $t_{\min}$ and $t_{\min}$ are user-specified lower and upper limits of integration.

## E.2 Fairness Metrics

In this study, we define fairness metrics as the predictive performance gaps between groups. This kind of metric is used ensure that the model maintains equal predictive performance across different groups. In particular, given a performance metric Er for TTE prediction task, we define fairness metric $\mathcal{F}_{\text{Er}}$ as follow:

$$\mathcal{F}_{\text{Er}}(h) = \max_{a,a' \in \mathcal{A}} |\text{Er}_a - \text{Er}_{a'}|$$

where $\mathcal{A}$ is the set of groups considered in TTE prediction task, $\text{Er}_a$ and $\text{Er}_{a'}$ are the predictive performance metrics calculated from subsets containing data from groups $a$ and $a'$, respectively. For each predictive performance metric defined above, we have a corresponding fairness metric as follows.

$$\mathcal{F}_{C^{td}} = \max_{a,a' \in \mathcal{A}} \left| C_a^{td} - C_{a'}^{td} \right|$$

$$\mathcal{F}_{AUC^{td}} = \max_{a,a' \in \mathcal{A}} \left| AUC_a^{td} - AUC_{a'}^{td} \right|$$

$$\mathcal{F}_{IBS} = \max_{a,a' \in \mathcal{A}} \left| IBS_a - IBS_{a'} \right|$$

where $C_a^{td}, AUC_a^{td}, IBS_a$ are predictive performance metrics calculated from the subset containing data from group $a$, and $C_{a'}^{td}, AUC_{a'}^{td}, IBS_{a'}$ are predictive performance metrics calculated from the subset containing data from group $a'$.

## E.3 Fairness-Utility Trade-Off Metric

The fairness metrics mentioned above do not capture the fairness-utility trade-off while in medical context, it is essential to balance fairness and utility to ensure that the model is not only fair but also accurate and effective for all groups. To handle this issue, we leverage the equity-scaling metric ($ES$) [44] that takes both utility and fairness into account for evaluation. Similar to fairness metric, for each predictive performance metric, we have a corresponding fairness-utility trade-off metric as follows.

$$ES_{C^{td}} = \frac{C_D^{td}}{1 + \sum_{a \in \mathcal{A}} \left| C_D^{td} - C_{D_a}^{td} \right|}$$

$$ES_{AUC^{td}} = \frac{AUC_D^{td}}{1 + \sum_{a \in \mathcal{A}} \left| AUC_D^{td} - AUC_{D_a}^{td} \right|}$$

$$ES_{IBS} = \frac{1 - IBS_D}{1 + \sum_{a \in \mathcal{A}} \left| IBS_D - IBS_{D_a} \right|}$$

The advantage of the equity-scaling metric lies in its intuitive interpretability. Specifically, a higher equity-scaling score indicates that the model is both more accurate and more equitable simultaneously.

# F Experimental Setup Details

## F.1 Implementation Details

### F.1.1 Hardware Usage

The experiments were conducted at a supercomputing center utilizing multiple compute nodes. Each node was equipped with an NVIDIA Volta V100 GPU with 16 GB of memory, an Intel Xeon CPU, and 32 GB of RAM, ensuring the computational resources necessary for large-scale experiments. In total, we trained over 20,000 models, requiring approximately 4.56 GPU years of computational effort, highlighting the extensive scale of our study.

### F.1.2 Package Usage

The FairTTE benchmark is implemented using Python 3, with PyTorch [47] serving as the framework for deep learning computations. The implementation of TTE models is built on the pycox [41]

package, while the evaluation metrics for TTE prediction leverage pycox, scikit-survival [50], and SurvivalEVAL [52]. Additionally, the training and evaluation pipeline for TTE prediction models is adapted from the demo code provided in [5], ensuring a robust and standardized framework for benchmarking.

## F.2 Data Split and Pre-processing

**Data Split.**  Each dataset in our study was divided into training, validation, and testing sets using a 60%:20%:20% split ratio. Models were trained on the training sets, evaluated on the testing sets, and the validation sets were used for model selection. Since a single patient may have multiple medical records, we took precautions to prevent data leakage during model training. Specifically, the data was split by patient, ensuring that no patient appearing in the testing set had any records in the training or validation sets. This approach maintains the integrity of the evaluation process and ensures that model performance is assessed on entirely unseen patient data.

**Data Pre-processing.**  Before being fed into the TTE prediction models, chest X-ray and color fundus images are resized to $224 \times 224$ pixels, while brain MRI scans are resized to $128 \times 128 \times 96$. Additionally, all pixel values are normalized to a range of 0 to 1 to ensure stability during training and improve model performance.

We consider binary group setting in our experiment. These groups were constructed according to the following criteria:

- Race: 'Non-White' (Group 0), 'White' (Group 1)
- Sex: 'Female' (Group 0), 'Male' (Group 1)
- Age:
    - MIMIC-CXR: '$\leq 60$' (Group 0), '$> 60$' (Group 1)
    - AREDS: '$\leq 70$' (Group 0), '$> 70$' (Group 1)
    - ADNI: '$\leq 80$' (Group 0), '$> 80$' (Group 1)

## F.3 Hyperparameter Search

To ensure a fair comparison, we perform a grid-based hyperparameter search using 10 random seeds. The details of the hyperparameter search for the methods used in our experiments are provided below.

- TTE prediction models
    - Learning rate: $10^x$ where $x \sim Uniform(-4, -3)$
    - Decay rate: $10^x$ where $x \sim Uniform(-6, -4)$
- Fair TTE prediction models
    - $\eta : 10^x$ where $x \sim Uniform(-3, -1)$ (DRO)
    - $\lambda : 10^x$ where $x \sim Uniform(-5, 2)$ (FRL)

For standard TTE prediction models, we select the best models based on their predictive performance metrics calculated on the validation sets. In contrast, for fair TTE prediction models, we prioritize fairness metrics when selecting the best models, allowing for up to a 5% reduction in accuracy compared to the baseline TTE models. This approach ensures a balanced trade-off between fairness and predictive performance.

## F.4 Quantifying Source of Bias

In order to quantify the degree of bias sources in each dataset and sensitive attribute setting, we use several metrics as follows.

**Disparity in mutual information between $X_Z$ and $Y$ across groups.**  We quantify the disparity in mutual information between $X_Z$ (i.e., image representation generated from the vision backbones)

and $Y$ across groups by computing the maximum difference in their normalized mutual information values across all groups, as defined below.

$$Bias_{MI(X_Z,Y)} = \max_{a,a'\in\mathcal{A}} \left| \frac{2I(X_Z,Y|A=a,\Delta=1)}{H(X_Z|A=a,\Delta=1)+H(Y|A=a,\Delta=1)} \right.$$
$$\left. - \frac{2I(X_Z,Y|A=a',\Delta=1)}{H(X_Z|A=a',\Delta=1)+H(Y|A=a',\Delta=1)} \right|$$

where $I(\cdot,\cdot|A=a,\Delta=1)$ represented the mutual information conditioned on $A=a$ and $\Delta=1$ and $H(\cdot|A=a)$ denotes the entropy conditioned on $A=a$ and $\Delta=1$.

**Disparity in mutual information between $X_Z$ and $\Delta$ across groups.** Similarly, we quantify the disparity in mutual information between $X_Z$ (i.e., image representation generated from the vision backbones) and $\Delta$ across groups by computing the maximum difference in their normalized mutual information values across all groups, as defined below.

$$Bias_{MI(X_Z,\Delta)} = \max_{a,a'\in\mathcal{A}} \left| \frac{2I(X_Z,\Delta|A=a)}{H(X_Z|A=a)+H(\Delta|A=a)} - \frac{2I(X_Z,\Delta|A=a')}{H(X_Z|A=a')+H(\Delta|A=a')} \right|$$

**Disparity in TTE distribution across groups.** We measure the disparity in TTE distributions across groups by calculating the maximum Wasserstein distance [63], normalized by the range of TTE, between the TTE distributions of each group. This is defined as follows:

$$Bias_{TTE} = \max_{a,a'\in\mathcal{A}} \left| \frac{\mathcal{W}\left(P(Y|A=a,\Delta=1), P(Y|A=a',\Delta=1)\right)}{\max_{y\in\mathcal{Y}} y} \right|$$

where $\mathcal{W}(\cdot,\cdot)$ denotes the Wasserstein-1 distance between the two distributions.

**Disparity in image distribution across groups.** We measure the disparity in image distributions across groups by calculating the maximum Wasserstein distance [63], normalized by the range of image feature values, between the image distributions of each group. This is defined as follows:

$$Bias_{Image} = \max_{a,a'\in\mathcal{A}} \left| \frac{\mathcal{W}\left(P(X_Z|A=a), P(X_Z|A=a')\right)}{\max_{y\in\mathcal{Y}} y} \right|$$

where $\mathcal{W}(\cdot,\cdot)$ denotes the Wasserstein-1 distance between the two distributions. Due to the high dimensionality of image representations, we implement sliced Wasserstein distance [3], a variant of the Wasserstein distance that approximates the full Wasserstein distance between high-dimensional distributions by projecting them onto one-dimensional subspaces and averaging the resulting 1D Wasserstein distances.

**Disparity in censoring rate across groups.** We quantify the disparity in censoring rates across groups by calculating the maximum normalized difference between the means of the censoring distributions for each group, as defined below.

$$Bias_{Censoring} = \max_{a,a'\in\mathcal{A}} \left| \frac{\mathbb{E}\left[\Delta|A=a\right] - \mathbb{E}\left[\Delta|A=a'\right]}{\mathbb{E}\left[\Delta\right]} \right|$$

### F.5 Constructing Causal Distribution Shift

To construct distribution shift between training and testing data, we modify the training data by introducing correlations between the sensitive attribute and other RVs in the causal graph (Figure 2). This adjustment simulates real-world scenarios where biases in data collection or underlying relationships may lead to disparities across groups. The details of this process, including the specific modifications applied to establish these correlations, are outlined below.

- **Distribution shift on $X$:** Images from disadvantaged groups are degraded using a Gaussian blur filter to simulate lower-quality data.
- **Distribution shift on $Y$:** TTE labels for disadvantaged groups are corrupted by adding noise sampled from a uniform distribution.

- **Distribution shift on $\Delta$:** To simulate biased censoring, we flip the censoring indicators for 90% of uncensored samples within disadvantaged groups.

We note that, although these distribution shifts are synthetic, they mimic real-world scenarios, as described below.

- Adding noise to images mimics real-world scenarios such as patients in different geographic locations are scans with different equipment. This causes the medical image to appear systematically different for groups in each location.

- Adding noise to TTE labels mimics real-world scenarios such as delayed or inaccurate event recording in EHR system.

- Flipping censoring indices mimics real-world scenarios in which certain groups experience less consistent access to care due to financial or geographic. Thus, these groups are more likely to drop out of care, resulting in a higher censoring rate.

## G  Causal Graphs for Fairness in TTE Prediction

### G.1  Causal Graphs for Biased and Unbiased Settings

Figure A1 presents the causal graphs for the unbiased and biased scenarios. In the unbiased scenario (Figure A1a), the sensitive attribute $A$ is unrelated to the TTE outcome and influences only $X_A$, with no effect on other variables in the graph. In contrast, in the biased scenarios (Figure A1b and Figure A1c), $A$ also affects additional variables, resulting in dependencies between $A$ and the TTE outcome. These causal pathways may be direct (Figure A1b), mediated through unobserved variables $U$ (Figure A1c), or both.

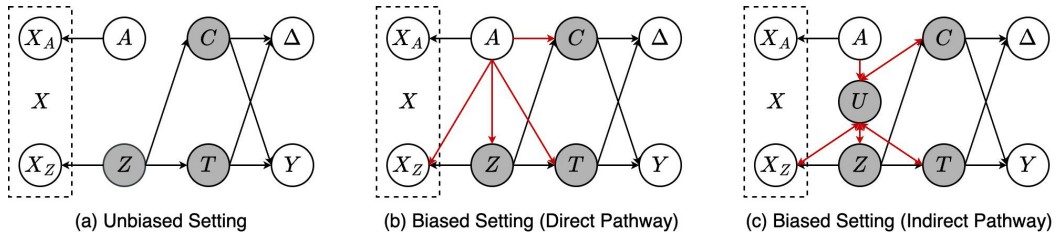

(a) Unbiased Setting     (b) Biased Setting (Direct Pathway)     (c) Biased Setting (Indirect Pathway)

Figure A1: Causal structure in TTE prediction. Gray circles denote unobserved random variables. (a) Unbiased setting, where the sensitive attribute $A$ influences only $X_A$. (b) Biased setting with direct causal pathways, where $A$ is directly associated (red arrows) with other variables in the graph. (c) Biased setting with indirect causal pathways, where $A$ influences (red arrows) other variables through unobserved variables $U$.

### G.2  Real-world Causal Graph Examples for Fairness in TTE Prediction

In this section, we present causal graphs illustrating real-world scenarios in time-to-event (TTE) prediction using medical imaging. Many of these examples are adapted from diagnostic settings in prior work [28]. We describe four scenarios in which the sensitive attribute $A$ influences other variables in the causal graph—namely, the medical image $X$, the underlying condition $Z$, the time-to-event $T$, and the censoring time $C$—leading to disparities in group-specific data distributions. For each scenario, we include two examples: one where the causal pathway from $A$ is valid (red arrows), appearing in both training and testing data, and one where the pathway is spurious (blue arrows), representing bias present only in the training data and absent in the testing data.

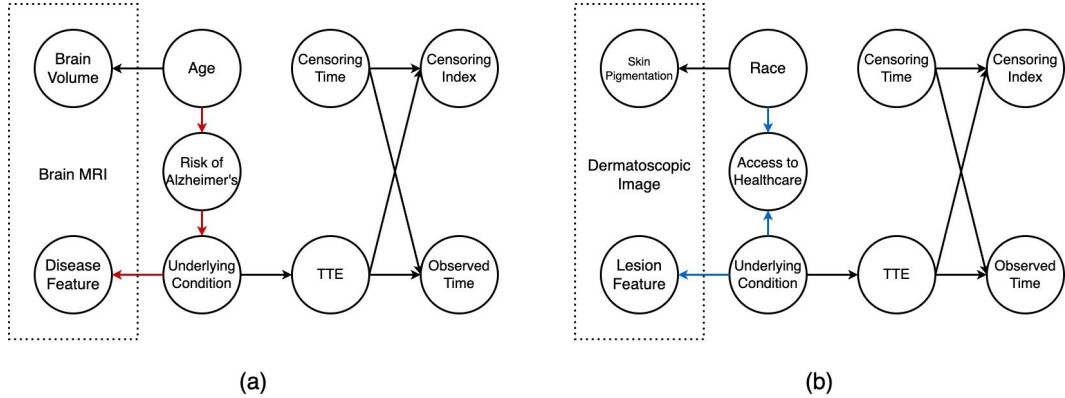

Figure A2: Causal graphs illustrating scenarios where the sensitive attribute $A$ affects the underlying condition $Z$. (a) Valid pathway: age is a known clinical risk factor for Alzheimer's disease. (b) Invalid pathway: race appears spuriously correlated with $Z$ due to disparities in healthcare access.

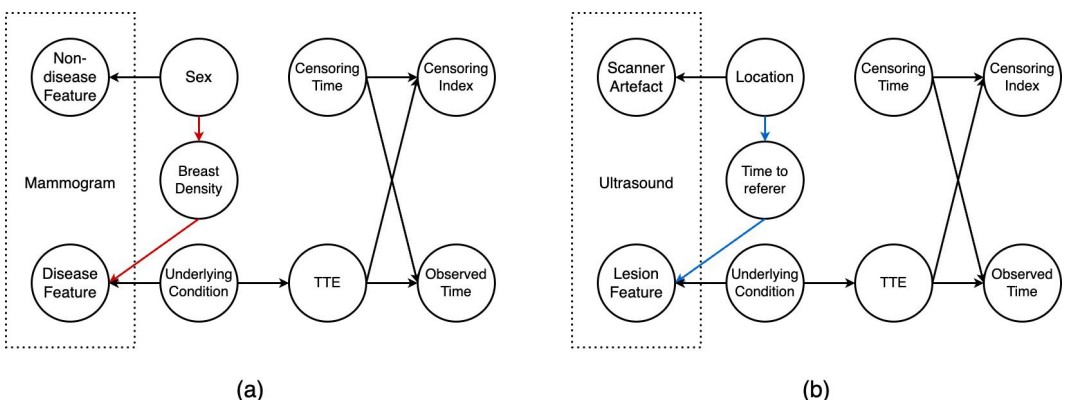

Figure A3: Causal graphs illustrating scenarios where the sensitive attribute $A$ influences the medical image $X$. (a) Valid pathway: breast cancer presents differently in men and women due to inherent differences in breast tissue. (b) Invalid pathway: spurious correlation arises when patients in different locations are imaged at varying disease stages due to inconsistent ultrasound referral policies.

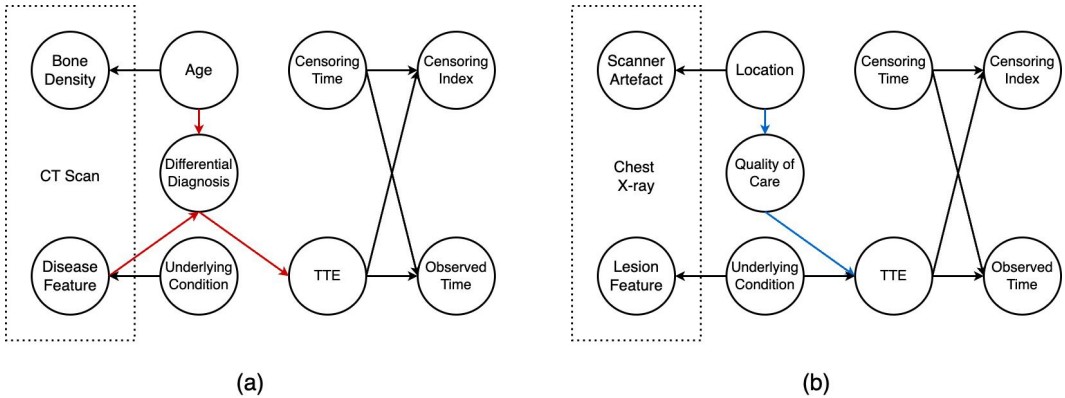

Figure A4: Causal graphs illustrating scenarios where the sensitive attribute $A$ influences the time-to-event outcome $T$. (a) Valid pathway: age contributes to differential diagnosis in epidemiology and legitimately affects disease progression. (b) Invalid pathway: a spurious correlation arises when patients from different locations receive healthcare services of varying quality, impacting $T$ in a non-causal manner.

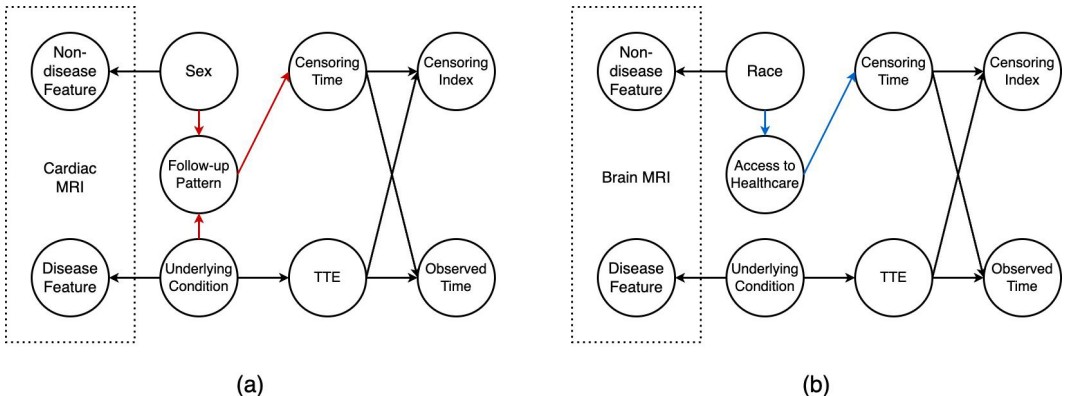

Figure A5: Causal graphs illustrating scenarios where the sensitive attribute $A$ affects the censoring time $C$. (a) Valid pathway: Women may be less likely to receive aggressive follow-up or diagnostic imaging for cardiac conditions, resulting in higher censoring for female patients. (b) Invalid pathway: race appears spuriously correlated with censoring time due to disparities in healthcare access.

# H Additional Results

## H.1 Predictive Performance and Fairness in TTE Prediction Models

Figure A6 presents the complete per-group performance results of TTE prediction models—DeepHit, Nnet-Survival, and PMF—across all dataset, sensitive attribute, and metric combinations, while Figure A7 reports the corresponding significance tests using the two-sided Wilcoxon signed-rank test. As shown, performance gaps between groups are observed across all settings.

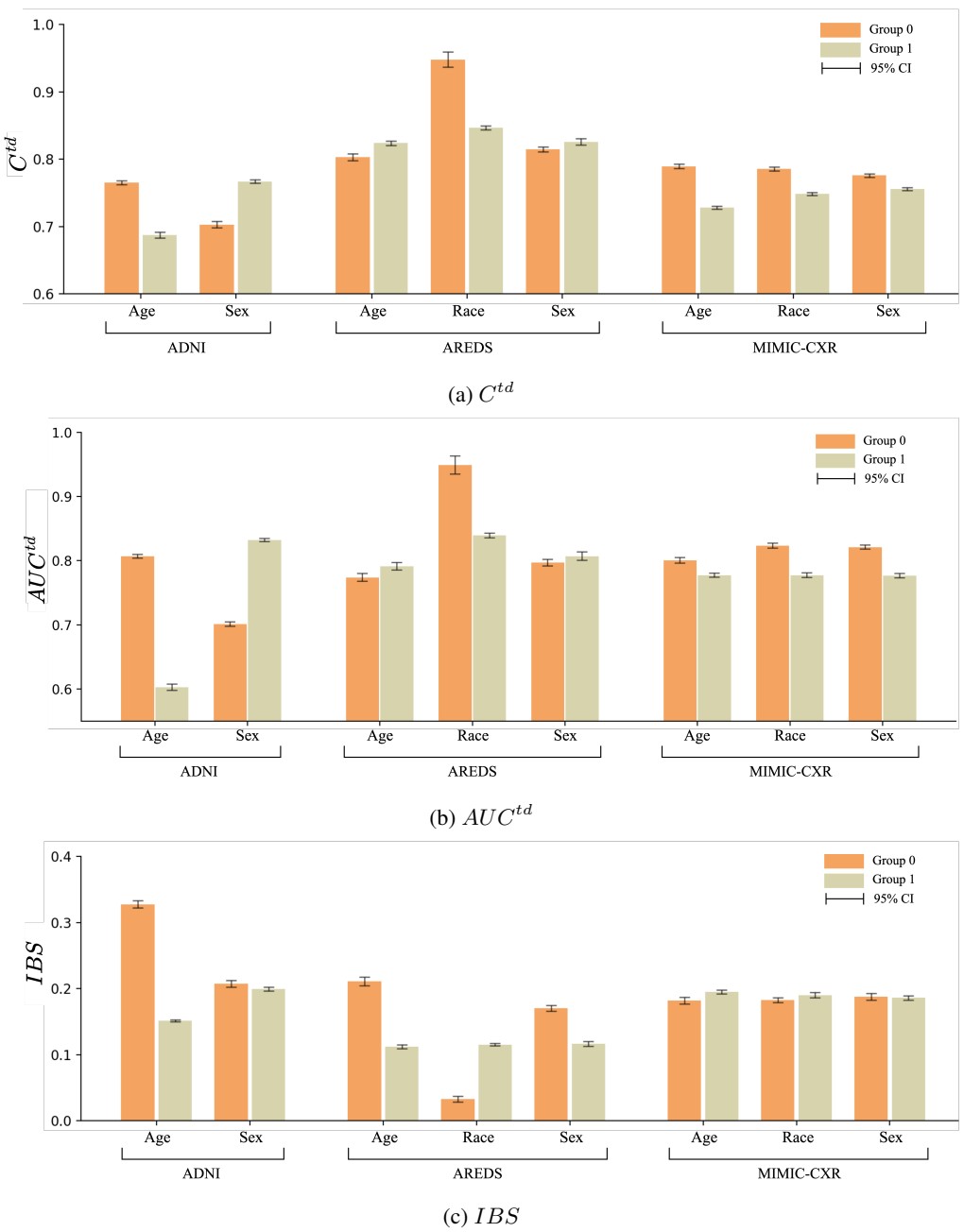

(a) $C^{td}$

(b) $AUC^{td}$

(c) $IBS$

Figure A6: Per-group predictive performances of TTE prediction models across various datasets and sensitive attribute combinations. The visualized performances correspond to the best models determined by model selection conducted on the validation sets. The 95% confidence intervals (CIs) are calculated using bootstrapping over the test sets. a) Results measured by $C^{td}$; b) Results measured by $C^{td}$; c) Results measured by $IBS$.

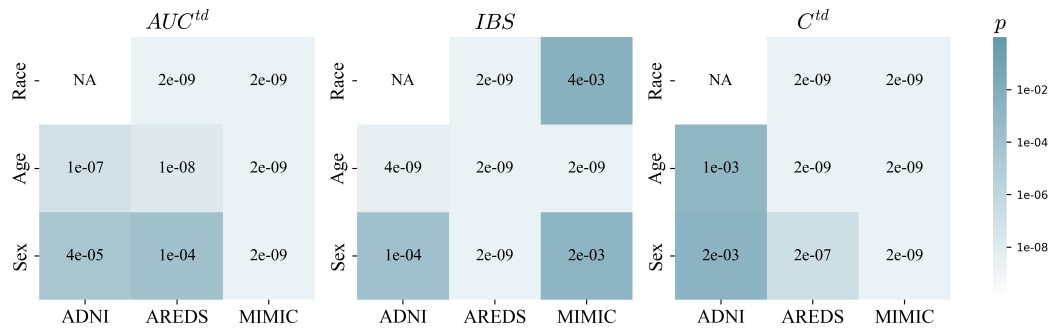

Figure A7: P-values from the two-sided Wilcoxon signed-rank test computed across all TTE prediction models and random seeds. A p-value $< 0.05$ indicates that there is significant differences in predictive performance between groups.

## H.2 Comparison between Pre-Training and Training from Scratch Strategies for TTE Prediction Models

### H.2.1 Comparison in Predictive Performance

Figure A8 presents the complete per-group predictive performance gap between pre-training and training from scratch approaches across all dataset, sensitive attribute, and metric combinations. As shown, pre-training consistently improves the predictive performance of TTE models across most settings.

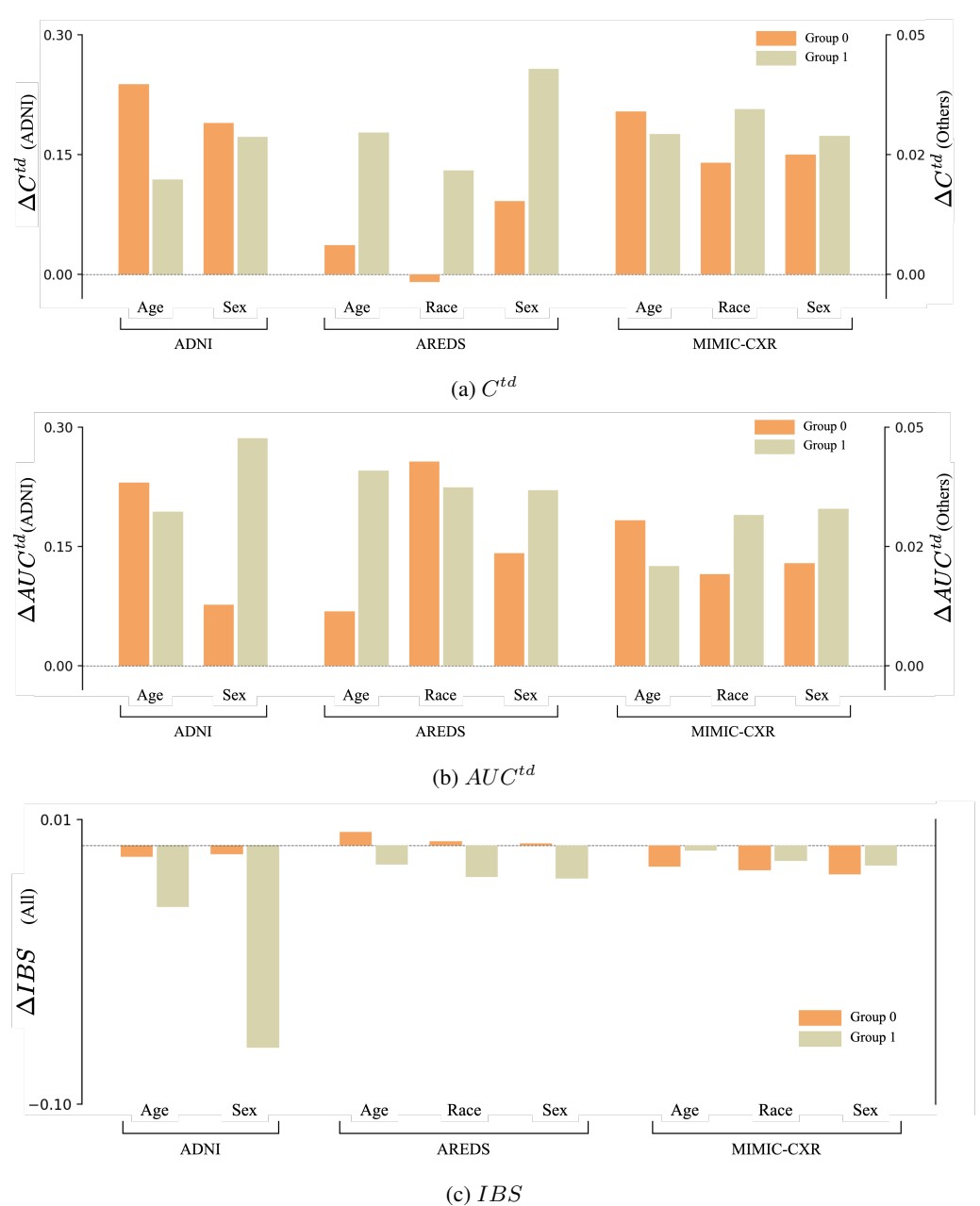

Figure A8: Per-group average performance gap for TTE prediction models using a pre-training strategy compared to training from scratch across various datasets and sensitive attribute combinations. Positive $\Delta C^{td}$ and $\Delta AUC^{td}$ and negative $\Delta IBS$ values indicate that the pre-training strategy enhances predictive performance relative to training from scratch. a) Results measured by $C^{td}$; b) Results measured by $C^{td}$; c) Results measured by $IBS$.

### H.2.2 Comparison in Fairness

Figure A9 presents the significant differences in terms of fairness between pre-training and training from scratch strategies. As shown, we do not observe a significant improvement with pre-training compared to training from scratch. Specifically, the p-values from one-sided Wilcoxon signed-rank tests are larger than 0.05 in 18 out of 24 settings, suggesting that pre-training does not lead to more equitable predictions in most cases.

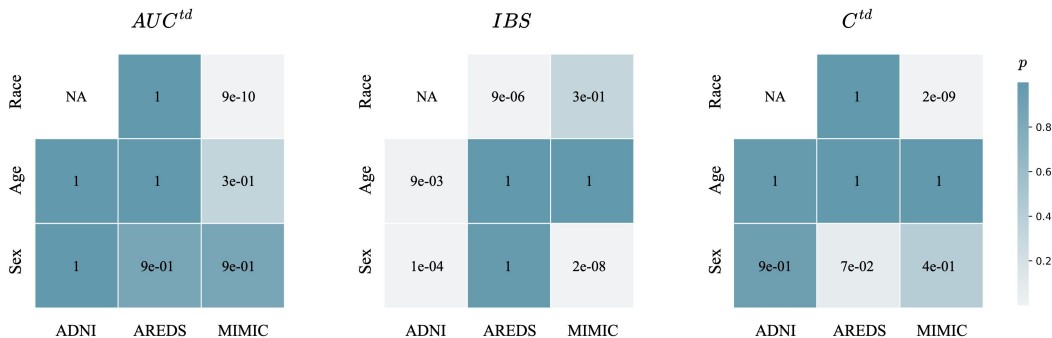

Figure A9: P-values from the one-sided Wilcoxon signed-rank test computed across all TTE prediction models and random seeds. A p-value $> 0.05$ suggests that pre-training does not result in significantly more equitable predictions compared to training from scratch.

### H.3 Comparison with Advanced Image Backbones and Medical Pre-training for TTE Prediction Models

To investigate whether more advanced vision backbones pretrained on medical imaging data can enhance the predictive performance and fairness of TTE prediction models, we conduct an additional experiment using the late AMD progression prediction task on the AREDS dataset. Specifically, we adopt RETFound—a widely recognized eye-specific foundation model based on the Vision Transformer (ViT) architecture [12] and pretrained on millions of retinal images—as the image backbone for our TTE models. As shown in Table A3, RETFound with ViT does not demonstrate any improvement over EfficientNet pretrained on ImageNet for this task, suggesting that general-purpose backbones may remain competitive despite the availability of domain-specific pretraining.

Table A3: Predictive and fairness performances of DeepHit using EfficientNet and Vision Transformer as vision backbones on AREDS dataset. EfficientNet is pretrained on ImageNet dataset while Vision Transformer is initialized with pretrained weights provided by RETFound.

| | | **EfficientNet** | | | **Vision Transformer** | | |
|---|---|---|---|---|---|---|---|
| **Sensitive Attribute** | | $AUC^{td} \uparrow$ | $IBS \downarrow$ | $C^{td} \uparrow$ | $AUC^{td} \uparrow$ | $IBS \downarrow$ | $C^{td} \uparrow$ |
| | Age | 78.41 | 15.37 | 81.30 | 78.04 | 15.22 | 80.65 |
| **Accuracy** | Sex | 79.08 | 15.36 | 81.77 | 78.01 | 14.51 | 80.72 |
| | Race | 81.78 | 11.74 | 84.53 | 80.99 | 11.99 | 83.91 |
| **Sensitive Attribute** | | $\mathcal{F}_{AUC^{td}} \downarrow$ | $\mathcal{F}_{IBS} \downarrow$ | $\mathcal{F}_{C^{td}} \downarrow$ | $\mathcal{F}_{AUC^{td}} \downarrow$ | $\mathcal{F}_{IBS} \downarrow$ | $\mathcal{F}_{C^{td}} \downarrow$ |
| | Age | 1.58 | 12.56 | 2.20 | 0.35 | 9.50 | 1.44 |
| **Fairness** | Sex | 0.76 | 3.84 | 1.32 | 0.85 | 4.26 | 0.44 |
| | Race | 14.00 | 10.14 | 11.09 | 9.32 | 10.47 | 10.27 |

## H.4 Fairness in Fair TTE Prediction Models

Table A4 and Figure A10 present the results of statistical significance testing for fair TTE prediction models, conducted using the Friedman test followed by the Nemenyi post-hoc test.

Table A4: P-values from the Friedman test followed by a Nemenyi post-hoc test computed across all dataset and sensitive attribute combinations. A p-value $< 0.05$ indicates that the significant difference in terms of fairness between the two corresponding methods.

| Metrics | Models | DI | CSA | DRO | DeepHit | FRL | SR |
|---|---|---|---|---|---|---|---|
| | DI | 1.000 | 0.995 | 0.684 | 0.420 | 1.000 | 0.967 |
| | CSA | 0.995 | 1.000 | 0.340 | 0.765 | 0.985 | 1.000 |
| $C^{td}$ | DRO | 0.684 | 0.340 | 1.000 | **0.011** | 0.765 | 0.206 |
| | DeepHit | 0.420 | 0.765 | **0.011** | 1.000 | 0.340 | 0.894 |
| | FRL | 1.000 | 0.985 | 0.765 | 0.340 | 1.000 | 0.937 |
| | SR | 0.967 | 1.000 | 0.206 | 0.894 | 0.937 | 1.000 |
| | DI | 1.000 | 0.206 | 0.894 | 0.894 | 0.340 | 0.596 |
| | CSA | 0.206 | 1.000 | 0.836 | 0.836 | 1.000 | 0.985 |
| $AUC^{td}$ | DRO | 0.894 | 0.836 | 1.000 | 1.000 | 0.937 | 0.995 |
| | DeepHit | 0.894 | 0.836 | 1.000 | 1.000 | 0.937 | 0.995 |
| | FRL | 0.340 | 1.000 | 0.937 | 0.937 | 1.000 | 0.999 |
| | SR | 0.596 | 0.985 | 0.995 | 0.995 | 0.999 | 1.000 |
| | DI | 1.000 | 0.995 | 0.985 | 0.985 | 0.596 | 0.894 |
| | CSA | 0.995 | 1.000 | 1.000 | 0.836 | 0.894 | 0.995 |
| $IBS$ | DRO | 0.985 | 1.000 | 1.000 | 0.765 | 0.937 | 0.999 |
| | DeepHit | 0.985 | 0.836 | 0.765 | 1.000 | 0.206 | 0.507 |
| | FRL | 0.596 | 0.894 | 0.937 | 0.206 | 1.000 | 0.995 |
| | SR | 0.894 | 0.995 | 0.999 | 0.507 | 0.995 | 1.000 |

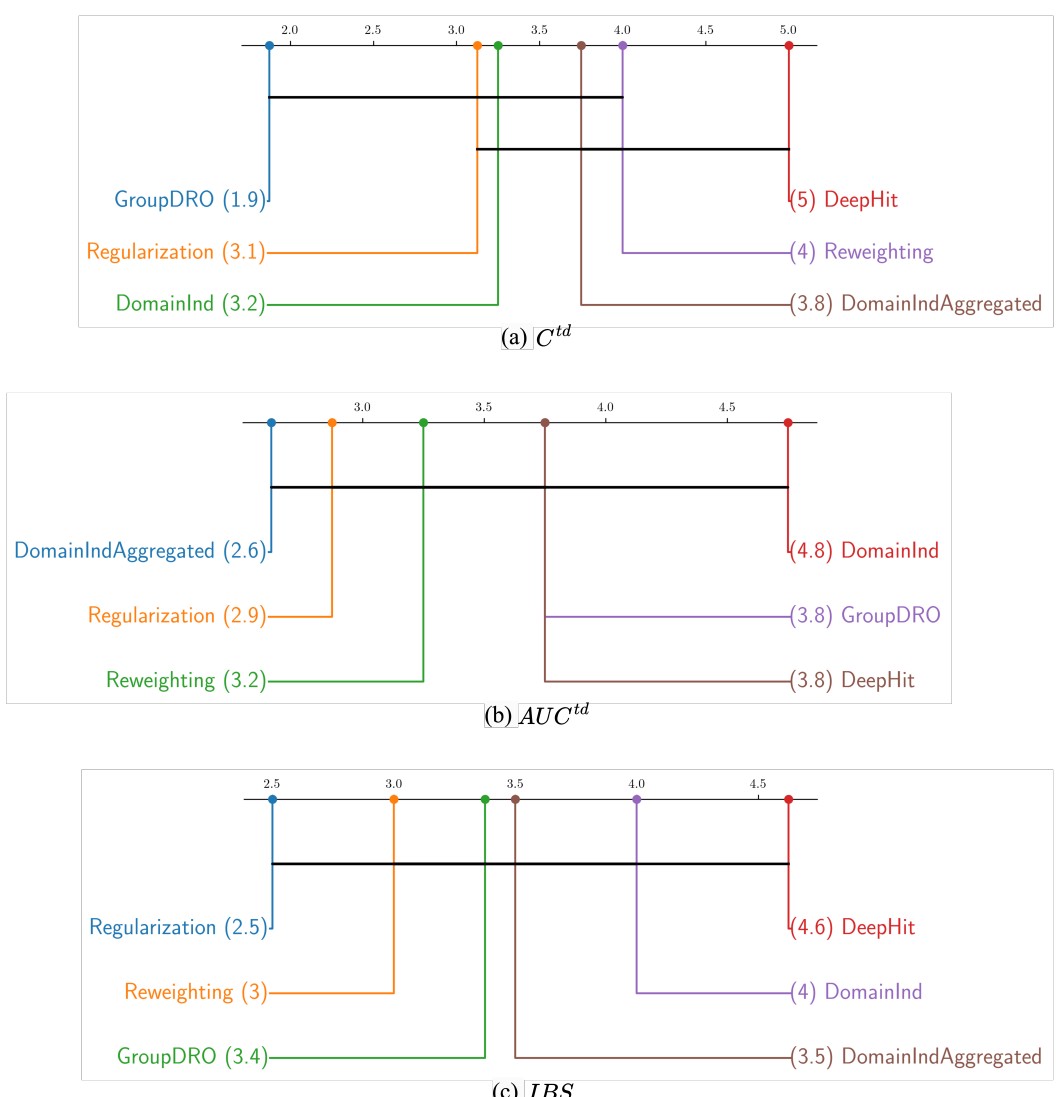

Figure A10: Critical Difference diagrams for all methods calculated from all dataset and sensitive attribute combinations. Although fairness algorithms are generally ranked higher than DeepHit in all settings, there is no significant difference in terms of fairness as indicated by the connections between fairness algorithms and DeepHit in the diagrams. a) Diagram for $C^{td}$; b) Diagram for $AUC^{td}$; c) Diagram for $IBS$.

## H.5 Fairness-Utility Trade-Off Results

Incorporating fairness shifts the objective from pure utility optimization to balancing utility and fairness. To assess this trade-off in fair TTE prediction methods, we compute equity scaling scores [44] across datasets and sensitive attributes under both in-distribution and distribution shift scenarios. As shown in Figures A11–A14, different methods exhibit varying fairness-utility trade-offs, with CSA achieving the most favorable balance in most settings.

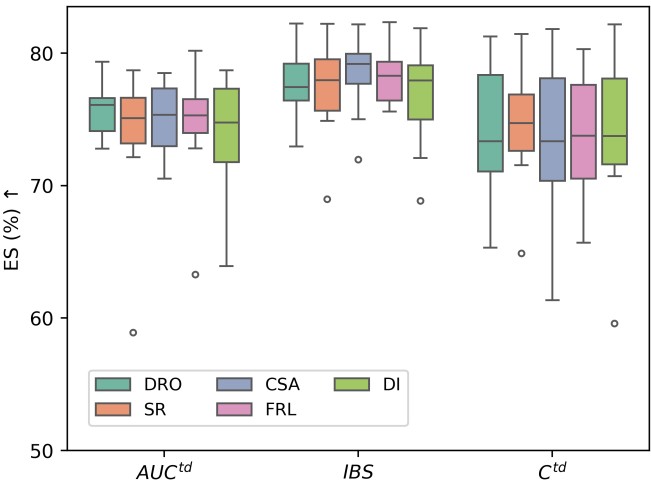

Figure A11: Fairness-utility trade-offs of fairness algorithms for TTE prediction across various utility metrics. For each metric, we compute the corresponding equity scaling score as a measure of the trade-off. The results for each fairness algorithm are aggregated across all dataset and sensitive attribute combinations.

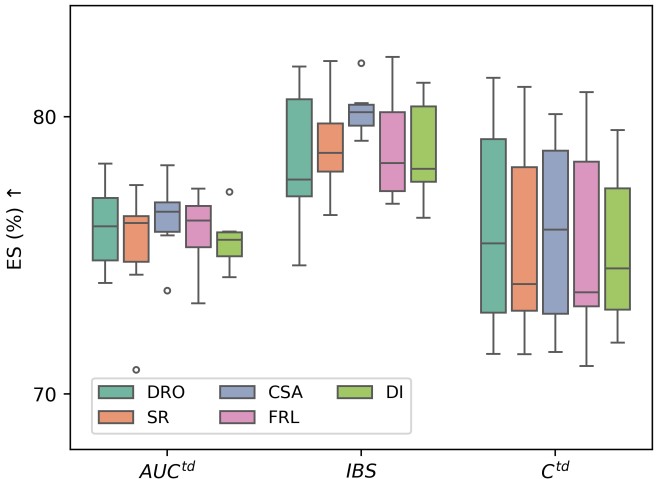

Figure A12: Fairness-utility trade-offs of fairness algorithms for TTE prediction across various utility metrics under distribution shift created by flipping censoring indices (shift on $X$). For each metric, we compute the corresponding equity scaling score as a measure of the trade-off. The results for each fairness algorithm are aggregated across all dataset and sensitive attribute combinations.

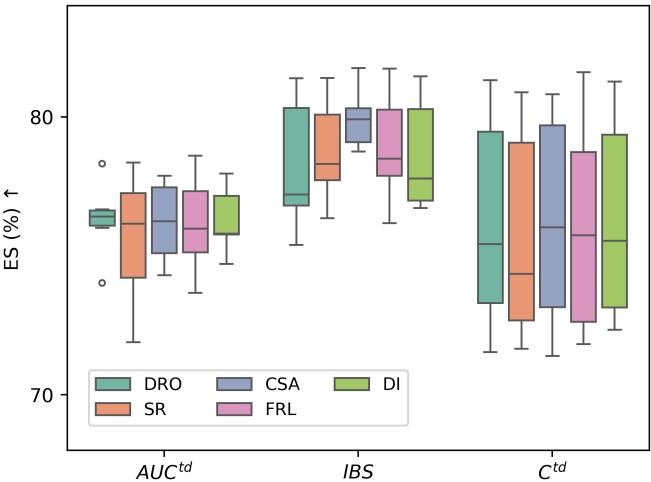

Figure A13: Fairness-utility trade-offs of fairness algorithms for TTE prediction across various utility metrics under distribution shift created by flipping censoring indices (shift on $Y$). For each metric, we compute the corresponding equity scaling score as a measure of the trade-off. The results for each fairness algorithm are aggregated across all dataset and sensitive attribute combinations.

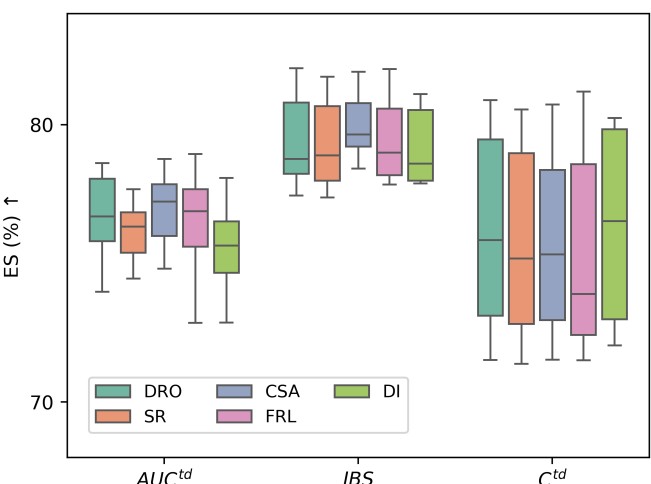

Figure A14: Fairness-utility trade-offs of fairness algorithms for TTE prediction across various utility metrics under distribution shift created by flipping censoring indices (shift on $\Delta$). For each metric, we compute the corresponding equity scaling score as a measure of the trade-off. The results for each fairness algorithm are aggregated across all dataset and sensitive attribute combinations.

## H.6 Additional Results for Predictive Performance and Fairness in Fair TTE Prediction Models under Distribution Shift

This section presents the complete results for fair TTE prediction under distribution shift scenarios. As illustrated in Figure 3, we define distribution shift as a setting where correlations between the sensitive attribute and other variables in the causal graph are present in the training data but absent in the testing data. To simulate such shifts, we intervene on one group (the intervened group) by corrupting specific aspects of the data—namely, the images ($X$), TTE labels ($Y$), or censoring indicators ($\Delta$)—while leaving the other group unchanged. Detailed procedures for generating these shifts are provided in Appendix F.5.

### H.6.1 Results for Distribution Shift in $X$

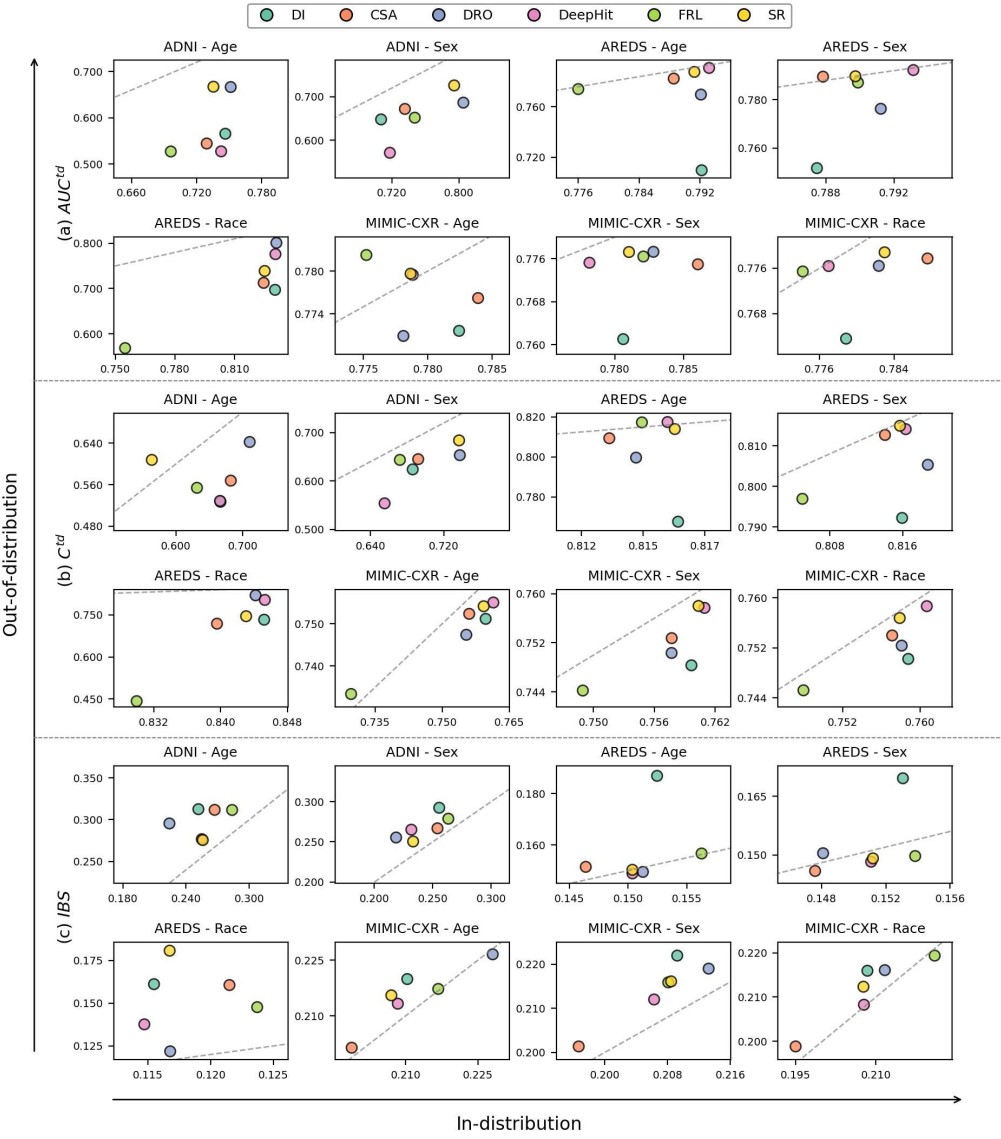

Figure A15: Comparison of predictive performance for (fair) TTE prediction models in in-distribution vs. out-of-distribution (i.e., shift in $X$) learning scenarios, evaluated across all dataset and sensitive attribute combinations. The displayed results represent the average performance across all random seeds. Points on the dashed line indicate equal performance in both scenarios. a) Results for $AUC^{td}$; b) Results for $C^{td}$; c) Results for $IBS$.

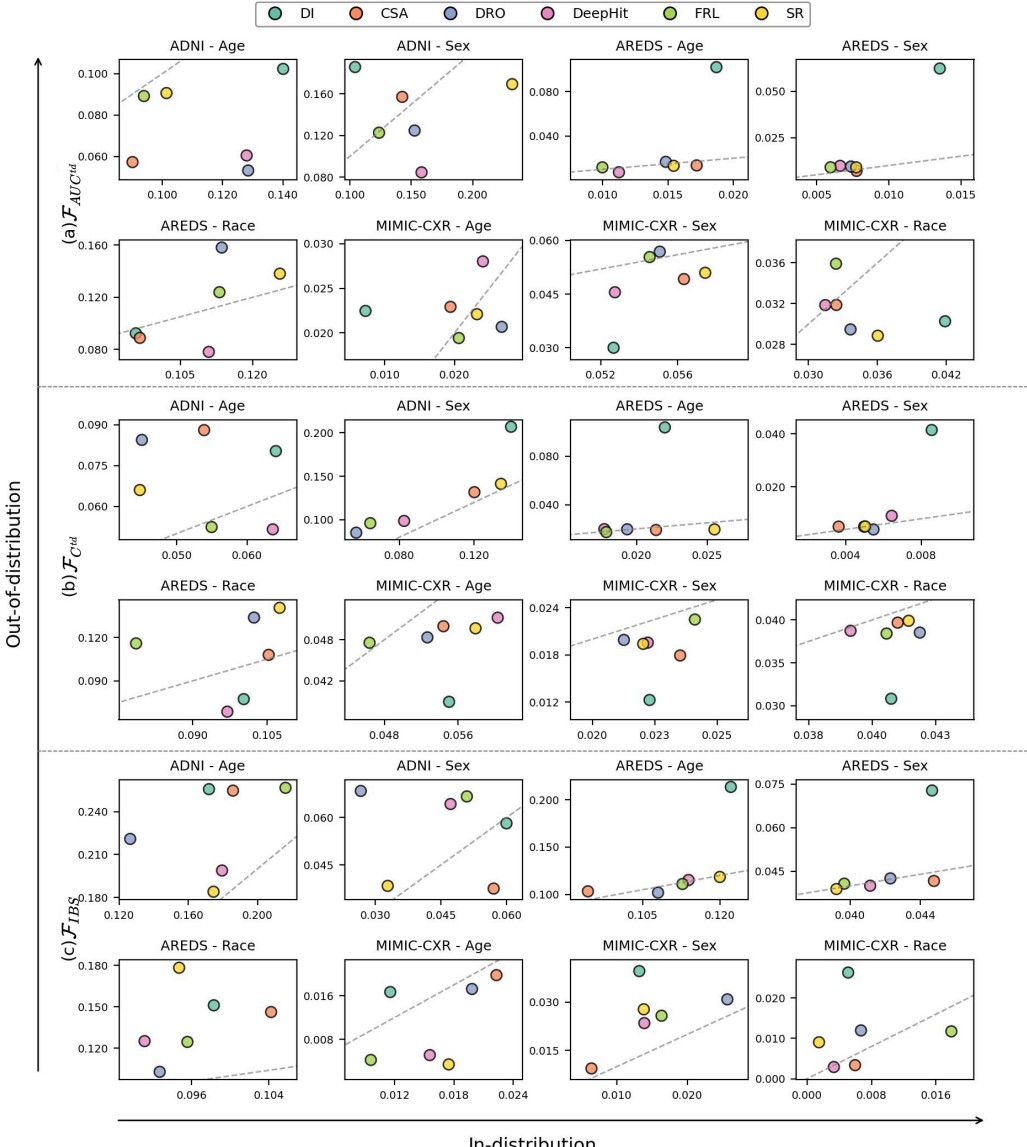

Figure A16: Comparison of fairness for (fair) TTE prediction models in in-distribution vs. out-of-distribution (i.e., shift in $X$) learning scenarios, evaluated across all dataset and sensitive attribute combinations. The displayed results represent the average performance across all random seeds. Points on the dashed line indicate equal performance in both scenarios. a) Results for $\mathcal{F}_{AUC^{td}}$; b) Results for $\mathcal{F}_{C^{td}}$; c) Results for $\mathcal{F}_{IBS}$.

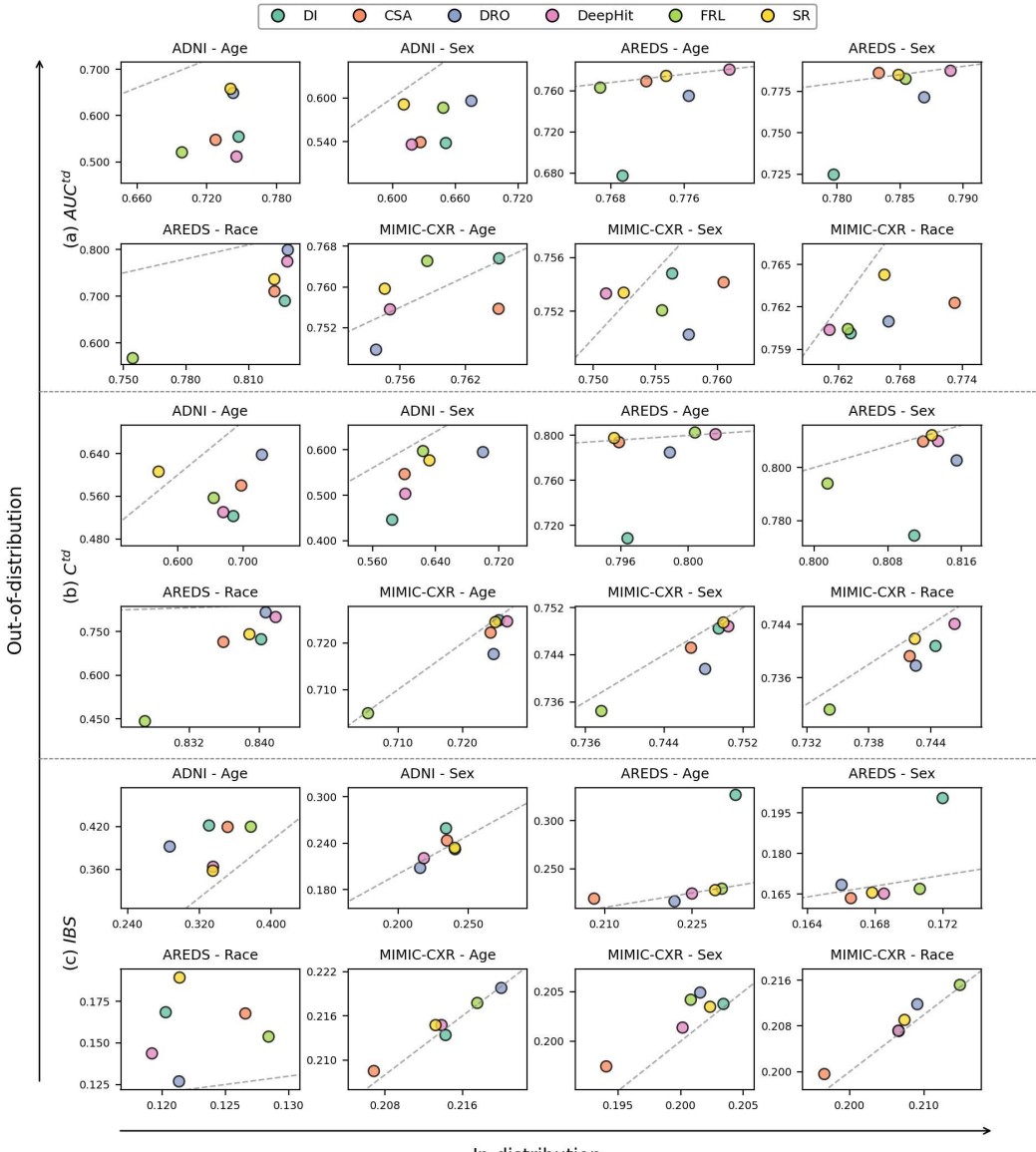

Figure A17: Comparison of predictive performance on the intervened group for (fair) TTE prediction models in in-distribution vs. out-of-distribution (i.e., shift in $X$) learning scenarios across all dataset and sensitive attribute combinations. The displayed results represent the average performance across all random seeds. Points on the dashed line indicate equal performance in both scenarios. a) Results for $AUC^{td}$; b) Results for $C^{td}$; c) Results for $IBS$.

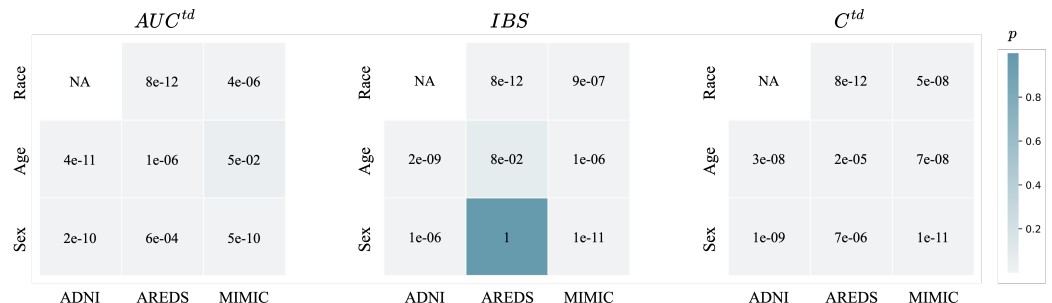

Figure A18: P-values from the one-sided Wilcoxon signed-rank test computed across all fair TTE prediction models and random seeds. A p-value $< 0.05$ suggests distribution shift on $X$ significantly degrades TTE predictive performance compared no distribution shift.

## H.6.2 Results for Distribution Shift in $Y$

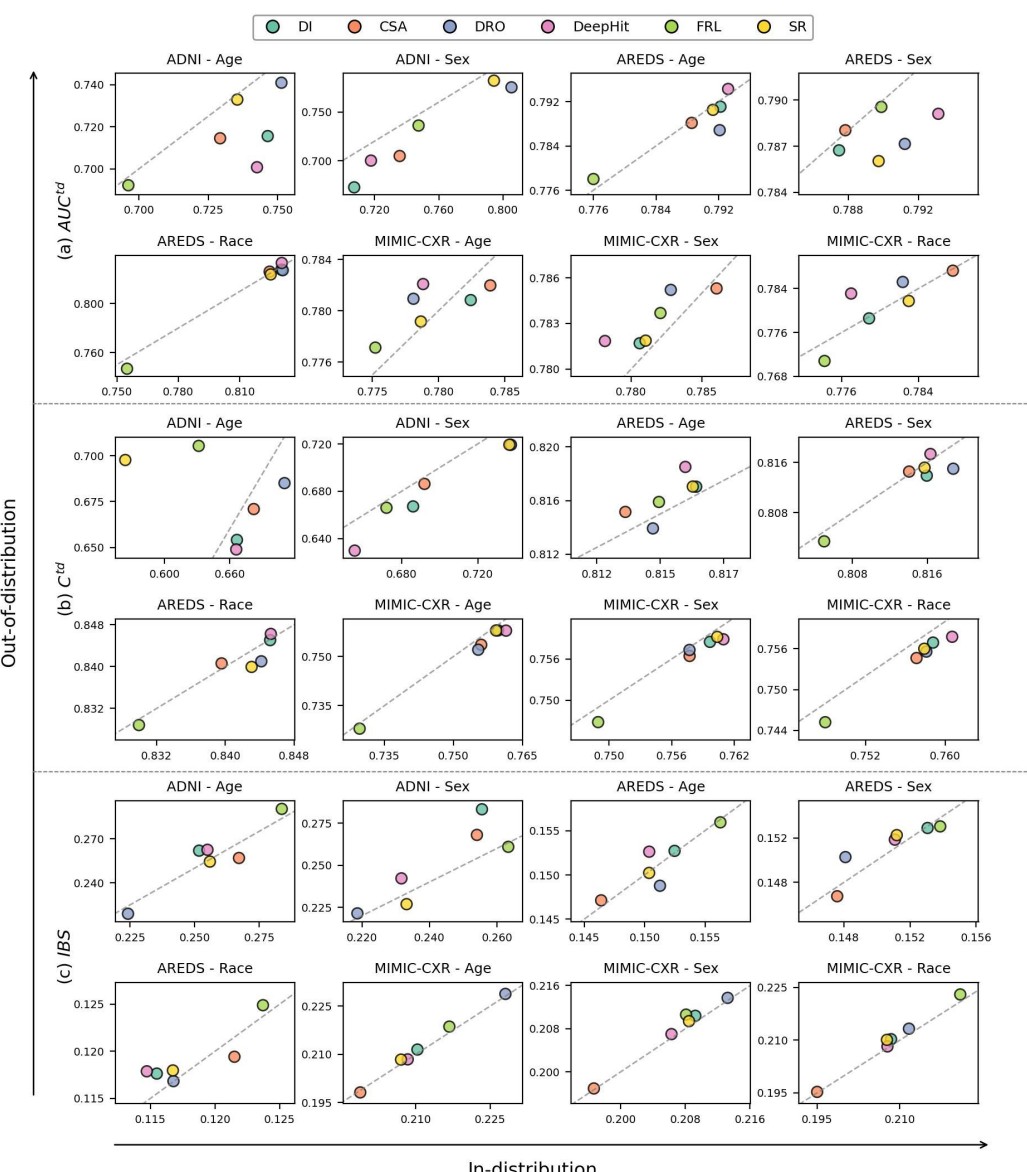

Figure A19: Comparison of predictive performance for (fair) TTE prediction models in in-distribution vs. out-of-distribution (i.e., shift in $Y$) learning scenarios, evaluated across all dataset and sensitive attribute combinations. The displayed results represent the average performance across all random seeds. Points on the dashed line indicate equal performance in both scenarios. a) Results for $AUC^{td}$; b) Results for $C^{td}$; c) Results for $IBS$.

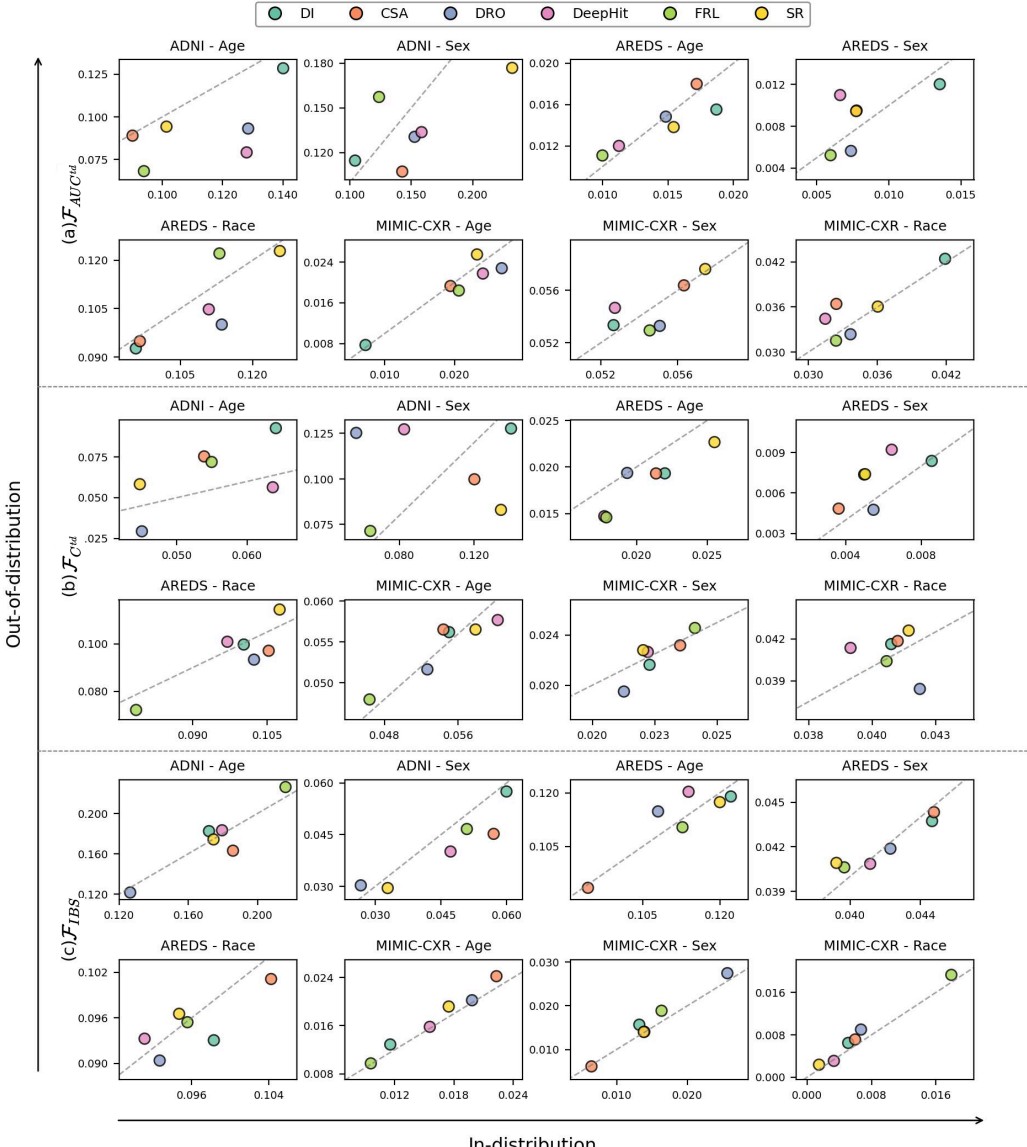

Figure A20: Comparison of fairness for (fair) TTE prediction models in in-distribution vs. out-of-distribution (i.e., shift in $Y$) learning scenarios, evaluated across all dataset and sensitive attribute combinations. The displayed results represent the average performance across all random seeds. Points on the dashed line indicate equal performance in both scenarios. a) Results for $\mathcal{F}_{AUC^{td}}$; b) Results for $\mathcal{F}_{C^{td}}$; c) Results for $\mathcal{F}_{IBS}$.

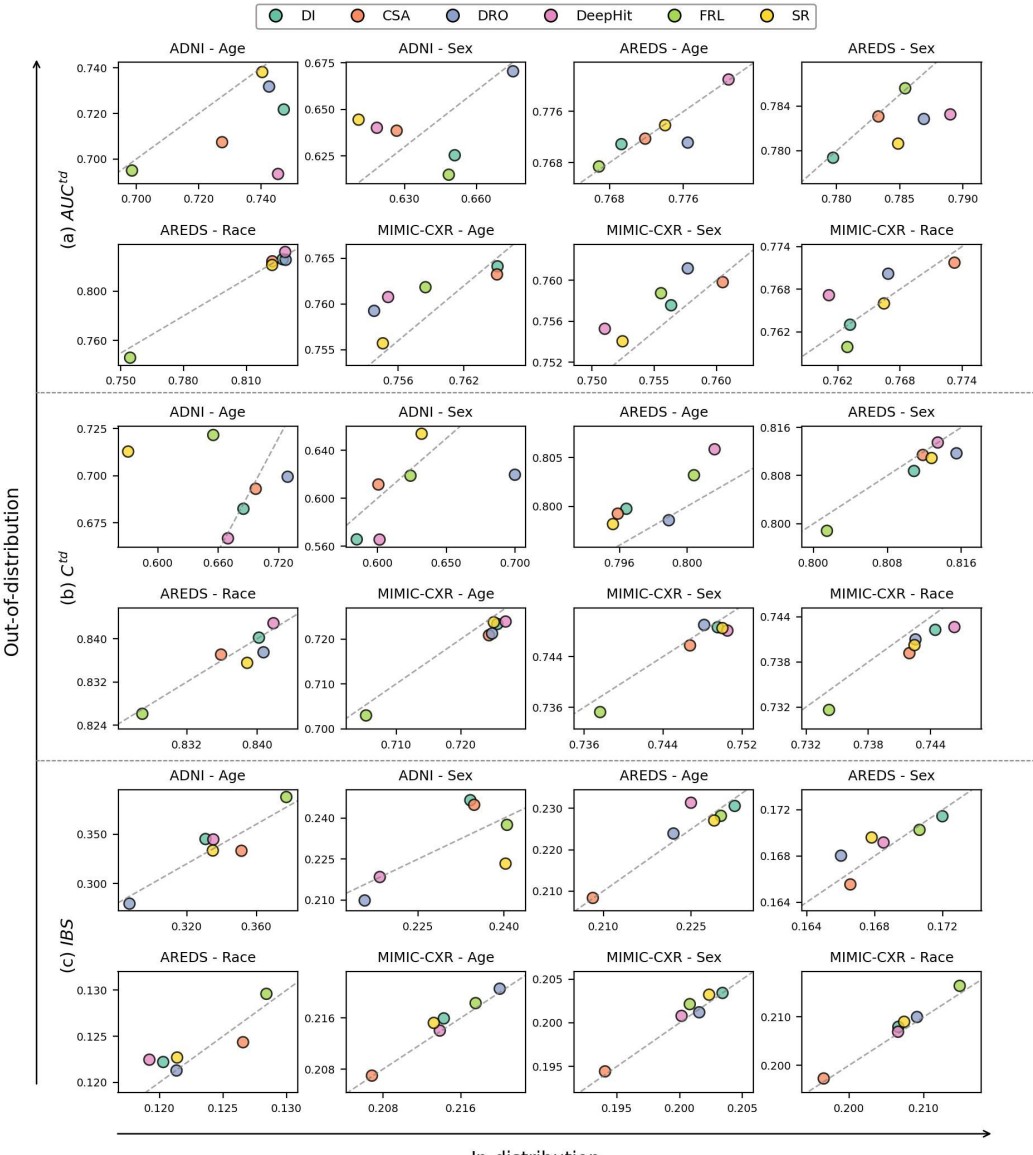

Figure A21: Comparison of predictive performance on the intervened group for (fair) TTE prediction models in in-distribution vs. out-of-distribution (i.e., shift in $Y$) learning scenarios across all dataset and sensitive attribute combinations. The displayed results represent the average performance across all random seeds. Points on the dashed line indicate equal performance in both scenarios. a) Results for $AUC^{td}$; b) Results for $C^{td}$; c) Results for $IBS$.

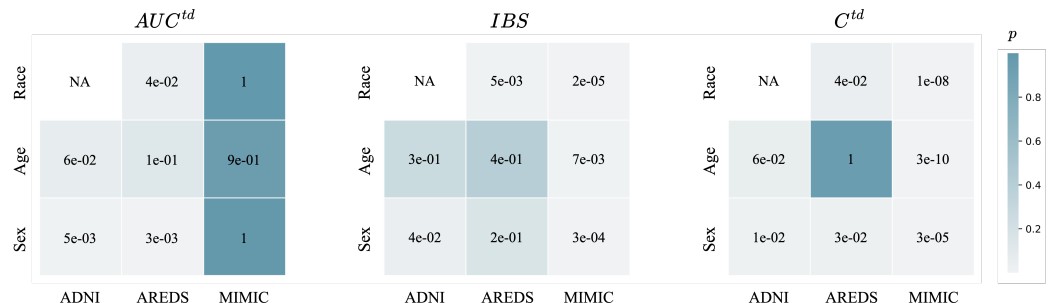

Figure A22: P-values from the one-sided Wilcoxon signed-rank test computed across all fair TTE prediction models and random seeds. A p-value $< 0.05$ suggests distribution shift on $Y$ significantly degrades TTE predictive performance compared no distribution shift.

## H.6.3 Results for Distribution Shift in $\Delta$

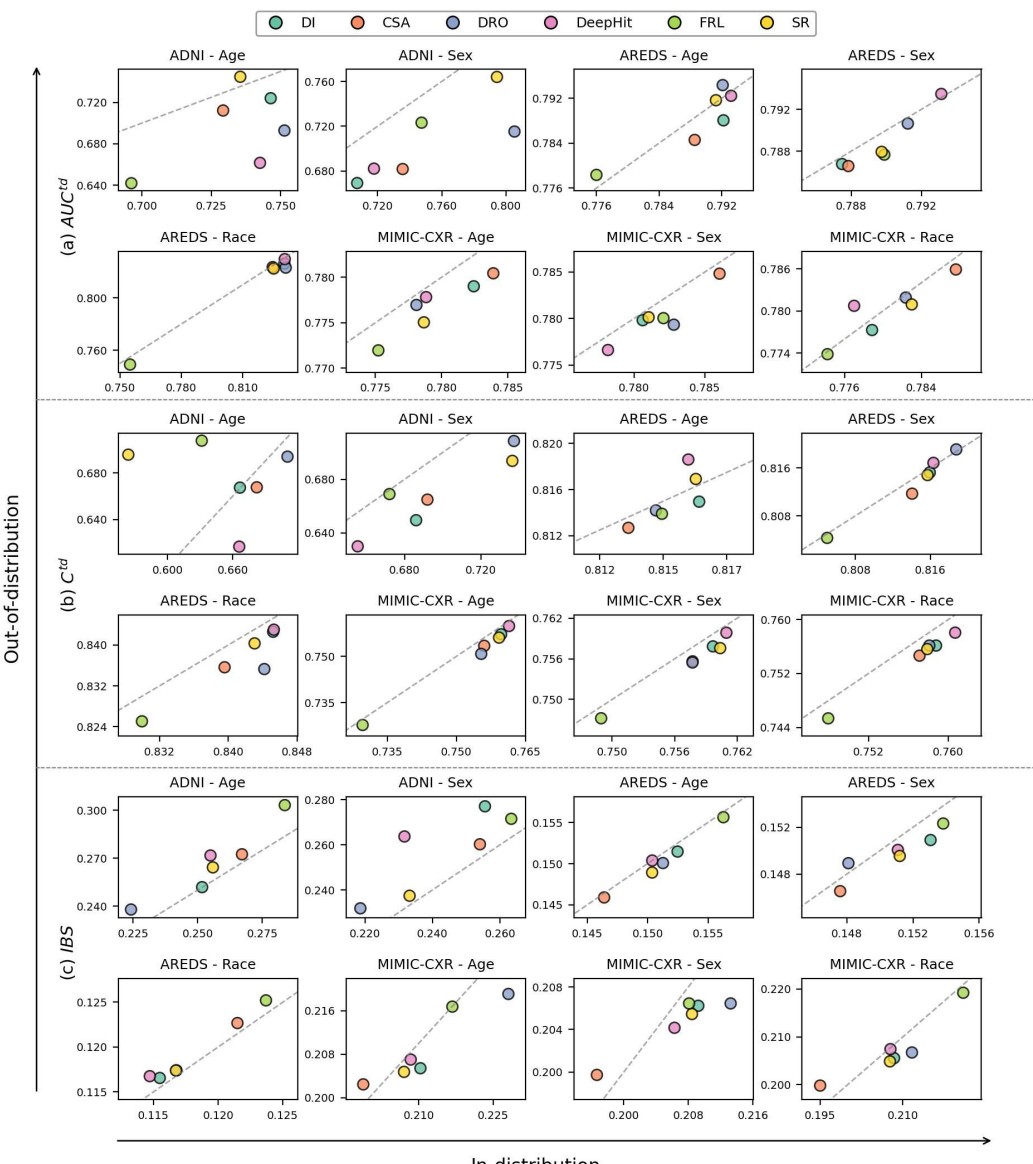

Figure A23: Comparison of predictive performance for (fair) TTE prediction models in in-distribution vs. out-of-distribution (i.e., shift in $\Delta$) learning scenarios, evaluated across all dataset and sensitive attribute combinations. The displayed results represent the average performance across all random seeds. Points on the dashed line indicate equal performance in both scenarios. a) Results for $AUC^{td}$; b) Results for $C^{td}$; c) Results for $IBS$.

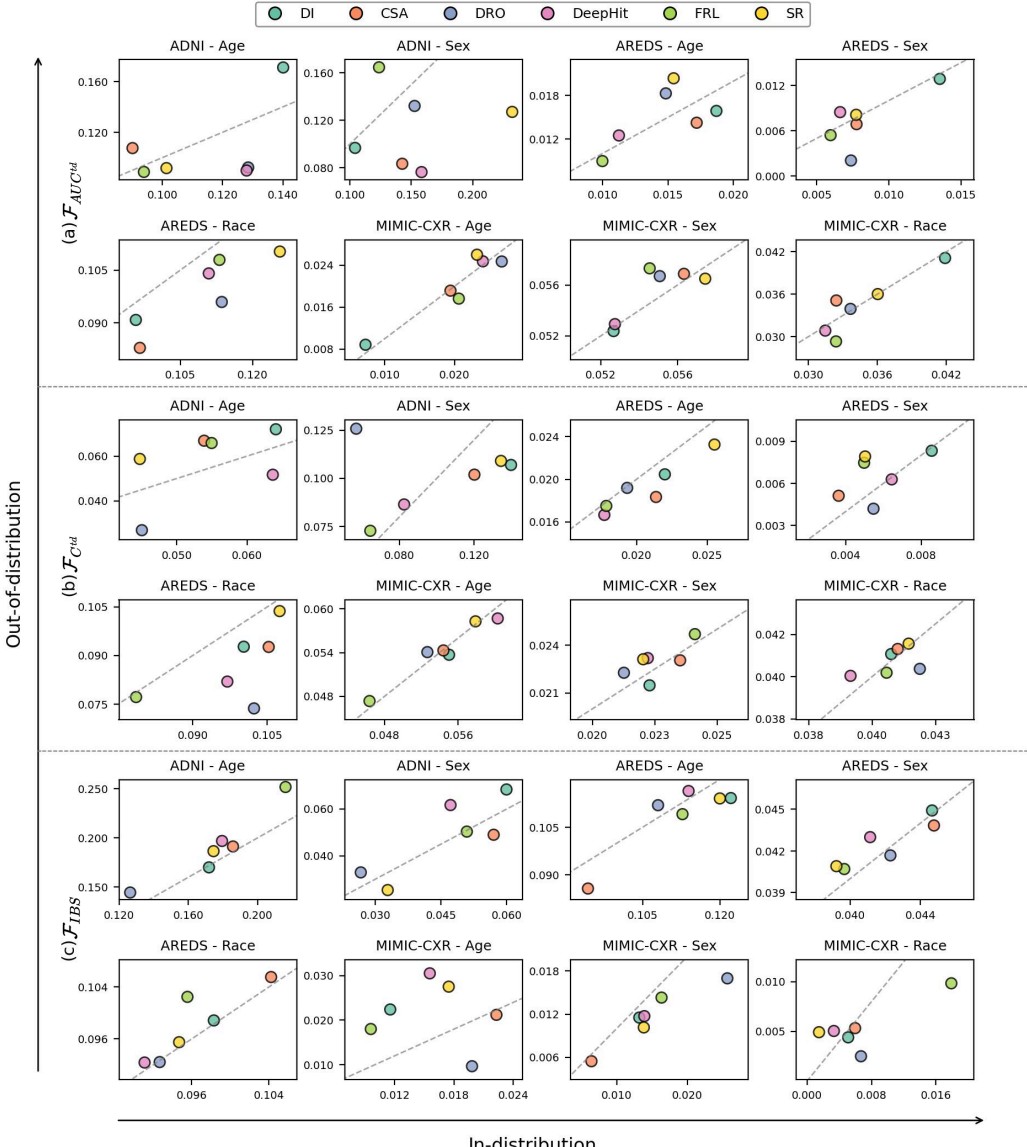

Figure A24: Comparison of fairness for (fair) TTE prediction models in in-distribution vs. out-of-distribution (i.e., shift in $\Delta$) learning scenarios, evaluated across all dataset and sensitive attribute combinations. The displayed results represent the average performance across all random seeds. Points on the dashed line indicate equal performance in both scenarios. a) Results for $\mathcal{F}_{AUC^{td}}$; b) Results for $\mathcal{F}_{C^{td}}$; c) Results for $\mathcal{F}_{IBS}$.

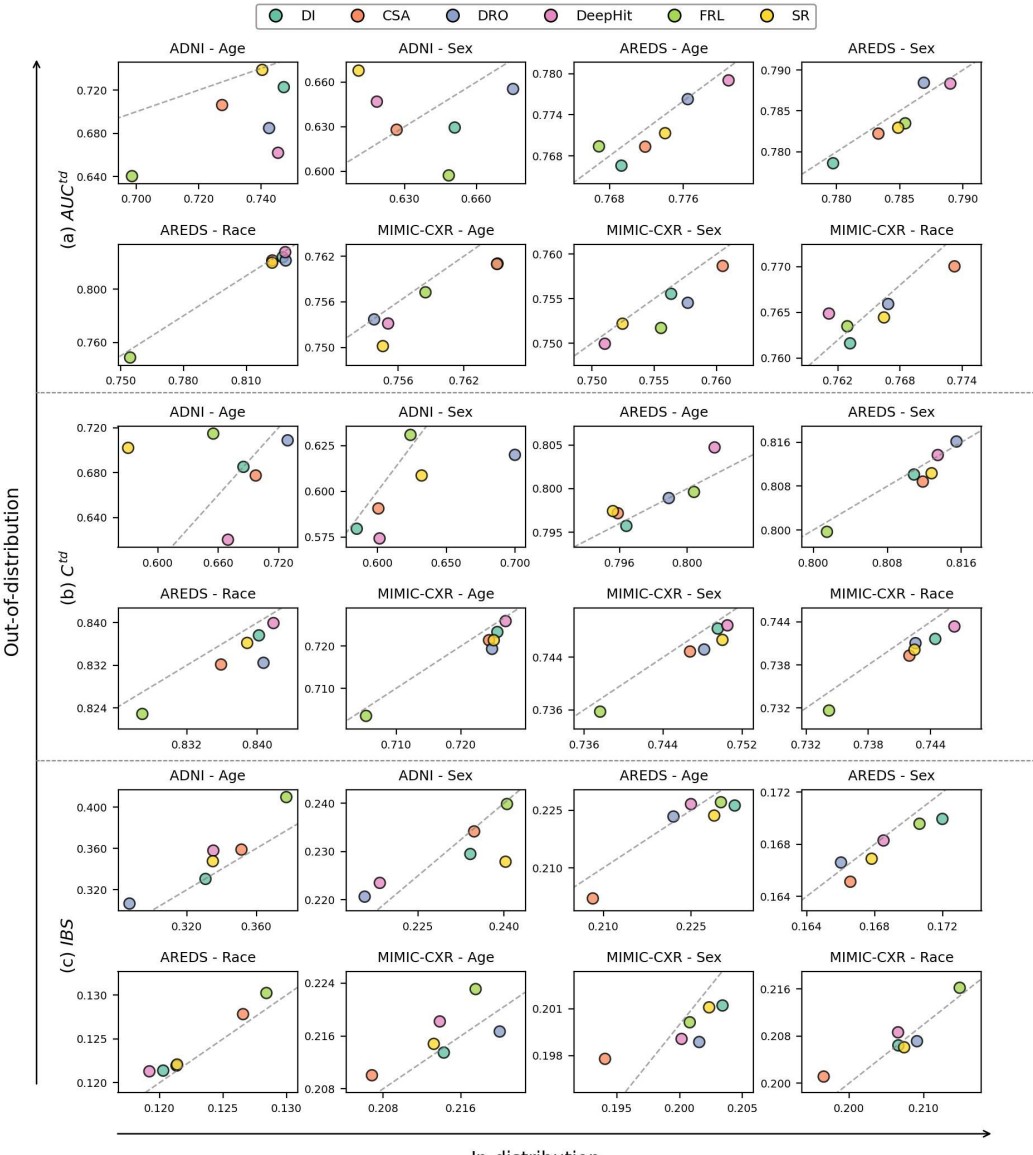

Figure A25: Comparison of predictive performance on the intervened group for (fair) TTE prediction models in in-distribution vs. out-of-distribution (i.e., shift in $\Delta$) learning scenarios across all dataset and sensitive attribute combinations. The displayed results represent the average performance across all random seeds. Points on the dashed line indicate equal performance in both scenarios. a) Results for $AUC^{td}$; b) Results for $C^{td}$; c) Results for $IBS$.

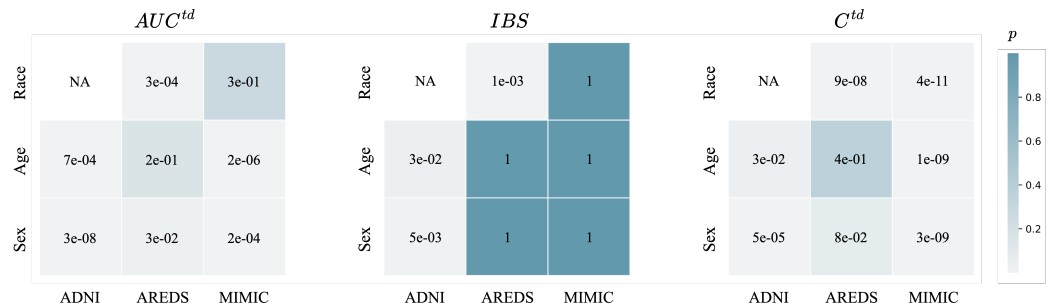

Figure A26: P-values from the one-sided Wilcoxon signed-rank test computed across all fair TTE prediction models and random seeds. A p-value < 0.05 suggests distribution shift on $\Delta$ significantly degrades TTE predictive performance compared no distribution shift.

