# OpenReview forum: "The Boundaries of Fair AI in Medical Image Prognosis: A Causal Perspective"
_NeurIPS.cc/2025/Conference — NeurIPS 2025 poster_

### Official Review · Reviewer_PBKE · 2025-06-27

**Clarity:** 4
**Significance:** 2
**Originality:** 2
**Rating:** 4
**Confidence:** 5

**Summary:**

The paper introduces a framework for assessing fairness in time-to-event settings (in the setting of censored observations), where the prediction of event times is based on image data and possibly other covariates. The fairness criterion considered looks at the difference in the error rates between demographic groups. The paper's results include bounds for the fairness metric in terms of appropriate group separability quantifications, and a decomposition of differences between prediction distributions into various contributions.

**Questions:**

- (Q1) The decomposition is Equation (4) claims to decompose a disparity. Usually, statistical or causal disparities are quantified through contrasts, such as differences/ratios of conditional expectations/potential outcomes. Curiously, in Equation (4), the LHS $P(y, \delta \mid x, a)$ does not seem to be a contrast. Clarifying this would be helpful (e.g., are ratios considered for different values of $a$?).
- (Q2) Is there a connection of the decomposition with causal decompositions appearing in the causal fairness literature, e.g. [3]?
- (Q3) It would be interesting to understand whether there is a more fundamental argument for the proposed fairness criterion. In the text, there are arguments about the relation of distribution shift and fairness. What about the following simple example
$$
\begin{align}
A &= \text{Bernoulli}(0.5) \\\\
Y &= A + (1+A) \epsilon,
\end{align}
$$
where $\epsilon$ follows an exponential distribution. In the above example (reflecting heteroscedastic noise across groups), why would equal error in prediction across groups be a useful criterion, if prediction is inherently more difficult for one group?

[3] Zhang, Junzhe, and Elias Bareinboim. "Fairness in decision-making—the causal explanation formula." Proceedings of the AAAI Conference on Artificial Intelligence. Vol. 32. No. 1. 2018.

**Ethical Concerns:**

["NO or VERY MINOR ethics concerns only"]

**Final Justification:**

Doing state-of-the-art medical survival analysis is impossible without considering informative censoring and competing risks, which the paper does not handle.

However, the paper does move the current baseline significantly in the right direction, offering a first at-scale evaluation of fairness in TTE prediction.

**Limitations:**

No, the limitations are not adequately addressed (see Weakness (1)), and some of the paper's claims seem to be stronger than the results and methodology support.

**Paper Formatting Concerns:**

None.

**Quality:**

3

**Strengths And Weaknesses:**

Strengths:
- (S1) The paper deals with an important, and timely topic of research,
- (S2) The paper is well-written and clear,
- (S3) Some parts of the analysis seems to be offering a novel approach,
- (S4) This is visibly a high-effort submission, with extensive empirical analyses.

Weaknesses:
- (W1) Survival analysis for medical data is plagued with multiple serious challenges, with some of the most significant issues being informative censoring [1] and competing risks [2]. The methodology proposed in the paper does not even mention these issues, or attempts to address such problems, which are known to be key for state-of-the-art survival analysis in medical data. This is also reflected in the empirical data analyses. For instance, in the MIMIC-CXR dataset, patients are censored 1-year after their last discharge. However, hospital discharge is not treated as censoring, meaning that patients remain in the risk set, even though they are not really at risk. A core issue is that there is no information in the dataset about mortality outcomes outside the hospital. Furthermore, competing risks are an important consideration, since admission patterns differ significantly for racial minorities, and different groups have different competing risk after hospital discharge. Thus, it remains unclear if the proposed analysis is appropriate. Similarly, the setting considered for the ADNI dataset seems to include events that can only happen on patient visits, which is a setting where interval censoring is commonly considered. In summary, non-informative censoring, which is assumed throughout the paper, is a major limitation that appears to be glossed over.

- (W2) Another point that may confound the reader are the claims about offering a causal perspective. Figures 2 and 3 seem to discuss the causal structure, but it remains unclear whether anything related to causal mechanisms is discussed. The figures specify DAGs that correspond to different settings that could be true or not, but these graphs are not used for any identification purposes (or specifying assumptions?). Furthermore, in the decomposition of Equation (4), the terms appearing seem to talk about mutual information, which are also not causal quantities, and the decomposition seems largely unrelated to different causal mechanisms (or at least this remains unclear). It seems from a distance that the analysis could rest on Bayesian networks entirely (relying only on notions of observational distributions), which makes the framing of a "causal perspective" somewhat unusual.

References

[1] Leung, Kwan-Moon, Robert M. Elashoff, and Abdelmonem A. Afifi. "Censoring issues in survival analysis." Annual review of public health 18.1 (1997): 83-104.

[2] Andersen, Per Kragh, et al. "Competing risks in epidemiology: possibilities and pitfalls." International journal of epidemiology 41.3 (2012): 861-870.

---

> ### Author Rebuttal · Authors · 2025-07-30
>
> Thank you for your thoughtful review. We appreciate your recognition that our work deals with an important and timely topic, offers novel approach, and provides extensive empirical analysis. We would now like to address your remaining concern below:
>
> > **Q1. Regarding the challenges of TTE prediction such as informative censoring and competing risks.**
>
> We completely agree that informative censoring and competing risks present important challenges in TTE prediction. However, we would like to clarify that these challenges arise primarily from the perspective of TTE modeling itself, rather than from the fairness perspective, which is the main focus of our study.
>
> In this work—the first comprehensive benchmark of fairness in medical image prognosis—we investigate the fairness behavior of data and models under the standard TTE prediction setting, which assumes a single risk and non-informative right censoring. This setting already presents significant experimental challenges, especially given the constraints of publicly available medical image datasets. Constructing datasets suitable for studying informative censoring or competing risks would require substantial effort during data collection and annotation, which we believe is currently not feasible at scale.
>
> That said, we greatly appreciate your valuable suggestion and will consider exploring these directions in future work. We also emphasize that our proposed framework FairTTE is flexible and can incorporate a wide range of TTE models and datasets, including those designed for settings involving competing risks or informative censoring, when such data become available. We will include a discussion of these future directions in the revised version of the manuscript.
>
> > **Q2. Regarding causal perspective used in our study.**
>
> We would like to clarify that the causal graphs proposed in our study serve as conceptual tools to illustrate how sources of bias may arise in medical image data—specifically, when there are causal pathways between sensitive attributes and other variables in the graph. In addition, our study focuses on group fairness notions, which are quantified using observed data distributions, rather than on counterfactual fairness, which requires causal path identification. As such, the causal graph is intended to aid in interpreting why disparities exist across subgroups, not to serve as a target for complete causal identification.
>
> > **Q3. Regarding decomposition formula.**
>
> As mentioned in our response to Q2, our study focuses on group fairness notions, which are quantified based on observed data distributions. The decomposition formula is designed to break down disparities between group-specific data distributions into quantifiable terms, enabling us to systematically analyze group fairness in our experiments. In other words, the decomposition formula is not related to causal or counterfactual fairness concepts explored in works such as [1], but rather provides a descriptive tool grounded in observed data.
>
> > **Q4. Regarding the appropriateness of fairness notion in medical domain.**
>
> We would like to clarify that there is no universal fairness notion suitable for all scenarios. Selecting an appropriate fairness criterion in the medical domain requires careful consideration of clinical context, ethical principles, and statistical validity. An effective fairness notion should align with the intended use of the model, its potential impact on different patient subgroups, and real-world constraints.
>
> In our study, we adopt group fairness metrics, such as performance gaps across subgroups, because they are widely used in medical imaging tasks and offer clinically meaningful interpretations [2,3,4]. However, we acknowledge that this does not imply such metrics are appropriate for all medical applications. In contexts where prediction is inherently more challenging for one group, as you mentioned, minimizing performance gaps may inadvertently reduce performance for the advantaged group, which can be counterproductive. In such cases, maximizing subgroup-wise performance may be a more suitable objective.
>
> Furthermore, by examining fairness through the lens of the data generation process, as we do in our study, we offer a principled framework for understanding the sources of bias. This perspective can help guide the selection of fairness notions that are best suited to the specific characteristics and ethical goals of a given application.
>
> **References**
>
> [1] Zhang, Junzhe, and Elias Bareinboim. "Fairness in decision-making—the causal explanation formula." Proceedings of the AAAI conference on artificial intelligence. Vol. 32. No. 1. 2018.
>
> [2] Seyyed-Kalantari, Laleh, et al. "Underdiagnosis bias of artificial intelligence algorithms applied to chest radiographs in under-served patient populations." Nature medicine 27.12 (2021): 2176-2182.
>
> [3] Vaidya, Anurag, et al. "Demographic bias in misdiagnosis by computational pathology models." Nature Medicine 30.4 (2024): 1174-1190.
>
> [4] Tian, Yu, et al. "FairSeg: A Large-Scale Medical Image Segmentation Dataset for Fairness Learning Using Segment Anything Model with Fair Error-Bound Scaling." The Twelfth International Conference on Learning Representations (2024)

---

> > ### Comment · Reviewer_PBKE · 2025-08-04
> > **Further follow-up**
> >
> > I thank the authors for their detailed response. Here are some final short follow-ups:
> >
> > ---
> >
> > 1. Q: Competing risks & informative censoring
> >
> > It would be great if the authors could share specifically text that they intend to use to describe these Limitations.
> >
> > ---
> >
> > 2. Q: Decomposition formula
> >
> > I understand that the decomposition formula is not intended to be causal or counterfactual. However, could you please clarify the original question, which was
> >
> > > The decomposition is Equation (4) claims to decompose a disparity. Usually, statistical or causal disparities are quantified through contrasts, such as differences/ratios of conditional expectations/potential outcomes. Curiously, in Equation (4), the LHS does not seem to be a contrast.
> >
> > To be sure, even if a statistical group fairness notion is used, one still expects to see a contrast, in order to be able to compare different groups.
> >
> > ---
> >
> > 3. Regarding the appropriateness of fairness notion in medical domain
> >
> > Here, I hope that a better justification could be provided in future work. Specifically, from the current response, it seems that the choice of a fairness notion is almost entirely arbitrary.
> >
> > While I understand that different notions could be appropriate in different settings, being a bit more descriptive about when exactly the notion proposed in the paper is justified would strengthen the presented work. Currently, it seems that this aspect has not been emphasized.

---

> > > ### Author Response · Authors · 2025-08-05
> > > **Further Response to Reviewer PBKE (Part 1 of 2)**
> > >
> > > Thank you for your timely feedback. We would like to address your remaining concerns as follows.
> > >
> > > > **Q1. Regarding the limitation.**
> > >
> > > We plan to include the following limitation paragraph in our revised version:
> > >
> > > > While FairTTE represents a significant step forward in advancing fairness research in medical image prognosis by providing the first comprehensive benchmark across multiple datasets and fairness algorithms, we acknowledge several limitations in the current study. Specifically, our analysis focuses on the standard TTE prediction setting, which assumes a single clinical risk and non-informative right censoring. In real-world clinical applications, however, patients often face multiple competing risks and may drop out of studies for reasons that introduce informative censoring, making the fairness landscape considerably more complex. Addressing fairness in such settings remains an important direction for future research. Additionally, although we adopt group fairness definitions (i.e., minimizing performance gaps across subgroups), we recognize that strictly enforcing these criteria can, in some clinical scenarios, degrade overall utility or harm performance for all groups. In such scenarios, alternative fairness notions may be more appropriate, depending on the clinical context and ethical objectives.
> > >
> > > > **Q2. Regarding the decomposition formula.**
> > >
> > > We would like to clarify that the decomposition formula presented in Equation (4) is designed to break down group-specific labeling functions into quantifiable components, which are then used to estimate sources of bias. Details on how bias sources are computed from these components in practice are provided in **Appendix F.4** of the manuscript.
> > >
> > > To illustrate how Equation (4) facilitates the quantification of bias sources, consider a simplified setting with two subgroups, $a_1$ and $a_2$. According to **Theorem 1**, statistical group fairness may be violated—i.e., a performance gap exists between the subgroups—when there are disparities in the distributions $P(x_z \mid a_1)$ vs. $P(x_z \mid a_2)$, and $P(y, \delta \mid x, a_1)$ vs. $P(y, \delta \mid x, a_2)$.
> > >
> > > While differences in $P(x_z \mid a_1)$ and $P(x_z \mid a_2)$ can be directly quantified using statistical distance measures (e.g., Wasserstein distance, as used in **lines 1418–1425** of **Appendix F.4**), directly comparing the conditional distributions $P(y, \delta \mid x, a_1)$ and $P(y, \delta \mid x, a_2)$ is more challenging.
> > >
> > > To address this, we leverage the decomposition formula to express the conditional distributions as follows:
> > >
> > > $$
> > > P(y, \delta \mid x, a_1) = \frac{P(x_z \mid y, \delta, a_1)}{P(x_z \mid \delta, a_1)} \cdot \frac{P(x_z \mid \delta, a_1)}{P(x_z \mid a_1)} \cdot P(y \mid \delta, a_1) \cdot P(\delta \mid a_1)
> > > $$
> > >
> > > $$
> > > P(y, \delta \mid x, a_2) = \frac{P(x_z \mid y, \delta, a_2)}{P(x_z \mid \delta, a_2)} \cdot \frac{P(x_z \mid \delta, a_2)}{P(x_z \mid a_2)} \cdot P(y \mid \delta, a_2) \cdot P(\delta \mid a_2)
> > > $$
> > >
> > > Rather than directly measuring the disparity between the full conditional distributions, we instead quantify disparities between the following components:
> > >
> > > 1. $\frac{P(x_z \mid y, \delta, a_1)}{P(x_z \mid \delta, a_1)}$ vs. $\frac{P(x_z \mid y, \delta, a_2)}{P(x_z \mid \delta, a_2)}$
> > > 2. $\frac{P(x_z \mid \delta, a_1)}{P(x_z \mid a_1)}$ vs. $\frac{P(x_z \mid \delta, a_2)}{P(x_z \mid a_2)}$
> > > 3. $P(y \mid \delta, a_1)$ vs. $P(y \mid \delta, a_2)$
> > > 4. $P(\delta \mid a_1)$ vs. $P(\delta \mid a_2)$
> > >
> > > All four of these components are **practically quantifiable** from observed data. The methodology for estimating these terms, including the generalization to settings with multiple subgroups, is described in detail in **Appendix F.4**. Specifically, the calculation for term (1) is provided in **lines 1404–1409**, term (2) in **lines 1410–1413**, term (3) in **lines 1414–1417**, and term (4) in **lines 1426–1428**.

---

> ### Author Response · Authors · 2025-08-05
> **Further Response to Reviewer PBKE (Part 2 of 2)**
>
> > **Q3. Regarding the appropriateness of fairness notion in medical domain**.
>
> Thank you for your valuable feedback. Based on our discussion, we plan to include the following paragraph in the revised version of our manuscript to clarify our rationale for the fairness criterion used in this study:
>
> > Selecting an appropriate fairness criterion in the medical domain requires careful consideration of the clinical context, ethical principles, and statistical validity. An effective fairness notion should align with the model’s intended use, its potential impact on different patient subgroups, and real-world constraints. In our study, we focus on statistical group fairness (i.e., predictive performance gaps across subgroups) which is widely examined in the medical image analysis literature. This choice is grounded in the assumption that any causal pathway from sensitive attributes represents an unfair influence in the causal graph and should be mitigated through fairness constraints. However, we acknowledge that in practical clinical scenarios, such causal pathways may reflect fair and clinically meaningful relationships. Enforcing group fairness in these cases may inadvertently remove relevant information and degrade predictive performance.
>
> > Moving forward, we suggest several directions for fairness research in medical image analysis: (1) **Identifying the causal nature of bias** to distinguish between fair and unfair sources, enabling models to address specific pathways appropriately; (2) **Developing fairness metrics and mitigation strategies** that preserve clinically relevant (fair) pathways while minimizing the effect of unfair ones; and (3) **Collaborating with clinicians and domain experts** to define context-specific fairness objectives that are aligned with both clinical utility and ethical standards.
>
> In addition to including this paragraph, we will also include the **limitations section** in the revised version to clearly acknowledge the scope of our fairness notion and to highlight the potential trade-offs in enforcing group fairness (see our response to Q1).

---

> ### Comment · Reviewer_PBKE · 2025-08-07
> **Thanks for clarifying**
>
> Thanks, the response is appreciated, I have increased the score from 3 to 4.
>
> While I am still of the opinion that doing state-of-the-art medical survival analysis is almost impossible without considering informative censoring and competing risks, it is true that the landscape is complex and one has to start from somwhere, and your effort is moving significantly in the right direction.

---

> > ### Author Response · Authors · 2025-08-08
> >
> > Thank you for your thoughtful follow-up and for increasing your score. We truly appreciate your recognition that our work is a step in the right direction, even though important challenges such as informative censoring and competing risks remain. We fully agree that these aspects are critical for advancing state-of-the-art medical survival analysis, and we view our current study as laying the groundwork for addressing these complexities in future research. We will also incorporate the key points discussed with you into the revised version of our manuscript to better clarify the scope, limitations, and future directions of our work. Your feedback has been invaluable in refining both the framing and presentation of our study.

---

### Official Review · Reviewer_jaCt · 2025-06-29

**Clarity:** 4
**Significance:** 3
**Originality:** 4
**Rating:** 5
**Confidence:** 4

**Summary:**

This paper introduces FairTTE, a comprehensive causal framework for investigating and quantifying fairness in time-to-event (TTE) prediction within medical imaging. Prior fairness research in medical AI has focused largely on diagnostic tasks (classification, segmentation). This work targets prognosis tasks, with additional challenges due to censoring, lack of standard fairness metrics, and complex data generation processes. The paper combines a theoretical causal analysis to decompose sources of bias with empirical studies on three real-world datasets (MIMIC-CXR, ADNI, AREDS). The theoretical analysis shows how bias arises, and extensive experiments corroborate that bias is pervasive across demographic groups. The analysis shows that existing fairness algorithms only partially mitigate bias and distribution shifts pose further challenges.

**Questions:**

-	Can the authors elaborate on how they envision FairTTE being used by practitioners—e.g., in regulatory audits, model certification, or dataset curation?

**Ethical Concerns:**

["NO or VERY MINOR ethics concerns only"]

**Final Justification:**

I have read the reviews and feedback.
I tend to think this paper brings some insight. While there are inevitably some weaknesses, none seem to be fatal. Therefore it could be accepted in my opinion.

**Limitations:**

OK.

**Paper Formatting Concerns:**

No problem.

**Quality:**

3

**Strengths And Weaknesses:**

Strengths:

1.	Prognosis is potentially even more important than diagnosis in medical AI, and fairness in this area is substantially understudied compared to fair diagnosis.

2.	Compared to many empirical papers on fairness, this paper deposes the ways in which bias can arise using structural causal models, which provides a clearer way to understand and think about this important problem.

3.	Empirical validation of state-of-the-art TTE and fairness methods is thorough, with good statistical analysis. I appreciate this good HPO protocol and statistical testing compared to many more ad-hoc papers in this area.

4.	Good that data and code are open for reproducibility.

5.	Insightful Findings: The theoretical analysis is helpful for thinking about and understanding the challenges of fairness. And the empirical study has interesting results such as the limited ability of existing fairness methods to ameliorate bias, a breakdown of the different sources of bias, and the fragility of fairness interventions under distribution shift.

6.	Generally the paper is very well written, if a bit dense at times. Including the appendix, it’s very thorough, and comprehensive.

7.	I  really appreciate the practical examples of the SCMs in App G2.



Weaknesses:

1.	Missed insight opportunity: It would have been great to have some suggestions about how to develop new algorithms or improve existing algorithms based on the insights of the SCM analysis.  Alternatively, some more concrete analysis on what is the limit of the best achievable fairness given the practical causal relations that mix the influence of conditions and sensitive attributes.

2.	Missed insight opportunity: In terms of the causal analysis of bias, it seems that some aspects are in common with standard diagnosis (eg: different image distribution across groups), and others are unique to prognosis (eg: Different TTE distribution across groups). It would have been nice to have an explicit compare & contrast to see what’s the unique part in this work that differs from standard SCM perspective on diagnosis.

3.	On comparing images: (i) L106-110 claims that medical images are high-dimensional points that are hard to compare. But then as part of the bias analysis (Fig5, App F4), actually the paper directly compares images in embedding space using Wasserstein distance. (ii) In F4: Is comparing the images in embedding space what we should do? It seems a bit odd, because it depends on the feature representation that each deep network learns, which will be different for different prognosis and mitigation algorithms.

4.	Evaluation: The test-set bootstrapping for CI calculation (Fig4) seems like not the important source of variability. My experience is that especially for small-ish medical datasets (eg: AREDS, ADNI), sensitive fairness algorithms, and  automated HPO, then the huge source of variability is in seeds for the training phase.

---

> ### Author Rebuttal · Authors · 2025-07-29
>
> Thank you for your thoughtful review. We appreciate your recognition that our work addresses an important problem in medical AI, presents a thorough evaluation, and offers insightful findings. We would now like to address your remaining concern below:
>
> > **Q1. Regarding suggestions to develop new fairness algorithm for TTE prediction based on the causal graph.**
>
> Thank you for this valuable feedback. We offer several suggestions for future research on fairness in medical image prognosis:
>
> - Identifying the causal nature of bias is essential for distinguishing between fair and unfair sources, enabling models to adjust for specific pathways rather than treating all disparities uniformly.
>
> - Developing appropriate bias mitigation strategies and fairness metrics that leverage fair causal pathways while minimizing the influence of unfair ones.
>
> - Collaborating with clinicians and domain experts to define application-specific fairness objectives ensures that fairness interventions align with clinical utility and ethical standards.
>
> In our opinion, a holistic approach to fairness should span the entire AI development pipeline (i.e., from data collection and fairness metric selection to model architecture and evaluation) to effectively mitigate bias in real-world applications.
>
> For example, if we identify an unfair causal pathway between a sensitive attribute and the underlying health condition during the data collection process, techniques such as fair representation learning may help remove spurious correlations. However, if the sensitive attribute is a clinically meaningful risk factor for the TTE outcome, the model should be encouraged to extract sensitive attribute-related features from the image to calibrate its predictions appropriately. In such cases, applying fairness methods that obscure this information like fair representation learning could inappropriately degrade performance by forcing the model to ignore clinically relevant signals.
>
> We will incorporate this discussion in the revised manuscript to highlight the potential impact of our framework beyond benchmarking.
>
> > **Q2. Regarding comparison with medical image diagnosis scenario.**
>
> We clarify that there are both shared and distinct aspects in the causal analysis of bias between diagnosis and prognosis tasks. While causal analysis has been applied to study biases in diagnostic settings, prognostic tasks introduce unique challenges that require specialized consideration. Specifically, TTE prediction involves longitudinal outcomes and censoring, which fundamentally differ from the static labels used in diagnosis. These temporal aspects introduce additional causal pathways through which bias can arise, such as disparities in event timing or censoring mechanisms across subgroups.
>
> Among the five sources of bias identified in our framework, disparities in image feature distributions have been explored in prior works on medical image diagnosis [1,2]. However, the remaining sources, particularly those related to TTE outcome and censoring, are unique to prognosis.
>
> We will make these distinctions more explicit in the revised manuscript to highlight how our framework extends fairness analysis in prognostic modeling beyond the conventional diagnostic setting.
>
> > **Q3. Regarding image comparison.**
>
> We would like to clarify that the image comparisons discussed in Section 2.1 (lines 106-116) and Appendix F.4 (lines 1418-1425) are conducted in different contexts. Specifically, in Section 2.1, the goal is to define a fairness metric that requires measuring distances between images in a clinically meaningful way, which is inherently a challenging task. In contrast, the analysis in Appendix F.4 involves comparing image features from a statistical perspective to quantify sources of bias. In this latter case, using a statistical distance measure such as the Wasserstein distance is sufficient and appropriate for the intended analysis.
>
> > **Q4. Regarding source of variability.**
>
> We would like to clarify that, in addition to calculating confidence intervals using test-set bootstrapping, we also performed statistical significance testing across results from all random seeds in our experiments to verify the robustness of our findings to training variability induced by random initialization.
>
> > **Q5. Regarding the practical utility of FairTTE.**
>
> We would like to clarify that FairTTE can be used as a versatile tool that can support practitioners in multiple stages of the ML pipeline for medical prognosis. Specifically:
>
> - FairTTE provides a standardized framework to evaluate fairness across subgroups under realistic clinical conditions, such as distribution shifts and censoring. Regulators could use FairTTE to assess whether a TTE prediction model exhibits consistent performance and equitable treatment across demographic groups, forming part of an evidence base for approval or deployment.
>
> - FairTTE also enables practitioners to systematically identify sources of bias, whether from image features, outcome disparities, labeling process, or censoring mechanisms. This can guide unbiased dataset curation process.
>
> - Developers can use FairTTE to benchmark the fairness behavior of candidate models or fairness-aware training strategies across multiple axes (datasets, attributes, metrics). This allows for more informed model selection and ensures that fairness-performance trade-offs are clearly understood before deployment.
>
> Ultimately, we see FairTTE as a practical and extensible framework that can be integrated into the clinical ML development lifecycle.
>
> **References**
>
> [1] Jones, Charles, Mélanie Roschewitz, and Ben Glocker. "The role of subgroup separability in group-fair medical image classification." International Conference on Medical Image Computing and Computer-Assisted Intervention. Cham: Springer Nature Switzerland, 2023.
>
> [2] Yang, Yuzhe, et al. "The limits of fair medical imaging AI in real-world generalization." Nature Medicine 30.10 (2024): 2838-2848.

---

### Official Review · Reviewer_Z8y7 · 2025-07-02

**Clarity:** 3
**Significance:** 4
**Originality:** 3
**Rating:** 5
**Confidence:** 2

**Summary:**

The authors propose a a comprehensive framework, FairTTE for assessing and analyzing fairness in TTE prediction for medical imaging.  The FairTTE framework involved training / evaluating thousands of TTE models across 3 major medical imaging datasets, diverse fairness algorithms and TTE algorithms, and performing causal analysis to identify and quantify five primary sources of bias in TTE datasets (image feature distributions, TTE distributions, censoring rates, and in the MI between images and outcomes).

The authors report several key findings:
- consistent performance gaps observed between different demographic groups across all tested imaging modalities and datasets.
- current SOTA algorithms often fail to consistently mitigate bias, and any gains in fairness are frequently accompanied by a decrease in the model's predictive accuracy.
- pre-training models on large datasets improves their accuracy but has a minimal impact on fairness.
- maintaining fairness becomes increasingly difficult under distribution shifts

**Questions:**

potentially consider adding pathology as an additional medical imaging modality given recent rise in popularity and adoption of digital pathology, and earlier work investigating fairness in various digital pathology learning algorithms and pretrained foundation models. e.g. Vaidya, A., Chen, R.J., Williamson, D.F.K. et al. Demographic bias in misdiagnosis by computational pathology models. Nat Med

That said, I am more than impressed by the current scope of investigation by the authors.

**Ethical Concerns:**

["NO or VERY MINOR ethics concerns only"]

**Final Justification:**

No change in assessment. The authors did perform additional experiments on pathology image data but I acknowledge the current scope is already sufficient.

**Limitations:**

Yes.

**Quality:**

4

**Strengths And Weaknesses:**

Strengths
- solid theoretical framework, large-scale comprehensive evaluation, clear presentation

Weaknesses
- nothing major, but could benefit from expanding to the scope of medical imaging to include pathology if time permits.

---

> ### Author Rebuttal · Authors · 2025-07-29
>
> Thank you for your thoughtful review. We appreciate your recognition of our work's solid theoretical framework, large-scale comprehensive evaluation, and clear presentation. We would like to address your remaining concern below:
>
> > **Q1. Regarding experiments with pathology images.**
>
> Thank you for the suggestion. Pathology images indeed play a critical role in medicine by providing detailed visual information about tissue morphology, which is essential for accurate disease diagnosis, such as in cancer. However, to the best of our knowledge, existing publicly available pathology image datasets lack either sensitive attributes or longitudinal clinical outcomes—both of which are necessary to study fairness in TTE prediction. For example, the dataset referenced in [1] does not include longitudinal outcomes, making it unsuitable for TTE analysis. Additionally, due to the time constraints of the rebuttal phase and the complexity involved in accessing (e.g., controlled access) and processing pathology images (e.g., whole-slide image handling, linking images with clinical outcomes) from sources such as TCGA [2], conducting a large-scale analysis using these data is currently not feasible. We will continue to monitor developments in this area and plan to incorporate pathology image datasets into our future studies as soon as appropriate resources become available.
>
> **References**
>
> [1] Vaidya, Anurag, et al. "Demographic bias in misdiagnosis by computational pathology models." Nature Medicine 30.4 (2024): 1174-1190.
>
> [2] https://www.cancer.gov/ccg/research/genome-sequencing/tcga

---

### Official Review · Reviewer_g759 · 2025-07-02

**Clarity:** 3
**Significance:** 3
**Originality:** 2
**Rating:** 4
**Confidence:** 3

**Summary:**

The authors propose a comprehensive framework, FairTTE, for evaluating and understanding fairness in medical image prognosis. They formalize a causal decomposition of bias sources in TTE prediction (e.g. disparities in image features, censoring, event labels, etc.) and use it to explain why fairness is hard to achieve in this setting. To support empirical study, the paper adopts three diverse medical imaging datasets with TTE outcomes: predicting in-hospital mortality from chest X-rays (MIMIC-CXR), late-stage AMD from retinal fundus images (AREDS), and Alzheimer’s progression from brain MRIs (ADNI). The experiments observe that current fairness interventions provide only limited mitigation. While some methods reduce group gaps, they often hurt overall accuracy, and none consistently outperform a baseline model in fairness. The paper also highlights that pre-training on large datasets improves overall TTE accuracy but does not consistently improve fairness, and that distribution shifts (changes between train/test data) further exacerbate unfairness.

**Questions:**

1. Given that no existing methods work consistently, what do the authors envision as potential holistic approaches to address all bias sources?
2. The paper adopts TTE models that assume discretized time intervals for event prediction. Could the authors clarify how the choice of discretization granularity (e.g., interval width or number of bins) impacts both fairness and predictive accuracy?

**Ethical Concerns:**

["NO or VERY MINOR ethics concerns only"]

**Limitations:**

Yes

**Quality:**

3

**Strengths And Weaknesses:**

Strengths:
1. Medical prognostic model with fairness learning is an important yet underexplored direction.
2. The paper provides a theoretical causal model and bias decomposition that rigorously characterize how sensitive attributes can influence TTE predictions.
3. The authors collect three real-world publicly datasets spanning different imaging modalities and outcomes, each annotated with demographic attributes. This is significant to provide a benchmark for fairness in medical image prognosis.
4. In addition, the evaluation is thorough. The authors train and evaluate a large number of models with varying sensitive attributes (race, sex, age), model types, and fairness algorithms.

Weaknesses:
1. The paper is primarily an analysis and benchmarking effort. It does not propose a novel algorithm to improve fairness in TTE predictions.
2. The causal bias framework distinguishes “fair” vs “unfair” pathways. While conceptually sound, it relies on unobservable constructs (true health state Z, latent confounders U) and strong assumptions (e.g., whether data disparities reflect spurious correlations or true causal effects of A). In practice, it may be hard to definitively categorize which biases are “unfair” (e.g., if age genuinely affects disease progression vs. if it’s a proxy for unequal care).
3. The authors simulate distribution shifts by adding noise or flipping censoring indicators for one group. While this stress-test is useful, it is somewhat artificial. The paper claims to base these on real medical scenarios, but the main text doesn’t elaborate much on how realistic these shifts are.

---

> ### Author Rebuttal · Authors · 2025-07-29
>
> Thank you for your thoughtful review. We appreciate your recognition that our work addresses an important yet underexplored problem and that our evaluation is thorough. We would now like to address your remaining concern below:
>
> > **Q1. Regarding the proposal of a new fairness method for TTE prediction.**
>
> We would like to clarify that our primary contribution lies in providing a novel and systematic framework to analyze the fairness behavior of (fair) TTE prediction models in medical image prognosis—a critical yet underexplored area. To the best of our knowledge, this is the first comprehensive study that rigorously examines how existing (fair) TTE prediction methods perform across different datasets, sensitive attributes, and fairness metrics under clinically relevant distribution shifts. Our framework enables the identification of key limitations in current approaches, offering actionable insights for the development of more robust and equitable models in the future. Thus, our work fills an important methodological gap and lays the foundation for principled fairness evaluation in medical image prognosis.
>
> > **Q2. Regarding identification of fair and unfair pathways in the causal graph.**
>
> We would like to clarify that the proposed causal graph serves as a conceptual tool to identify and explain potential sources of bias and to guide fairness analysis in a more structured manner. As stated in our main paper, our primary goal is to measure bias sources from medical images and TTE prediction models, which can be quantified using observed data; therefore, full identification of causal pathways is not required for our study. That said, we acknowledge your point that identifying causal pathways can be challenging in some medical contexts, and input from domain experts may be necessary to assess the fairness of specific causal pathways (as noted at the end of page 6 in our paper). Ultimately, fairness decisions should be informed by context-specific ethical and clinical considerations. Our framework is intended to support that reasoning—not to replace it.
>
> > **Q3. Regarding distribution shift settings.**
>
> We would like to clarify that distribution shifts created by intervening causal graph in our study mimic real-world scenarios as follows.
>
> - Adding noise to images mimics real-world scenarios such as patients in different geographic locations are scans with different equipment. This causes the medical image to appear systematically different for groups in each location.
>
> - Adding noise to TTE labels mimics real-world scenarios such as delayed or inaccurate event recording in EHR system.
>
> - Flipping censoring indices mimics real-world scenarios in which certain groups experience less consistent access to care due to financial or geographic. Thus, these groups are more likely to drop out of care, resulting in a higher censoring rate.
>
> By leveraging these data intervention mechanisms, we can analyze how different sources of bias impact the fairness behavior of TTE prediction models. While studying real-world distribution shifts is important, it is often challenging to identify the exact sources of bias and distribution shift mechanisms which complicates the fairness analysis. Based on your suggestion, we will add this discussion into our revised version.
>
> > **Q4. Regarding development of potential holistic approaches for fair TTE prediction.**
>
> We would like to clarify that a holistic approach to fairness should span the entire AI development pipeline (i.e., from data collection and fairness metric selection to model architecture and evaluation) to effectively mitigate bias in real-world applications. In particular, we provide some suggestions for future research on fairness in medical image prognosis as below:
>
> - Identifying the causal nature of bias is essential for distinguishing between fair and unfair sources, enabling models to adjust for specific pathways rather than treating all disparities uniformly.
>
> - Developing appropriate bias mitigation strategies and fairness metrics that leverage fair causal pathways while minimizing the influence of unfair ones.
>
> - Collaborating with clinicians and domain experts to define application-specific fairness objectives ensures that fairness interventions align with clinical utility and ethical standards.
>
> For example, if we identify an unfair causal pathway between a sensitive attribute and the underlying health condition during the data collection process, techniques such as fair representation learning may help remove spurious correlations. However, if the sensitive attribute is a clinically meaningful risk factor for the TTE outcome, the model should be encouraged to extract sensitive attribute-related features from the image to calibrate its predictions appropriately. In such cases, applying fairness methods that obscure this information like fair representation learning could inappropriately degrade performance by forcing the model to ignore clinically relevant signals.
>
> > **Q5. Regarding the impact of discretization granularity on both fairness and accuracy in TTE prediction models.**
>
> We would like to clarify that the choice of discretization granularity in TTE prediction models can influence both predictive accuracy and fairness outcomes. However, based on our preliminary experiments, varying the discretization scheme did not alter the key empirical findings of our study. Therefore, we chose to fix the discretization scheme across all experiments to isolate the effects of fairness interventions and distribution shifts on both fairness and accuracy in TTE prediction.
>
> To provide additional context, discretization granularity affects model behavior in several ways. Finer-grained discretization allows the model to capture more precise temporal information, potentially improving predictive accuracy. However, it can also increase model complexity and lead to data sparsity within individual bins, particularly for underrepresented subgroups, which may exacerbate disparities and fairness concerns. On the other hand, coarser discretization reduces data sparsity and simplifies the learning task, which can enhance model stability and, in some cases, mitigate fairness disparities. This benefit, however, comes at the cost of lower temporal resolution, which may limit the model’s ability to make accurate TTE predictions.
> We will expand this discussion in the revised manuscript to better clarify the trade-offs involved.

---

### Official Review · Reviewer_VK6y · 2025-07-03

**Clarity:** 3
**Significance:** 3
**Originality:** 3
**Rating:** 4
**Confidence:** 3

**Summary:**

The paper addresses a critical gap in fair AI research: while fairness in medical image diagnosis (e.g., classification, segmentation) has been studied, fairness in medical prognosis (predicting disease progression or outcomes over time) remains overlooked. Prognosis is often framed as time-to-event (TTE) prediction (e.g., time until death, disease recurrence). The authors argue that TTE prediction poses unique fairness challenges due to temporal dynamics, censored data (incomplete follow-ups), and the lack of standardized fairness metrics suited for survival analysis.

**Questions:**

It would be better to add a paragraph to the discussion section that explores the challenges and potential consequences of using alternative fairness metrics in this context. For example, what would be the clinical and ethical implications of enforcing "similar predicted outcomes" (demographic parity) for a condition where a sensitive attribute like age is a known and legitimate risk factor? Briefly exploring this "what if" scenario would demonstrate a comprehensive understanding of the broader fairness landscape and proactively address potential questions from readers about why certain metrics were not chosen.

**Ethical Concerns:**

["NO or VERY MINOR ethics concerns only"]

**Final Justification:**

The authors have addressed most of my comments. Due to the unclear complexity within TTE scenarios, I would retain my original rating.

**Limitations:**

Yes

**Paper Formatting Concerns:**

I did not identify any major formatting issues.

**Quality:**

3

**Strengths And Weaknesses:**

Strengths

- This study evaluates the methods on three large-scale public datasets (AREDS, MIMIC-CXR, ADNI) spanning multiple imaging modalities (fundus, X-ray, MRI) and clinical outcomes (AMD progression, mortality, Alzheimer’s).

- This study quantifies five distinct bias mechanisms (e.g., censoring disparities, feature distribution gaps) and shows their real-world prevalence.

- This work demonstrates that while pre-training improves accuracy, it does not inherently improve fairness, a crucial insight for medical AI.

Weaknesses

- The authors' large-scale evaluation, involving over 20,000 trained models, reveals that while these methods can sometimes reduce performance gaps, the improvements are not guaranteed and can be accompanied by a decrease in predictive accuracy. Table 1 demonstrates this inconsistency; for instance, in the ADNI-Sex setting, nearly all fairness algorithms, including DRO, SR, FRL, and CSA, show a negative impact on the Integrated Brier Score (IBS) accuracy compared to the DeepHit baseline.

- Similar to the point above, the paper notes that when sources of bias are small, applying these fairness methods can paradoxically worsen performance disparities. Statistical analysis confirms this, with Friedman and Nemenyi post-hoc tests showing that current algorithms do not offer a significant improvement in mitigating bias over the baseline DeepHit model. This indicates a fundamental gap in existing methodologies, which struggle to address multiple, co-existing sources of bias without a trade-off in utility.

- The theoretical analysis (page 4) relies on performance metrics that satisfy triangle inequality and symmetry properties, which, while described as "relatively mild" , may not hold for all metrics used in practice, a limitation the authors acknowledge. More critically, the paper's framework for identifying sources of bias relies on being able to quantify disparities in data distributions, but it concedes that identifying the true causal pathways that create these disparities is a significant challenge that requires deep clinical insight. The study simulates distribution shifts by synthetically corrupting images, TTE labels, or censoring indicators for specific groups. While this provides a controlled environment for analysis, these artificial shifts, such as applying a Gaussian blur or flipping 90% of censoring indicators, may not fully represent the complex, subtle, and multifaceted nature of distribution shifts encountered in real-world clinical settings.

- The study also finds that pre-training, a widely used technique for improving model performance, offers limited benefits for fairness. While pre-training was shown to improve predictive accuracy, especially for the smaller and more complex ADNI dataset , it had a "minimal impact on fairness". Statistical tests showed no significant improvement in fairness in 18 out of 24 experimental settings, suggesting pre-training alone is insufficient to produce equitable TTE predictions. This weakness implies that simply using larger, more general models is not a panacea for fairness issues and that dedicated fairness interventions are still necessary, even if they themselves are currently of limited effectiveness.

- The selection of fairness metrics, while justified, narrows the scope of the evaluation. The authors focus on metrics that ensure equal predictive performance across groups, arguing that other categories of fairness metrics are less suitable for medical imaging. They dismiss metrics requiring a similarity measure as "difficult to establish for medical images" and deem metrics that enforce similar predicted outcomes as potentially "nonsensical" when a sensitive attribute like age is a strong risk factor. While these are reasonable justifications, this focus excludes a broader discussion of individual fairness or alternative group fairness definitions that might be relevant, thereby constraining the paper's conclusions to a specific type of fairness.

---

> ### Author Rebuttal · Authors · 2025-07-29
>
> Thank you for your thoughtful review. We appreciate your recognition that our work addresses a critical gap in fair AI research. Below, we address your remaining concerns:
>
> > **Q1. Regarding the limitations of existing fairness methods for TTE prediction.**
>
> We would like to clarify that the observed limitations are not weaknesses of our work. Instead, our study is the first systematic investigation into the fairness behavior of existing (fair) TTE prediction algorithms. Through extensive evaluation, we identified several key limitations of current methods:
>
> - Fairness algorithms exhibit inconsistent performance across datasets, sensitive attributes, and fairness metrics.
>
> - Enforcing fairness may harms predictive performance, especially in case sources of bias are small, causing fairness-accuracy trade-off.
>
> - Pre-training improves model accuracy but has minimal impact on fairness.
>
> In sum, these findings highlight shortcomings in existing methods that our framework helps to reveal, not deficiencies in our own contribution.
>
> > **Q2. Regarding triangle inequality and symmetric assumptions for TTE performance metrics.**
>
> While some metrics such as the concordance index do not satisfy the triangle inequality or symmetry assumptions, they can be modified to do so. For example, prior work [1] proposed the symmetric discordance index, a variant of the concordance index that satisfies both assumptions. Moreover, these assumptions were introduced solely to facilitate the proof of Theorem 1 and do not affect our other theoretical results, including Proposition 2 and the decomposition formula in Equation (4), or our empirical findings.
>
> > **Q3. Regarding causal pathway identification.**
>
> We would like to clarify that causal pathway identification is not required for our work. Our goal is to measure bias from medical images which can be quantified from observed data. The causal graph serves as a conceptual tool to explain why disparities exist in data distributions across subgroups, not as a target for full identification.
>
> > **Q4. Regarding distribution shift settings.**
>
> We would like to clarify that distribution shifts created by intervening causal graph in our study mimic real-world scenarios as follows.
>
> - Adding noise to images mimics real-world scenarios such as patients in different geographic locations are scans with different equipment. This causes the medical image to appear systematically different for groups in each location.
>
> - Adding noise to TTE labels mimics real-world scenarios such as delayed or inaccurate event recording in EHR system.
>
> - Flipping censoring indices mimics real-world scenarios in which certain groups experience less consistent access to care due to financial or geographic. Thus, these groups are more likely to drop out of care, resulting in a higher censoring rate.
>
> By leveraging these data intervention mechanisms, we can analyze how different sources of bias and distribution shift types impact the fairness behavior of TTE prediction models. While studying real-world distribution shifts is important, it is often challenging to identify the exact sources of bias, which complicates the fairness analysis.
>
> > **Q5. Regarding fairness metrics for TTE prediction.**
>
> We would like to clarify that performance gaps across subgroups are appropriate fairness metrics and not narrows the scope of our study. These metrics are widely used in medical imaging tasks [2,3,4] and are clinically meaningful. While other fairness metrics exist in the TTE literature, they may not be appropriate for the clinical domain. For example:
>
> - Enforcing similar predicted TTE outcomes across subgroups may compromise clinical relevance and accuracy. A case study in Alzheimer’s disease illustrates this point well: requiring young and elderly individuals to have similar predicted times to disease onset would not make sense.
> - Enforcing similarity between similar patients is impractical in medical imaging, where well-defined measures of semantic similarity between images are lacking. This could lead to forcing similarity between inherently different cases.
>
> Based on your suggestion, we will expand the discussion of fairness metrics for medical image prognosis in the revised manuscript.
>
> **References**
>
> [1] Shaker, Ammar, and Carolin Lawrence. "Multi-source survival domain adaptation." Proceedings of the AAAI conference on artificial intelligence. Vol. 37. No. 8. 2023.
>
> [2] Seyyed-Kalantari, Laleh, et al. "Underdiagnosis bias of artificial intelligence algorithms applied to chest radiographs in under-served patient populations." Nature medicine 27.12 (2021): 2176-2182.
>
> [3] Vaidya, Anurag, et al. "Demographic bias in misdiagnosis by computational pathology models." Nature Medicine 30.4 (2024): 1174-1190.
>
> [4] Tian, Yu, et al. "FairSeg: A Large-Scale Medical Image Segmentation Dataset for Fairness Learning Using Segment Anything Model with Fair Error-Bound Scaling." The Twelfth International Conference on Learning Representations (2024).

---

> > ### Comment · Reviewer_VK6y · 2025-08-06
> > **Acknowledgement**
> >
> > Thank the authors for their responses.
> >
> > The responses might not fully address my or my fellows' questions, e.g.,
> >
> > Reviewer VK6y: Trade-off between fairness and utility is unresolved.
> >
> > Reviewer PBKE: Competing risks & informative censoring are ignored. No sensitivity analysis is provided to quantify how violating these assumptions would affect fairness conclusions.

---

> > > ### Author Response · Authors · 2025-08-06
> > >
> > > Thank you for your feedback. We would like to clarify the remaining points as follow.
> > >
> > > > **Q1. Regarding the Fairness–Accuracy Trade-off.**
> > >
> > > We would like to clarify that we have already conducted experiments to investigate the **fairness–accuracy trade-off** of existing fairness algorithms for TTE prediction. The results are presented in **Appendix H.5** of our manuscript. Specifically, **Figure A11** illustrates the trade-off in the IID setting, while **Figures A12, A13, and A14** demonstrate the trade-off under various distribution shift scenarios.
> > >
> > > We would also like to reiterate that the **core contribution** and **novelty** of our work lie in proposing a new framework for **systematic investigation** of the fairness behavior of existing (fair) TTE algorithms, rather than introducing a new fairness algorithm. We believe that offering a new perspective to deeply understand the problem is equally important as developing new methods. Within this scope, we analyze the fairness–accuracy trade-off exhibited by current approaches and identify their limitations in simultaneously achieving fairness and predictive performance.
> > >
> > > Although we do not propose a new algorithm, our work provides **valuable insights** that can guide the development of future methods aimed at improving the fairness–accuracy balance. As discussed in our rebuttals to reviewers **g759**, **jaCt**, and **PBKE**, we believe that the following directions are crucial for achieving better trade-offs:
> > >
> > > * **Identifying the causal nature of bias** is key to distinguishing between fair and unfair sources, allowing models to adjust for specific pathways rather than treating all disparities uniformly.
> > > * **Designing appropriate mitigation strategies and fairness metrics** that preserve fair causal pathways while minimizing the influence of unfair ones.
> > > * **Collaborating with clinicians and domain experts** to define application-specific fairness objectives, ensuring that interventions are aligned with clinical utility and ethical standards.
> > >
> > > > **Q2. Regarding informative censoring and competing risk.**
> > >
> > > As noted in our response to reviewer **PBKE**, we completely agree that **informative censoring** and **competing risks** present important challenges in TTE prediction. However, we would like to clarify that these challenges arise primarily from the perspective of **TTE modeling** itself, rather than from the **fairness** perspective, which is the main focus of our study.
> > >
> > > In this work—the first comprehensive benchmark of fairness in medical image prognosis—we investigate the fairness behavior of data and models under the **standard TTE prediction** setting, which assumes a **single risk** and **non-informative right censoring**. This setting already presents significant experimental challenges, especially given the constraints of publicly available medical image datasets. Constructing datasets suitable for studying informative censoring or competing risks would require substantial effort during data collection and annotation, which we believe is currently not feasible at scale.
> > >
> > > That said, we do not ignore these complexities. As stated in our response to reviewer **PBKE**, we plan to include the following paragraph in the revised manuscript to clarify the current scope and acknowledge these limitations:
> > >
> > > > While FairTTE represents a significant step forward in advancing fairness research in medical image prognosis by providing the first comprehensive benchmark across multiple datasets and fairness algorithms, we acknowledge several limitations in the current study. Specifically, our analysis focuses on the standard TTE prediction setting, which assumes a single clinical risk and non-informative right censoring. In real-world clinical applications, however, patients often face multiple competing risks and may drop out of studies for reasons that introduce informative censoring, making the fairness landscape considerably more complex. Addressing fairness in such settings remains an important direction for future research. Additionally, although we adopt group fairness definitions (i.e., minimizing performance gaps across subgroups), we recognize that strictly enforcing these criteria can, in some clinical scenarios, degrade overall utility or harm performance for all groups. In such scenarios, alternative fairness notions may be more appropriate, depending on the clinical context and ethical objectives.
> > >
> > > We hope this clarifies our position and the scope of our current work. We agree that extending fairness analysis to more complex TTE scenarios is a valuable and necessary direction for future research.

---

> > > > ### Author Response · Authors · 2025-08-08
> > > >
> > > > Dear reviewer VK6y,
> > > >
> > > > As the deadline for the author–reviewer discussion is approaching, we wanted to check in to see if our responses have fully addressed your concerns. Your feedback has been very helpful in refining the paper, and we would be happy to provide further clarification or additional details on any points that remain unclear.

---

### Decision · Program_Chairs · 2025-09-17

**Decision:**

Accept (poster)

**Comment:**

The paper introduces FairTTE, a unified framework for analyzing fairness in time-to-event prediction in medical imaging. The authors provide theoretical analyses from a causal perspective, and then run a large-scale study across multiple datasets, TTE models, and fairness interventions.

**Strengths**

Reviewers broadly agree this is an important, underexplored problem, and that the authors have conducted a thorough empirical evaluation. Writing and presentation are clear. and several reviewers view the work as impactful even without a new algorithm.

**Initial Weaknesses**

1. The most serious weakness is ignoring information censoring and competing risks (PBKE, VK6y).
2. The distribution-shift simulations may not reflect clinical reality (VK6y, g759).
3. The paper focuses on group fairness, which may not be appropriate in certain cases (PBKE, VK6y).

**Rebuttal Period**

The rebuttal provided many clarifications. The authors have acknowledged some of the limitations above, and promised to mention them in the revision. All reviewers had a positive assessment of the paper after the rebuttal period.

**Overall Evaluation**

I believe this is a comprehensive, principled framework that will be impactful to the field in shaping future proposed methods. While the survival analysis caveats are valid, the authors have acknowledged them in their rebuttals, and should include these limitations in the camera ready. I recommend acceptance.